# An activity-specificity trade-off encoded in human transcription factors

Julian Naderi[1,2,8], Alexandre P. Magalhaes [1,8], Gözde Kibar[3], Gregoire Stik [4,7], Yaotian Zhang[1], Sebastian D. Mackowiak [1], Hannah M. Wieler[1], Francesca Rossi [1], Rene Buschow [5], Marie Christou-Kent[4], Marc Alcoverro-Bertran[4], Thomas Graf [4,6], Martin Vingron [3] & Denes Hnisz [1] ✉

Transcription factors (TFs) control specificity and activity of gene transcription, but whether a relationship between these two features exists is unclear. Here we provide evidence for an evolutionary trade-off between the activity and specificity in human TFs encoded as submaximal dispersion of aromatic residues in their intrinsically disordered protein regions. We identified approximately 500 human TFs that encode short periodic blocks of aromatic residues in their intrinsically disordered regions, resembling imperfect prion-like sequences. Mutation of periodic aromatic residues reduced transcriptional activity, whereas increasing the aromatic dispersion of multiple human TFs enhanced transcriptional activity and reprogramming efficiency, promoted liquid–liquid phase separation in vitro and more promiscuous DNA binding in cells. Together with recent work on enhancer elements, these results suggest an important evolutionary role of suboptimal features in transcriptional control. We propose that rational engineering of amino acid features that alter phase separation may be a strategy to optimize TF-dependent processes, including cellular reprogramming.

Cell-specific transcriptional programmes in metazoans are established by transcription factors (TFs) binding specific DNA elements mostly within transcriptional enhancers[1–3]. However, the principles governing how thousands of enhancers and hundreds of TFs active in any cell type interact to produce cell-specific transcriptional programmes are largely unknown[3–5]. One major challenge is that virtually all genome-scale studies focus on characterizing sequences in enhancers and transcriptional regulators that have strong transcriptional activity measured in gene reporter systems[6–13]. However, emerging evidence suggests that critical developmental information is encoded in enhancers that drive weak tissue-specific expression patterns[14–16]. Such weak enhancers contain suboptimal TF-binding motifs and spacing, and mutant enhancers with optimized motifs drive elevated but less-specific patterns of transcription, leading to developmental defects[14–17]. These results suggest an important evolutionary trade-off between activity and specificity encoded within weak enhancers, also referred to as 'suboptimization'[14]. Whether such a trade-off is encoded in TFs themselves is unclear. If so, understanding the sequence features that encode such a trade-off could enable the design of natural TF variants with customized cellular reprogramming and other functionalities.

[1]Department of Genome Regulation, Max Planck Institute for Molecular Genetics, Berlin, Germany. [2]Institute of Chemistry and Biochemistry, Department of Biology, Chemistry and Pharmacy, Freie Universität Berlin, Berlin, Germany. [3]Department of Computational Molecular Biology, Max Planck Institute for Molecular Genetics, Berlin, Germany. [4]Centre for Genomic Regulation, The Barcelona Institute of Science and Technology, Barcelona, Spain. [5]Microscopy Core Facility, Max Planck Institute for Molecular Genetics, Berlin, Germany. [6]Universitat Pompeu Fabra, Barcelona, Spain. [7]Present address: Josep Carreras Leukaemia Research Institute, Badalona, Spain. [8]These authors contributed equally: Julian Naderi, Alexandre P. Magalhaes. ✉e-mail: hnisz@molgen.mpg.de

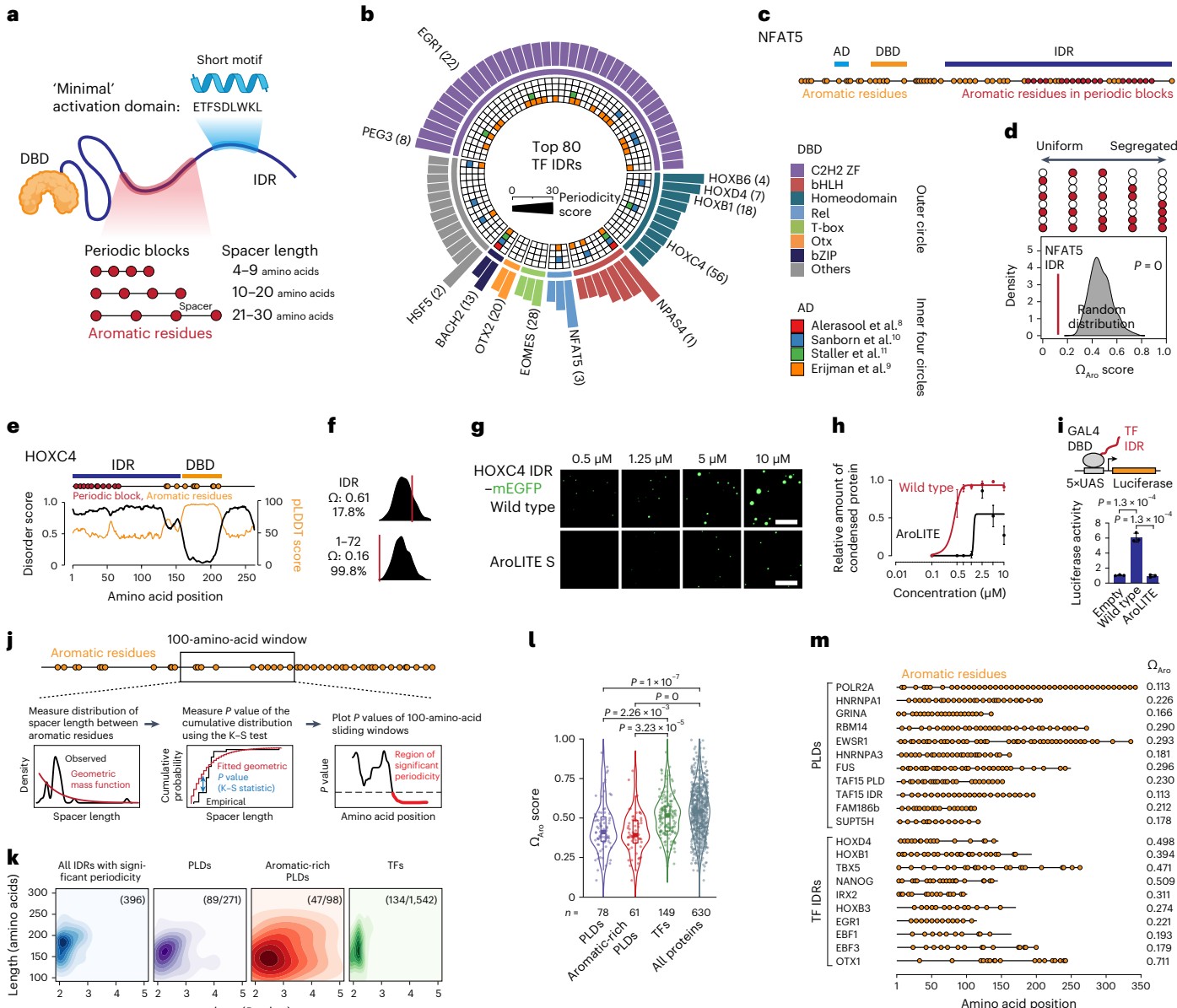

**Fig. 1 | Traces of aromatic periodicity in human TF IDRs. a**, Model of a TF (top) and the method used to identify aromatic periodic blocks (bottom). **b**, The top 80 TFs ranked according to the IDR periodicity score. Ranks are shown in parentheses. The height of the bars in the outer circle is proportional to the periodicity score. The inner circles indicate whether the IDR contains a minimal activation domain (AD) identified in the four studies. **c**, Positioning of aromatic residues in NFAT5. Red dots indicate the position of aromatic residues in periodic block; yellow dots indicate the position of all other aromatic residues. **d**, Omega plot of the NFAT5 IDR. The empirical *P* value is reported. Red dots indicate aromatic residues, white dots indicate any other residue. **e**, Disorder plot (Metapredict; black) and AlphaFold2 pLDDT score (yellow) for HOXC4. **f**, Omega plots of the HOXC4 IDR (top) and the portion encoding the periodic aromatic block (bottom). The coordinates, $\Omega_{Aro}$ scores and the percentage of randomly generated sequences that have a lower $\Omega_{Aro}$ score than the actual sequence are provided. **g**, Representative images of droplet formation of purified recombinant HOXC4 IDR–mEGFP proteins. Scale bars, 5 μm. **h**, Relative amount of condensed protein in the droplet assays. Data are the mean ± s.d. of *n* = 10 images from two replicates. The curves were generated as nonlinear regressions to a sigmoidal

curve function. **i**, Schematic (top) and results of luciferase reporter assays (bottom). The luciferase values were normalized to an internal *Renilla* control and the values are displayed as percentages of the activity measured using an empty vector. Data are the mean ± s.d. of *n* = 3 biological replicates. *P* values are from two-sided unpaired Student's *t*-tests. **j**, Pipeline for the identification of regions with significant periodicity. **k**, Density plot of protein regions with significant periodicity. The length of the region is plotted against the lowest *P* value from the K–S test within the region. The depth of the colour is proportional to the density of the dots. The numbers of proteins that contain a region with significant periodicity over the total number of proteins in each category are shown. **l**, Omega scores of IDRs in various protein classes. *P* values are from one-way analysis of variance with Tukey's multiple comparisons post test. For the box plots, the centre line shows the median, the bounds of the box correspond to interquartile (25th–75th) percentile, and whiskers extend to Q3 + 1.5× the interquartile range and Q1 − 1.5× the interquartile range; the dots beyond the whiskers show Tukey's fences outliers. **m**, Schematic models of prion-like domains (PLDs) and TF IDRs, and their omega scores.

The investigation of trade-offs in TFs is impeded by current models that TF specificity and activity are encoded in separate protein portions. The activity of mammalian TFs is thought to be mediated by sequence motifs that comprise a 'minimal' activation domain, which

is distinct from the DNA-binding domain (DBD) that determines binding specificity (Fig. 1a). Minimal activation domains are typically short (9–40 amino acids) and tend to assume secondary structure when bound to co-activators[10,11,13]. The minimal activation domains however

are almost invariably embedded within much longer intrinsically disordered regions (IDRs) that do not have a stable secondary structure (Fig. 1a)[6–11]. An emerging view suggests that TF IDRs may contribute to transcriptional activity by engaging in multivalent weak interactions. Such interactions can drive phase separation of TFs in vitro and partitioning of TFs into condensates enriched in co-activators and RNA polymerase II (RNAPII) in cells[18–23]. Whether the ability of TFs to form condensates is important for their in vivo function is debated[18–20,24]. Nevertheless, the deletion of IDRs of yeast TFs was shown to reduce genomic binding[25], suggesting that TF IDRs may contribute to transcriptional activity and also to binding specificity.

In this study we set out to investigate whether human TF IDRs are suboptimized (that is, their activity and specificity are submaximal because they are in a trade-off). To do so, we took inspiration from recent insights into prion-like IDRs of RNA-binding proteins to identify a single sequence feature in human TF IDRs that contributes to both transcriptional activity and binding specificity. Prion-like IDRs of RNA-binding proteins (for example, FUS, HNRNPA1 and TDP-43) encode regularly spaced aromatic residues whose number and periodic arrangement promote phase separation[26,27]. We found that hundreds of TF IDRs encode traces of aromatic periodicity. Optimization of aromatic dispersion enhanced the activity and reduced the specificity of TFs, with consistent changes in in vitro phase separation.

## Results

### Human TFs encode short periodic blocks of aromatic residues

Prion-like domains of RNA-binding proteins contain periodically arranged aromatic residues that promote phase separation[27] but is not known whether TFs contain periodically arranged aromatic residues. To gain initial insights into the extent of periodicity of aromatic residues in human TFs, we developed a computational pipeline to identify short blocks of periodically arranged aromatic residues with varying spacer lengths in approximately 1,500 human TFs that had been previously curated (Fig. 1a)[1]. We filtered for periodic blocks of at least four aromatic residues that overlap IDRs and identified 531 TF IDRs containing at least one periodic block (Fig. 1b, Extended Data Fig. 1a,b and Supplementary Table 1). Only 60 of the 531 TF IDRs that contained a short periodic block also contained a minimal activation domain annotated from four recent studies[8–11] and they overlapped in only 31 TF IDRs (Fig. 1a–c, Extended Data Fig. 1c and Supplementary Table 1), suggesting that the periodic blocks are distinct from minimal activation domains. Transcription factor IDRs with periodic blocks were enriched for aromatic residues and serines, and were depleted of charged residues (Extended Data Fig. 1d–f), consistent with typical aromatic 'stickers' and serine/glycine-rich 'spacers' in prion-like domains[26–28].

To quantify the extent of periodicity, we generated a 'periodicity score' as a weighted sum of the periodic blocks, and ranked TFs based on the periodicity score of their IDRs (Fig. 1b and Supplementary Table 1). The periodicity score was further validated by calculating a previously described patterning parameter (the omega score, $\Omega_{Aro}$)[27]. The $\Omega_{Aro}$ score measures the extent of mixing of aromatic residues—where high dispersion leads to a low $\Omega_{Aro}$ value—which is then compared with the mean dispersion of 1,000 randomly generated sequences[27]. For example, the 30 aromatic residues in the NFAT5 IDR are more uniformly dispersed than in 1,000/1,000 randomly generated sequences of identical composition ($\Omega_{Aro} = 0.124$, empirical $P = 0$; Fig. 1c,d). These results suggest that approximately 30% of human TF IDRs contain short blocks of periodically arranged aromatic residues and some of the observed periodicity seems to be non-random.

Three TF IDRs that encode periodic aromatic blocks were selected for functional testing (HOXB1, HOXD4 and HOXC4). All three purified recombinant monomeric enhanced green fluorescent protein (mEGFP)-tagged IDRs formed droplets in a concentration-dependent manner in the presence of a crowding agent (10% polyethylene glycol

8000 (PEG 8000)); Fig. 1e–h and Extended Data Fig. 2a–d). The droplets underwent fusion and wetted the surface of the microscopy slide (Supplementary Videos 1–6), which are hallmarks of liquid–liquid phase separation[29]. Substitution of aromatic residues (AroLITE) reduced droplet formation (Fig. 1g,h and Extended Data Fig. 2c,d,f–h). As a test of transcriptional activity, the wild-type IDRs fused to the GAL4 DBD activated transcription of a luciferase reporter driven by five repeats of the upstream activation sequence (5×UAS) when transfected into various cells ($P < 0.05$, Student's t-test) and substitution of aromatic residues virtually abolished activity of all six IDRs tested (Fig. 1i and Extended Data Fig. 2e,i–m). These findings suggest that aromatic residues are necessary for in vitro phase separation and transactivation capacities of TF IDRs that contain periodic blocks of aromatic residues.

### Submaximal periodicity of aromatic residues in TF IDRs

We noted that many TF IDRs contain short periodic aromatic blocks, but their overall periodicity tends to be limited (Fig. 1e,f). Thus, we hypothesized that aromatic dispersion of TF IDRs might be lower than the theoretical maximum. To test this idea, we quantified periodicity using several approaches. We developed a method to identify protein regions with significant periodicity, independent of sequence length and composition. The spacer length between adjacent aromatic residues was calculated for each protein and the observed distribution of spacer lengths within a sequence was compared with the expected geometric distribution using the Kolmogorov–Smirnov (K–S) test (Fig. 1j). The mean of the geometric distribution was extrapolated from the proportion of aromatic residues, implicitly modelling their occurrence by a Poisson process. The method was applied to 100-amino-acid-long regions using a sliding window approach and the $P$ value of the K–S test was plotted against the position of each window in every protein in the human proteome. The $P$ value and length of the regions encompassing 100 residue windows below the $P$ value threshold were used to define regions with significant periodicity (Fig. 1j). Of note, our approach captured the previously described periodic region in HNRNPA1 (Extended Data Fig. 3a)[27].

Regions with significant periodicity were identified in 2,202 human proteins and 396/2,202 of the periodic regions overlapped IDRs annotated by Metapredict (Extended Data Fig. 3b,c and Supplementary Table 2). The proteins containing regions of significant periodicity were enriched for prion-like proteins and were not enriched for TFs (Fig. 1k and Extended Data Fig. 3d,e). Only 134/1,542 TFs were found to contain a region of significant periodicity and only 63 of these regions were in the IDR (Fig. 1k). Furthermore, the average $\Omega_{Aro}$ score of IDRs in TFs was significantly higher than that of prion-like domains ($P < 1 \times 10^{-4}$, one-way analysis of variance; Fig. 1l,m). These results demonstrate that TF IDRs have lower periodicity than prion-like domains and suggest that the periodicity of TF IDRs may be submaximal.

### Increasing aromatic dispersion enhances transactivation

If TF IDRs have submaximal aromatic dispersion, one could expect that increasing their aromatic dispersion enhances activity. We tested this idea using the HOXD4 IDR as a proof-of-concept (Fig. 2a). We first substituted seven non-aromatic residues with tyrosines in regions of spacer lengths of >15 amino acids in the IDR, increasing its periodicity (AroPLUS; Fig. 2a). Purified mEGFP-tagged AroPLUS IDR protein formed droplets at a lower concentration ($C_{sat}$) than the wild-type HOXD4 IDR in vitro (Fig. 2b,c) and had a twofold higher activity in the GAL4-DBD transactivation assay ($P = 0.032$, Student's t-test; Fig. 2a), which was specific to adding aromatic residues in positions that increase periodicity (Fig. 2a–c). We also generated a HOXD4 IDR mutant in which aromatic residues in the native sequence were uniformly dispersed (AroPERFECT; Fig. 2a). The AroPERFECT IDR formed liquid-like droplets at a similar $C_{sat}$ to the wild-type IDR in vitro (Fig. 2b,c and Supplementary Videos 1,2,7,8). However, fluorescence recovery after photobleaching (FRAP) analyses revealed an increase in the recovery of fluorescence (Fig. 2d

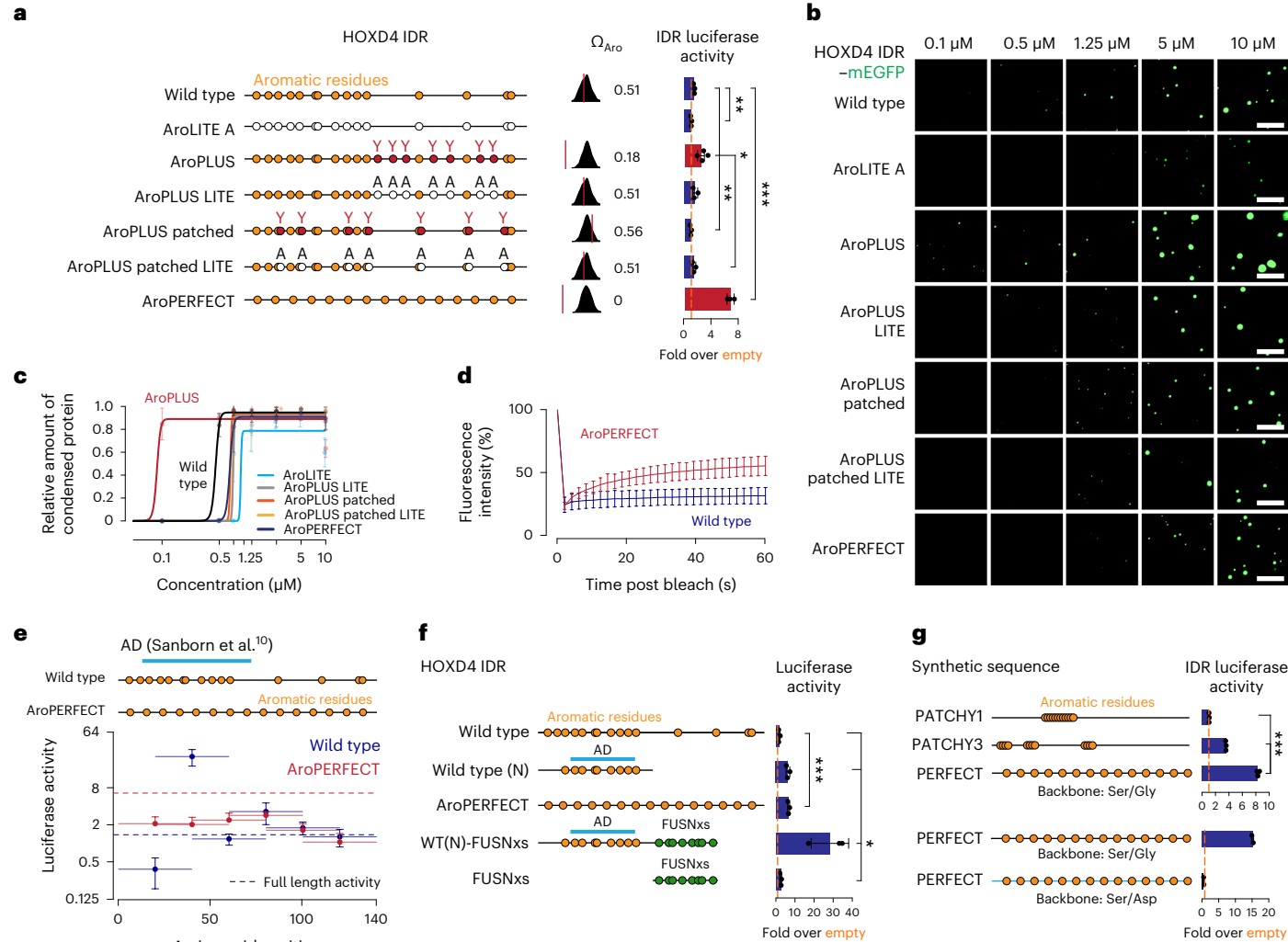

**Fig. 2 | Increasing aromatic dispersion in TF IDRs enhances transactivation.**
**a**, Schematic models of HOXD4 IDRs (left). Aromatic residues (orange dots) and alanine mutations (white dots) are highlighted. Additionally introduced tyrosines are also shown as red dots. Omega plots of the HOXD4 IDRs and $\Omega_{Aro}$ scores (middle). Results of luciferase reporter assays (right). Data are from three biological replicates. **b**, Representative images of droplet formation of purified HOXD4 IDR–mEGFP fusion proteins at the indicated concentrations in droplet formation buffer. Scale bars, 5 μm. **c**, Relative amount of condensed protein per concentration quantified in the droplet formation assays. Data are the mean ± s.d. of $n = 15$ images from three replicates. The curves were generated as nonlinear regressions to a sigmoidal curve function. **d**, Fluorescence intensity of wild-type and AroPERFECT HOXD4 in vitro droplets before, during and after photobleaching. Data are the mean ± s.d. of $n = 20$ images from two replicate imaging experiments. **e**, Results of a HOXD4 IDR tiling experiment using

luciferase reporter assays. Sequences were tiled into fragments of 40 amino acids with 20-amino-acid overlaps. The activities of the full-length IDRs are indicated with dashed horizontal lines. A predicted activation domain (AD) in the HOXD4 wild-type IDR is highlighted (light blue bar). Luciferase activity is reported as the fold change relative to cells transfected with empty vector. **f**, Results of luciferase reporter assays of the indicated HOXD4 IDR constructs. The position of the 40-mer tile containing the AD in **e** is illustrated. Data are from three biological replicates. **g**, Schematic models of synthetic sequences (left); tyrosine residues are highlighted (orange dots). Results of luciferase reporter assays (right). Data are from two (bottom) or three (top) biological replicates. **a,e–g**, Luciferase values were normalized to an internal *Renilla* control and the values are displayed as percentages normalized to the activity measured using an empty vector. Data are the mean ± s.d. *$P < 0.05$, **$P < 0.01$ and ***$P < 1 \times 10^{-3}$; two-sided unpaired Student's *t*-test.

and Extended Data Fig. 4a), suggesting enhanced liquid-like features of IDR droplets. Moreover, the AroPERFECT IDR had a ~five-fold higher activity in the GAL4-DBD transactivation assay compared with the wild-type IDR ($P < 1 \times 10^{-4}$, Student's *t*-test; Fig. 2a and Extended Data Fig. 4b). These results suggest that increased aromatic dispersion in the HOXD4 IDR enhances its activity.

Further mutagenesis of the HOXD4 IDR revealed that increasing the aromatic dispersion enhances transactivation within the confines of additional sequence features but independent of predicted structural elements. The HOXD4 IDR contains a predicted minimal activation domain (Fig. 2e). A 40-amino-acid fragment containing this element, however, had lower activity in the AroPERFECT sequence (Fig. 2e).

Furthermore, the elevated activity of the AroPERFECT IDR could not be explained by the creation of additional minimal activation domains (Fig. 2e) and no correlation with short linear motifs[13] was apparent (Extended Data Fig. 4c,d and Supplementary Table 3). A shift of the uniformly spaced aromatic residues by two positions, but not by one position, towards the amino (N) terminus led to moderately elevated activity (Extended Data Fig. 4c,d), and the degree of enhancement correlated with the number of small inert residues adjacent to aromatic residues (Extended Data Fig. 4e), consistent with previous studies on prion-like sequences[28,30–32]. Finally, we complemented the IDR portion downstream of the minimal activation domain with a short periodic portion of the FUS IDR, which also enhanced activity (WT(N)-FUSNxs; Fig. 2f).

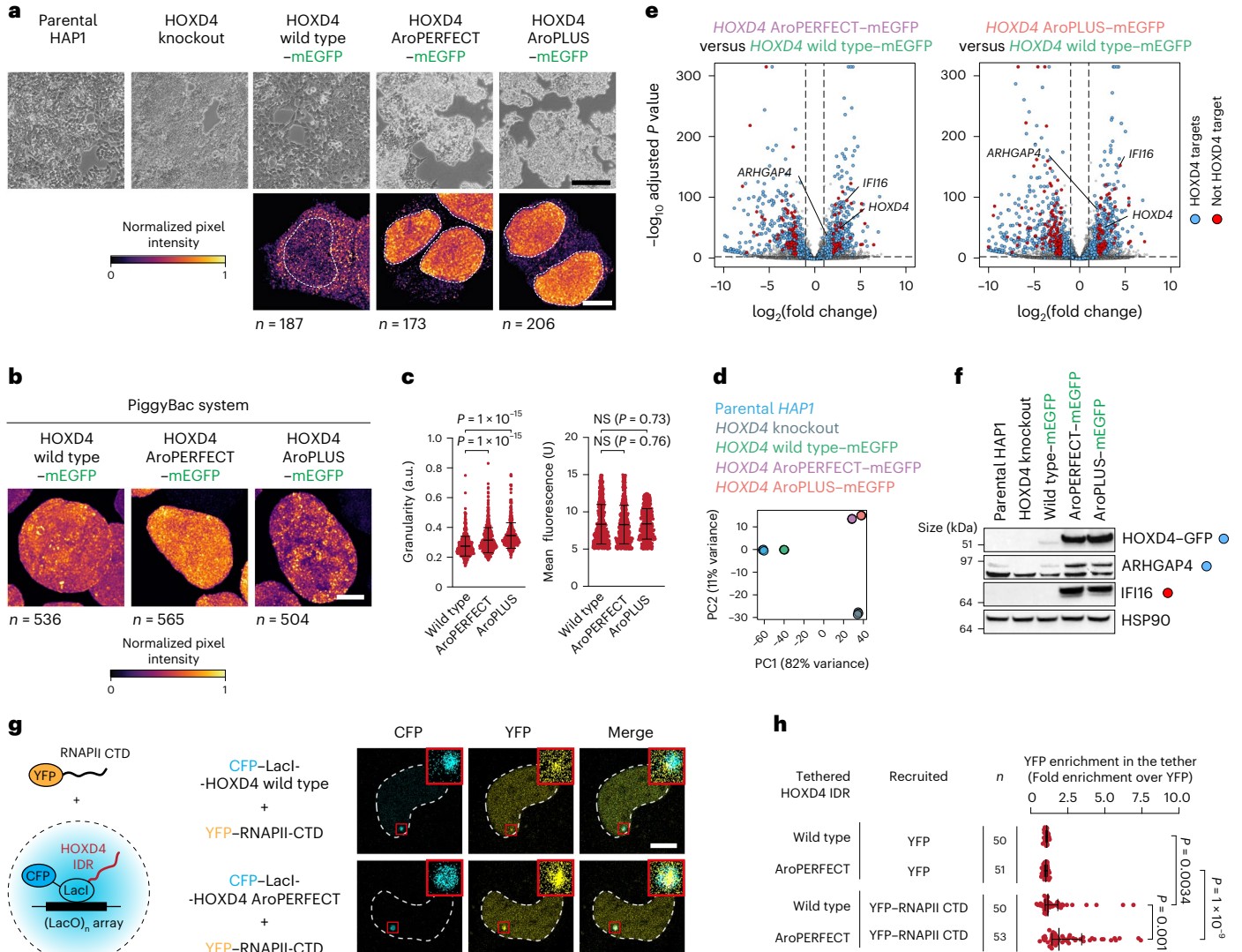

**Fig. 3 | Evidence for gain-of-function of periodic HOXD4 mutants in vivo.**
**a**, Differential interference contrast (DIC) microscopy of the indicated cell lines (top). Representative fluorescence microscopy images of cell nuclei (bottom). The fusion proteins were visualized using anti-GFP immunofluorescence in fixed cells. Dashed white lines represent the nuclear contour. Scale bars, 0.4 mm (DIC microscopy) and 10 μm (fluorescence microscopy). **b**, Representative images of HAP1 HOXD4 wild type–mEGFP, HOXD4 AroPERFECT–mEGFP and HOXD4 AroPLUS–mEGFP nuclei after 24 h of HOXD4 expression. The fusion proteins were visualized using mEGFP fluorescence in fixed cells. The number of individual nuclei per condition is provided. Scale bar, 5 μm. **a,b**, The normalized signal intensity was calculated by dividing the s.d. of the mEGFP signal of each nucleus by the corresponding mean mEGFP signal. **c**, Granularity scores of nuclei with the corresponding mean nuclear mEGFP intensities. Data are the mean ± s.d. of n = 536 (wild-type), 565 (AroPERFECT) and 504 (AroPLUS) nuclei pooled from two independent replicates. a.u., arbitrary units. **d**, Principal component (PC) analysis of the RNA-seq expression profiles of parental HAP1,

*HOXD4*-knockout and the indicated knock-in HAP1 cell lines. **e**, Differential expression analysis of HOXD4 AroPERFECT–mEGFP and HOXD4 AroPLUS–mEGFP versus HOXD4 wild type–mEGFP HAP1 cells. P values were determined using the Benjamini–Hochberg method. **f**, Western blot analysis of HOXD4–mEGFP, IFI16 and ARHGAP4 in the indicated cell lines. HOXD4–mEGFP proteins were probed with anti-GFP. HSP90 was used as the loading control. HOXD4 targets (blue dot) and non-HOXD4 targets (red dot) are highlighted. **g**, Schematic model of the condensate tethering system (left). Fluorescence images of ectopically expressed YFP–RNAPII CTD in live U2OS cells cotransfected with the indicated cyan fluorescent protein (CFP)–LacI-HOXD4 IDR fusion constructs (right). The dashed line represents the nuclear contour. Inserts: magnified views of the regions in the red boxes. Scale bars, 10 μm (main images) and 40 μm (inserts). **h**, Relative YFP signal intensity in the tether foci. Data are the mean ± s.d. of n = 50 (wild-type YFP and wild-type YFP–RNAPII CTD), 51 (AroPERFECT YFP) and 53 (AroPERFECT YFP–RNAPII CTD) nuclei pooled from two independent replicates. **c,h**, P values are from two-sided unpaired Student's t-tests; NS, not significant.

Increased aromatic dispersion enhanced the transcriptional activity of multiple other TF IDRs (HOXC4, OCT4, PDX1 and FOXA3; Extended Data Figs. 4f–k and 5a–c), whereas reducing aromatic dispersion of the periodic EGR1 IDR reduced activity (Extended Data Fig. 5e,f). The spacer residues seemed to constrain the effect of aromatic dispersion, as increased aromatic dispersion of the HOXB1 IDR did not enhance its already strong activity (Extended Data Fig. 5g). Supporting this model, aromatic dispersion in a synthetic neutral IDR backbone correlated with activity, but in a negatively charged backbone it did not

(Fig. 2g). These results suggest that optimizing aromatic dispersion can enhance the activity of TFs but not without limitations that require further investigation.

## Evidence for gain-of-function of periodic HOXD4 mutants

To investigate the impact of the periodic HOXD4 mutants in vivo, we generated HAP1 cell lines in which monomeric enhanced GFP (mEGFP)-tagged full-length HOXD4 variants were knocked-in into the endogenous locus (Extended Data Fig. 6a–d). Surprisingly, knock-in of

the AroPERFECT and AroPLUS HOXD4 mutants altered the morphology of the colonies, suggesting a gain-of-function effect (Fig. 3a and Supplementary Fig. 1a). The wild-type HOXD4−mEGFP protein was modestly enriched in the nucleus, whereas AroPERFECT and AroPLUS HOXD4 were expressed at higher levels and formed intense nuclear clusters (Fig. 3a and Supplementary Fig. 1a). To probe nuclear HOXD4 clusters in cells that express the three variants at comparable levels, we integrated doxycycline (DOX)-inducible mEGFP-tagged alleles using a PiggyBac transposon. The average granularity (that is, normalized s.d. of the fluorescence signal) in cells expressing AroPERFECT and especially AroPLUS HOXD4 transgenes was higher compared with cells expressing wild-type HOXD4 (Fig. 3b,c and Supplementary Fig. 1b). These results suggest that increased aromatic periodicity in the HOXD4 IDR has a gain-of-function effect in vivo.

To gain insights into the genes that are deregulated by the periodic HOXD4 mutants, we performed RNA sequencing (RNA-seq) on the HAP1 cell lines that encode integrated HOXD4 variants at the endogenous locus. Principal component analysis of approximately 16,000 quantified transcripts revealed that the expression profile of AroPERFECT and AroPLUS HOXD4-expressing cells were distinct from that of the wild-type and *HOXD4*-knockout cells (Fig. 3d). We annotated 1,133 HOXD4 target genes based on differential expression between the parental and *HOXD4*-knockout cells. In the AroPERFECT and AroPLUS cells, 76% of the HOXD4 target genes were deregulated in the same direction as in knockout cells, consistent with loss of heterodimerization with PBX factors[33] (Fig. 3e, Extended Data Fig. 6e and Supplementary Table 4). However, we identified 396 genes that were upregulated in the AroPERFECT- and AroPLUS-expressing cells but downregulated in the knockout cells. One of the genes was *HOXD4* itself, consistent with previous studies showing that HOXD4 autoregulates its own gene[34–36]. The elevated levels of HOXD4 and ARHGAP4 were validated with western blots (Fig. 3f). We also identified 43 genes that were upregulated in the AroPERFECT-expressing cells and 64 genes that were upregulated in the AroPLUS-expressing cells, which were not HOXD4 targets—for example, *IFI16* (Fig. 3f and Extended Data Fig. 6e,f). Morphology and expression phenotypes were confirmed in PiggyBac cells expressing similar levels of wild-type and periodic *HOXD4* transgenes (Extended Data Fig. 6f–i and Supplementary Fig. 1c). These results indicate that increased aromatic dispersion in the HOXD4 IDR is associated with enhanced activity and altered gene specificity, which seems to be partly gain-of-function.

To further probe the link between aromatic dispersion, transcriptional activity and condensates, we measured RNAPII CTD recruitment into HOXD4 IDR condensates using a cell-based condensate system[37]. Wild-type or AroPERFECT HOXD4 IDRs were tethered to a LacO array in U2OS cells expressing an ectopic RNAPII CTD−yellow fluorescent protein (YFP) fusion protein (Fig. 3g). RNAPII CTD was mildly enriched in the tethered HOXD4 wild-type IDR condensates and its enrichment was significantly higher in the AroPERFECT IDR condensates (Fig. 3g,h). These results suggest that the enhanced activity and altered gene specificity of periodic HOXD4 IDR is associated with reduced heterodimerization and enhanced RNAPII interaction.

### Optimizing C/EBPα enhances transactivation

Transcription factors can reprogramme cell identity[4,5]; we therefore tested the impact of optimizing aromatic dispersion of well-known reprogramming TFs.

C/EBPα is a master regulator of myeloid cell differentiation[38] (Fig. 4a). Purified recombinant mEGFP-tagged C/EBPα IDRs formed in vitro droplets with liquid-like features (Fig. 4b and Supplementary Videos 9–14) and had transactivation capacity in the GAL4-DBD luciferase system (Fig. 4a). IDR droplet formation and transactivation was dependent on the presence of aromatic residues (Fig. 4a,b and Extended Data Fig. 7a). To test the impact of increased aromatic dispersion, we generated an IDR in which the aromatic residues were

dispersed with perfectly uniform spacing (AroPERFECT IS15). Increased dispersion did not affect the $C_{sat}$ for droplet formation (Fig. 4a,b and Extended Data Fig. 7a) but enhanced recovery after photobleaching in droplets (Fig. 4c) and enhanced transactivation twofold in the GAL4-DBD luciferase system compared with the wild-type IDR ($P < 1 \times 10^{-4}$, Student's *t*-test; Fig. 4a and Extended Data Fig. 7b). Moreover, RNAPII CTD was more enriched in AroPERFECT IS15 condensates compared with wild-type IDR condensates tethered onto the LacO array (Fig. 4d,e). In vitro, an increase in both the number of aromatic residues and their dispersion (AroPERFECT IS10) resulted in a decrease in FRAP (Fig. 4c) as well as decreased transactivation in the GAL4-DBD luciferase system compared with the wild-type IDR ($P < 1 \times 10^{-3}$, Student's *t*-test; Fig. 4a). These results suggest that increased aromatic dispersion enhances transactivation of the C/EBPα IDR but the increase in aromaticity inhibits it.

Further mutagenesis of the C/EBPα IDR revealed that increased aromatic dispersion enhances transactivation within the confines of additional sequence features. The C/EBPα IDR encodes a minimal activation domain[39]. The activity of this element was lower in the AroPERFECT IS15 IDR and the elevated activity of the AroPERFECT IS15 IDR was not caused by the creation of additional minimal activation domains (Fig. 4f). Second, when we increased the aromatic dispersion only in the portion of the C/EBPα IDR downstream of the activation domain (WT(N)-IS15), the activity of the IDR was elevated threefold compared with the wild type and twofold compared with the N-terminal portion (Fig. 4g). Third, replacement of the downstream IDR portion with portions of the periodic FUSN-IDR (WT(N)-FUSN and WT(N)-FUSNxs) enhanced activity over the wild-type C/EBPα IDR (Fig. 4g). Fourth, a shift of the aromatic pattern of AroPERFECT IS15 IDR by one amino acid towards the carboxy (C) terminus resulted in higher transactivation compared with the wild type, whereas a shift by two positions did not (Extended Data Fig. 7c), and the magnitude of change correlated with the proportion of small inert residues adjacent to the aromatic residues (Extended Data Fig. 4e). Aromatic dispersion therefore enhances transactivation independent of the known C/EBPα activation domain and within the confines of the spacer residues.

### Optimizing C/EBPα enhances macrophage reprogramming

We next measured the cellular reprogramming capacity of stably transduced C/EBPα variants in a leukaemic human B cell line (RCH-rtTA cells). In this system, induction of C/EBPα by DOX reprogrammes B cells into terminally differentiated macrophages while arresting the cell cycle[40,41]. Cell conversion was monitored through fluorescence-activated-cell-sorting (FACS) analysis of the B cell marker CD19 and the macrophage marker Mac1 (also known as CD11b; encoded by the gene *ITGAM*; Fig. 5a,b and Extended Data Fig. 7d)[40,41]. As expected, C/EBPα expression led to a gradual increase in the proportion of Mac1⁺CD19⁻ macrophages among the GFP⁺ cell population over seven days (Fig. 5c and Extended Data Fig. 7d,e). Expression of the AroPERFECT IS15 C/EBPα mutant increased both the speed of appearance and proportion of Mac1⁺ cells among the GFP⁺ population (Fig. 5c and Extended Data Fig. 7d,e).

To gain insights into the transcriptional programmes driven by the C/EBPα proteins, we performed single-cell RNA-seq (scRNA-seq) of cultures expressing wild-type and AroPERFECT IS15 C/EBPα variants after seven days of transgene induction. The culture expressing the transcriptionally inert AroPERFECT IS10 C/EBPα variant was included as a negative control. Cross-referencing the clusters on the combined scRNA cell-state map of the three cultures with marker genes of known cell populations identified terminally differentiated macrophages, macrophage precursors and various B cell subpopulations in our data (Fig. 5d and Extended Data Fig. 8a–g). Consistent with the FACS analysis, the proportion of late macrophages was higher among the GFP⁺ cells in the AroPERFECT IS15-transduced population (Fig. 5e), indicating enhanced reprogramming capacity. A comparative analysis

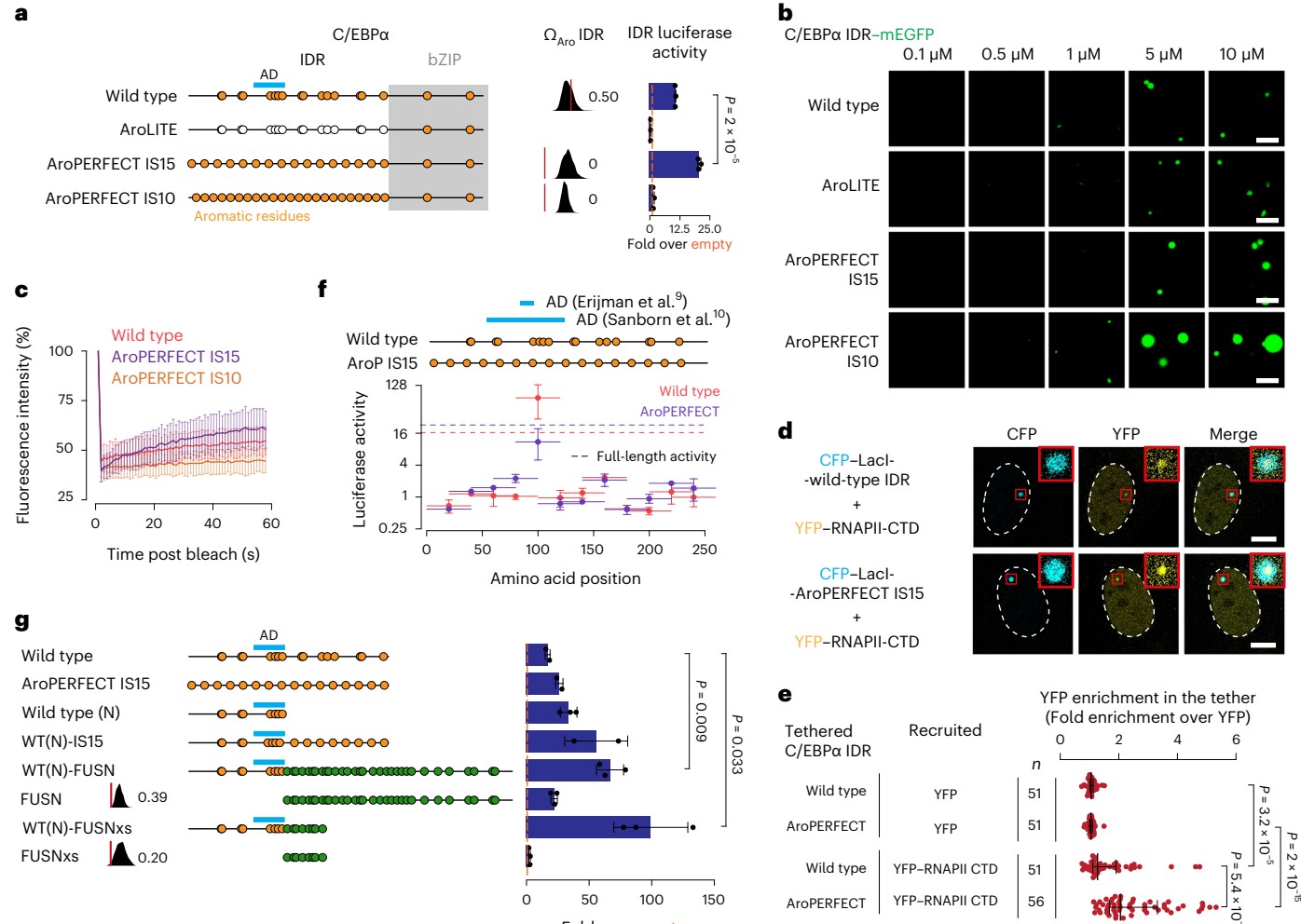

**Fig. 4 | Optimizing aromatic dispersion in C/EBPα enhances transactivation.**
**a**, Schematic models of wild-type and mutant C/EBPα proteins (left). The positions of the bZIP DBD (grey box) and aromatic residues (orange dots) are indicated. Omega plots and $\Omega_{Aro}$ scores (middle). Results of luciferase reporter assays (right). Data are the mean ± s.d. of $n = 3$ biological replicates with three technical replicates each. **b**, Representative images of droplet formation of purified C/EBPα IDR–mEGFP fusion proteins at the indicated concentrations in droplet formation buffer. Scale bars, 5 μm. **c**, Fluorescence intensity of C/EBPα wild type, AroLITE and AroPERFECT IS15 IDR in in vitro droplets before, during and after photobleaching. Data are the mean ± s.d. of $n = 15$ (wild-type) and 14 (AroPERFECT IS15 and AroPERFECT IS10) droplets from two replicates. **d**, Fluorescence images of ectopically expressed YFP–RNAPII CTD in live U2OS cells that were cotransfected with the indicated CFP–LacI-C/EBPα IDR fusion constructs. The dashed line represents the nuclear contour. Inserts: magnified views of the regions in the red boxes. Scale bars, 10 μm (main images) and 40 μm (inserts). **e**, Relative YFP signal intensity in the tether foci. Data are the mean ± s.d. of $n = 51$ (wild-type YFP, AroPERFECT YFP and wild-type YFP–RNAPII CTD) and 56 (AroPERFECT YFP–RNAPII CTD) nuclei pooled from two independent replicates. **f**, Results of a C/EBPα IDR tiling experiment using luciferase reporter assays. C/EBPα wild type and AroPERFECT IS15 IDR sequences were tiled into fragments of 40 amino acids with 20-amino-acid overlaps. The activities of the full-length IDRs are indicated with dashed horizontal lines. **g**, Results of luciferase reporter assays of the indicated IDR constructs. **a**,**f**,**g**, Luciferase values were normalized to an internal *Renilla* control and the values are displayed as percentages normalized to the activity measured using an empty vector. **f**,**g**, Data are the mean ± s.d. of $n = 3$ biological replicates. **a**,**e**,**g**, $P$ values are from a two-sided unpaired Student's $t$-tests.

---

of the transcriptomes of late macrophages expressing wild-type or AroPERFECT IS15 C/EBPα revealed largely similar expression profiles; however, the AroPERFECT IS15 macrophages expressed a small set of 31 genes that were not detected in the wild-type C/EBPα-expressing macrophages (Extended Data Fig. 8h,i and Supplementary Table 5), suggesting slightly altered gene specificity.

**Optimizing C/EBPα leads to stronger genomic binding**
To dissect the molecular basis of enhanced reprogramming we performed chromatin immunoprecipitation with sequencing (ChIP–Seq) of C/EBPα–GFP proteins, using an anti-GFP antibody, after 24 and 48 h of transgene induction in isolated clonal cell lines (Extended Data Fig. 8j). The majority of sites bound by wild-type C/EBPα were also bound by AroPERFECT IS15 C/EBPα, but the read densities at the

bound sites were consistently higher in the AroPERFECT IS15 samples (Fig. 5f, Extended Data Fig. 8k and Supplementary Fig. 2a,b). Overall, approximately 100× more differentially bound peaks had higher read densities in AroPERFECT IS15 than the other way around (Fig. 5f and Supplementary Fig. 2a).

Differential genomic binding of AroPERFECT IS15 C/EBPα was associated with differences in motif composition at the binding sites. For these analyses, we used approximately 28,000 ChIP–Seq peaks that were identified as 'shared' by both wild-type and AroPERFECT IS15 C/EBPα, and approximately 60,000 ChIP–Seq peaks that were uniquely bound by AroPERFECT IS15 C/EBPα at least at one time point (Fig. 5f). Cross-referencing the peaks with published C/EBPα ChIP–Seq datasets revealed that approximately 50,000 of the sites were previously reported as binding sites of wild-type C/EBPα ('peaks unique to

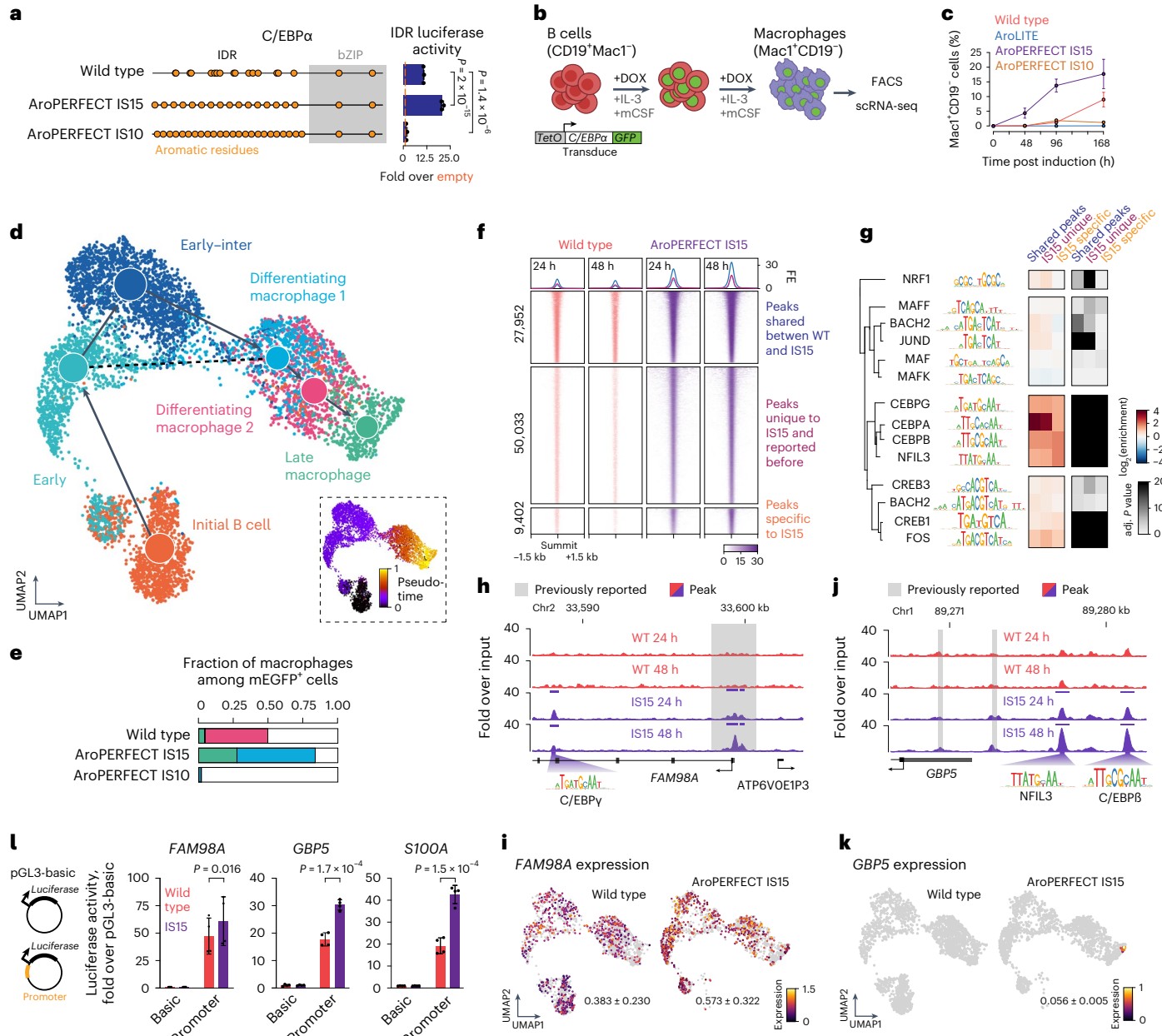

**Fig. 5 | Optimizing aromatic dispersion in C/EBPα enhances macrophage reprogramming, and leads to stronger and more promiscuous genomic binding. a**, Schematic models of wild-type and mutant C/EBPα proteins. The transactivation data are identical to the data displayed in Fig. 4a. *P* values are from two-sided unpaired Student's *t*-tests. **b**, Schematic model of C/EBPα-mediated transdifferentiation of B cells to macrophages. **c**, FACS quantification of GFP⁺ RCH-rtTA cells encoding C/EBPα overexpression cassettes. The proportions of CD19⁻ Mac1⁺ cells were measured 48, 96 and 168 h after transgene induction. Data are the mean ± s.d. of *n* = 5 (wild type and AroPERFECT IS15) and 3 (AroLITE and AroPERFECT IS10) independent experiments. **d**, Graph-based clustering (uniform manifold approximation and projection, UMAP) of the scRNA-seq data of C/EBPα-mediated transdifferentiation. Clusters were annotated based on marker genes. Overlayed is the partition-based graph abstraction (PAGA) showing the cell trajectory based on dynamic modelling of RNA velocity. Inset: pseudotime plot. **e**, Proportion of mEGFP⁺ cells in the macrophage clusters (colour-coded as in **d**). **f**, Heatmap representation of ChIP–Seq read densities of wild-type and AroPERFECT IS15 C/EBPα within a

1.5-kb window around all shared C/EBPα peaks and differentially enriched peaks in AroPERFECT IS15 C/EBPα. 'Peaks unique to IS15 and reported before' denotes binding sites differentially enriched in IS15 binding that overlap C/EBPα peaks reported in previous literature. FE, fold enrichment. **g**, Enrichment scores of bZIP TF motifs and adjusted (adj.) *P* values of enrichment at the three indicated peak sets. *P* values were determined using the Benjamini–Hochberg method. **h,j**, AroPERFECT IS15 C/EBPα shows enhanced binding at the *FAM98A* (**h**) and *GBP5* (**j**) loci. Displayed are genome browser tracks of ChIP–Seq data of C/EBPα 24 and 48 h after C/EBPα induction. The coordinates are hg38 genome assembly coordinates. **i,k**, UMAPs coloured on *FAM98A* (**i**) and *GBP5* (**k**) expression. The numbers denote the mean ± s.d. expression in the whole samples. **l**, Luciferase assays using the indicated reporter plasmids cotransfected with expression vectors encoding either wild-type or AroPERFECT IS15 C/EBPα. Luciferase values were normalized to an internal *Renilla* control and the values are displayed as percentages of the activity measured using the 'basic' vector. Data are the mean ± s.d. of four biological replicates. *P* values are from two-sided unpaired Student's *t*-tests.

IS15, reported before' in Fig. 5f) and about 10,000 were specific to our AroPERFECT IS15 C/EBPα data ('peaks specific to IS15' in Fig. 5f). The shared binding peaks and peaks unique to IS15 reported previously were highly enriched for the same canonical C/EBPα motif but the peaks specific to IS15 were less enriched for the C/EBPα motif and more enriched for other basic-leucine zipper (bZIP) TF motifs, including C/EBPβ and NFIL3 (Fig. 5g).

The impact of differential binding on gene expression was confirmed using multiple approaches. IS15-specific binding at several loci was associated with detectable IS15-specific expression of the gene in the scRNA-seq data of B cell and macrophage clusters (Fig. 5h–k and Supplementary Fig. 2c–f). Furthermore, cloning of IS15-specific peaks in a luciferase reporter revealed elevated activity when cotransfected with an AroPERFECT IS15 C/EBPα vector compared with the wild type (Fig. 5l). Finally, differential expression was confirmed with FACS analysis of the products of two macrophage-restricted genes: CD66 (the product of the *CEACAM* genes; Extended Data Fig. 8l–n) and FCGR2A (Extended Data Fig. 8o–q). Together, these results suggest that the enhanced reprogramming capacity of AroPERFECT IS15 C/EBPα is associated with stronger and more promiscuous genomic binding.

### Optimizing NGN2 enhances neural differentiation

As a second proof-of-concept, we tested the impact of optimizing aromatic dispersion on the reprogramming capacity of the neurogenic TF neurogenin-2 (NGN2; ref. 42; Fig. 6a).

Wild-type recombinant, mEGFP-tagged NGN2 C-terminal IDR (C-IDR) formed liquid-like droplets in a concentration-dependent manner, dependent on the presence of aromatic residues (Extended Data Fig. 9a–c and Supplementary Videos 15,16). Similar to results with the IDRs of C/EBPα, HOXD4 and HOXC4, a mutant NGN2 C-IDR in which the five aromatic residues uniformly dispersed (AroPERFECT C-IDR) formed droplets similar to the wild-type IDR in vitro and had a small statistically non-significant difference in FRAP (Fig. 6b). None of the IDRs had measurable activity in the GAL4-DBD luciferase system (Extended Data Fig. 9a), consistent with a report that a minimal activation domain is located within the NGN2 DBD[43].

To assay the reprogramming capacity of NGN2 mutants, DOX-inducible FLAG-tagged NGN2 transgenes were stably integrated in ZIP13K2 human induced pluripotent stem cells (iPSCs) using a PiggyBac transposon (Fig. 6c and Extended Data Fig. 9d,e). The transposon also encoded mEGFP separated by a T2A sequence. Following 24 h of DOX induction, mEGFP+ cells were FACS-sorted and replated at a defined density. After 48 h, the medium was exchanged with medium supporting neural differentiation and the cells were eventually characterized by staining nuclei and tubulin (Fig. 6c). Twice as many sorted cells expressing the AroPERFECT NGN2 mutant survived and half as many cells expressing the AroLITE NGN2 mutant survived compared with the wild-type NGN2-expressing cells after five days of transgene induction ($P < 0.05$, Student's $t$-test; Fig. 6d,e). Consistent with these data, the density of cell projections was significantly higher in the AroPERFECT NGN2-expressing cultures compared with cultures of cells expressing wild-type NGN2 after five days of transgene induction ($P < 0.05$, Student's $t$-test; Fig. 6d,f and Supplementary Fig. 3a–c). These results indicate that the increased aromatic dispersion in the C-terminal IDR of NGN2 enhances its capacity to reprogramme iPSCs into neuron-like cells.

To investigate the molecular basis of enhanced reprogramming by the AroPERFECT NGN2 mutant, we performed RNA-seq after five days as well as NGN2 ChIP–Seq 24 and 48 h after transgene induction. The global RNA-seq profiles of cultures expressing wild-type, AroLITE and AroPERFECT NGN2 proteins were largely similar and included NGN2 target genes annotated based on previous studies (Fig. 6g,h and Extended Data Fig. 9f,g), consistent with media conditions promoting the survival of neurons but not iPSCs after the media switch on day 2 (Fig. 6b). The ChIP–Seq data revealed that most sites bound

by wild-type NGN2 were also bound by the AroLITE and AroPERFECT protein (Fig. 6i and Extended Data Fig. 9h) but the read densities at the binding sites were consistently lower in the AroLITE-expressing cells and moderately higher in AroPERFECT-expressing cells at 24 h (Fig. 6i,j and Extended Data Fig. 9i). The basic helix–loop–helix (bHLH) TF motif composition of the binding peaks was largely similar (Extended Data Fig. 9j). Consistent with these results, measurements of genome-wide nascent transcription after short-term NGN2 induction revealed elevated transcription of NGN2 target genes in AroPERFECT-expressing cells (Fig. 6k, Extended Data Fig. 9k,l and Supplementary Fig. 4). These results suggest that optimizing the aromatic dispersion in the NGN2 C-terminal IDR enhances neural reprogramming and slightly alters genomic binding.

### Optimizing MYOD1 enhances myotube differentiation

Finally, we tested the impact of optimizing aromatic dispersion on the function of the myogenic TF MYOD1 (ref. 44; Fig. 7a). Both the N-terminal and C-terminal MYOD1 IDRs had transactivation capacity in the GAL4-DBD luciferase system in myoblasts (Fig. 7a). Increased aromatic dispersion of aromatic residues abolished transactivation of the N-terminal IDR that contains a minimal activation domain but increased transactivation of the C-terminal IDR (Fig. 7a and Extended Data Fig. 10a), and the enhanced activity of the AroPERFECT C-IDR was not caused by the creation of minimal activation domains (Extended Data Fig. 10b).

To assay the reprogramming capacity of MYOD1 mutants, DOX-inducible *MYOD1* transgenes were stably integrated into C2C12 murine myoblasts using a PiggyBac transposon (Fig. 7b). The transposon also encoded mEGFP separated by a T2A sequence from MYOD1 (Fig. 7b). In this system, forced expression of MYOD1 differentiates myoblasts into multinucleated myotubes within a few days[45]. Cell fusion was quantified as the percentage of 4,6-diamidino-2-phenylindole (DAPI)-stained nuclei in multinucleated cells visualized using the mEGFP fluorescence signal as the cytoplasmic marker[46]. Approximately 50% of nuclei expressing wild-type MYOD1 were found in fused cells after three days of transgene induction (Fig. 7c,d and Extended Data Fig. 10c). Mutation of the aromatic residues into alanines in both IDRs (AroLITE) prevented fusion, whereas mutation of the aromatic residues in the C-terminal IDR (AroLITE C) had a negligible effect (Fig. 7c,d). Expression of the MYOD1 mutant with enhanced periodicity in its C-terminal IDR (AroPERFECT C) led to a significant increase in fusion after three days ($P < 0.05$, Student's $t$-test; Fig. 7c,d). These results suggest that increased periodicity of aromatic residues in the C-terminal IDR of MYOD1 enhances myotube differentiation.

RNA-sequencing analysis of differentiating cells expressing various MYOD1 proteins revealed signatures consistent with observed morphological differences. Principal component analysis of the RNA-Seq data demonstrated that the global expression profiles of AroLITE-expressing cells were similar to that of the parental myoblasts (Fig. 7e and Extended Data Fig. 10d). The expression profile of AroPERFECT C-expressing cells was largely similar to cells expressing wild-type MYOD1 but included 290 differentially expressed genes, 197 of which were MYOD1 targets and were enriched for genes implicated in cell adhesion (Fig. 7e,f and Extended Data Fig. 10d–g). These results suggest that morphologies are associated with differences in gene expression profiles of differentiating myotubes expressing various MYOD1 proteins.

### Discussion

The results presented here support a model that human TFs have suboptimal transcriptional activity. We present evidence that suboptimality in several TFs is encoded as submaximal dispersion of aromatic residues in their IDRs. In several cellular reprogramming systems, an increase in aromatic dispersion enhanced the activity and compromised gene specificity of the TFs. Together with previous

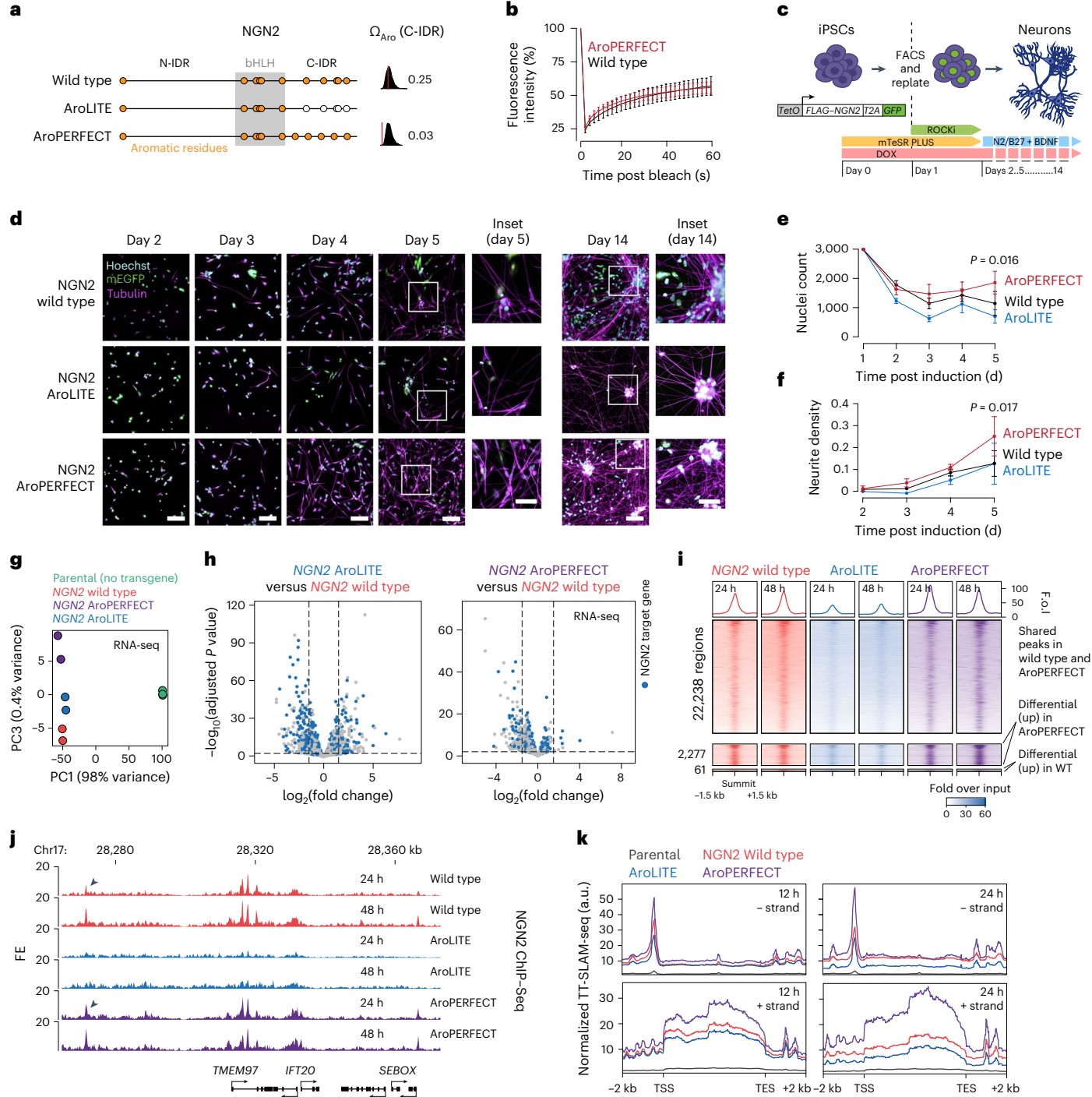

**Fig. 6 | Optimizing aromatic dispersion in NGN2 enhances neural differentiation. a**, Schematic models of wild-type and mutant NGN2 proteins (left). The positions of the bHLH DBD (grey box) and aromatic amino acids (yellow dots) are indicated. Omega plots and $\Omega_{Aro}$ scores (right). **b**, Fluorescence intensity of NGN2 wild-type and AroPERFECT IDR in in vitro droplets before, during and after photobleaching. Data are the mean ± s.d. of $n = 20$ droplets pooled from two independent replicates. **c**, Schematic model of the NGN2-mediated human iPSC-to-neuron differentiation experiment. ROCKi, Rho-kinase inhibitor. **d**, Representative fluorescence microscopy images of differentiating human iPSCs expressing the indicated NGN2 proteins. Hoechst dye was used as a nuclear counterstain; mEGFP, NGN2-T2A–mEGFP. Insets: magnified views of the regions in the white boxes. Scale bars, 0.1 mm (main images) and 0.05 mm (insets). **e**, Number of cells, based on Hoechst nuclear staining, in the NGN2-directed differentiation experiments. **f**, Neurite density (fraction of tubulin-covered area) in the NGN2-directed differentiation experiments. **e**,**f**, Data are the mean ± s.d. of $n = 6$ images

pooled from two independent experiments. $P$ values from a two-sided unpaired Student's $t$-test. **g**, Principal component analysis of the RNA-seq expression profiles of parental ZIP13K2 human iPSCs and human iPSCs expressing the indicated *NGN2* transgenes. **h**, Differential expression analysis of human iPSCs expressing the indicated transgenes. NGN2 target genes are highlighted. $P$ values were determined using the Benjamini–Hochberg method. **i**, Heatmap representation of ChIP–Seq read densities of cells expressing wild-type, AroLITE and AroPERFECT NGN2 within a 1.5 kb window around all shared NGN2 peaks (top), differentially enriched peaks in AroPERFECT NGN2 (centre) and differentially enriched peaks in wild-type NGN2 (bottom). FE, fold over input. **j**, NGN2 differential binding at the *TMEM97* locus. Genome browser tracks of ChIP–Seq data after 24 and 48 h of NGN2 expression are displayed. The arrowhead highlights a differentially bound peak at 24 h. The coordinates are hg38 genome assembly coordinates. **k**, Nascent transcription (TT-SLAM-Seq) metagene profiles at approximately 9,000 NGN2 target genes. TSS, transcription start site; TES, transcription end site.

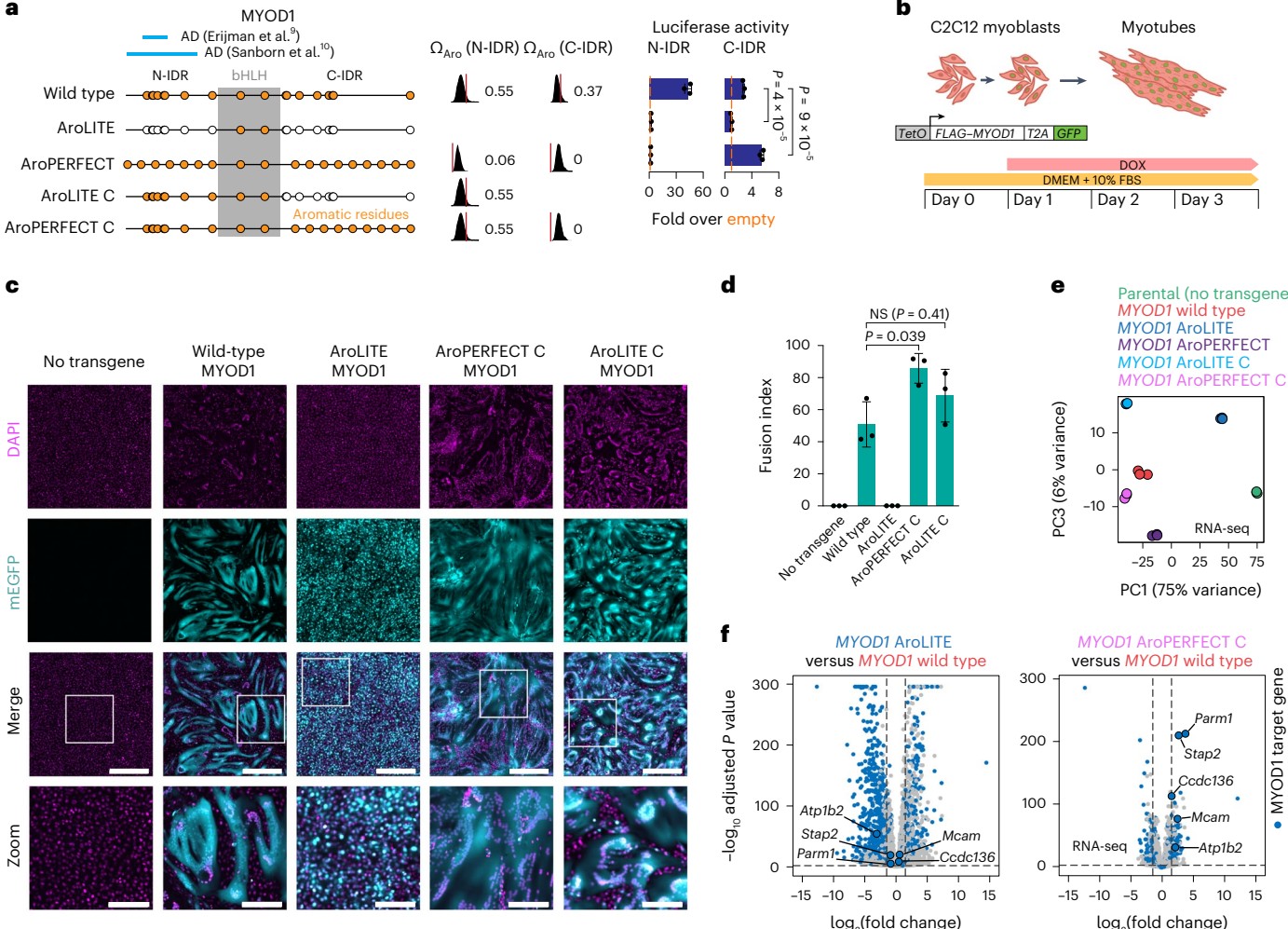

**Fig. 7 | Optimizing aromatic dispersion in MYOD1 enhances myotube differentiation. a**, Schematic models of wild-type and mutant MYOD1 proteins (left). The position of the bHLH DBD (grey box) and aromatic amino acids (orange dots) are indicated. Omega plots and $\Omega_{Aro}$ scores of the N-terminal and C-terminal IDRs (middle). Results of luciferase reporter assays in C2C12 mouse myoblasts (right). Luciferase values were normalized to an internal *Renilla* control and the values are displayed as percentages normalized to the activity measured using an empty vector. Data are the mean ± s.d. of three biological replicates. *P* values are from two-sided unpaired Student's *t*-tests. **b**, Schematic model of the MYOD1-mediated myotube differentiation experiment. **c**, Representative fluorescence microscopy images of differentiating C2C12 myoblasts expressing the indicated MYOD1 proteins on day 3 after DOX induction. The mEGFP signal of the MYOD1-T2A–mEGFP construct was used as a cytoplasmic marker. Nuclear counterstain (DAPI) is shown in magenta. Magnified views of the regions in the white boxes are provided (zoom; bottom). Scale bars, 0.5 mm (main images) and 0.2 mm (zoom). **d**, MYOD1-driven myotube differentiation efficiency. The fusion index was calculated as the percentage of nuclei in fused cells (cells containing at least three nuclei). Data are the mean ± s.d. of *n* = 15 images per genotype pooled from three biological replicates. *P* values are from two-sided unpaired Student's *t*-tests. **e**, Principal component analysis of RNA-seq expression profiles of parental C2C12 cells as well as cells expressing the indicated *MYOD1* transgenes. **f**, Differential expression analysis of C2C12 cells expressing AroLITE or AroPERFECT C MYOD1 versus C2C12 cells expressing wild-type MYOD1. MYOD1 target genes are represented as blue dots. Highlighted genes were differentially expressed and are involved in cell adhesion. *P* values were calculated using the Benjamini–Hochberg method.

work showing that enhancer DNA sequences are suboptimal for TF binding[14,16], the results suggest an important evolutionary trade-off between activity and specificity at multiple levels in eukaryotic transcriptional control.

The results provide insights into how human TFs work. Some TFs encode short linear motifs that can fold into secondary structures and mediate specific interactions with effector proteins[47]. Such sequences are typically identified as minimal activation domains that are sufficient to activate transcription of a reporter gene[7–10,13,48,49]. Our results suggest that some TF IDRs encode periodically arranged aromatic residues that contribute to activity via multivalent interactions with other disordered protein regions. This mode of activity may be distinct from, and complementary to, the transcriptional activity conferred by minimal activation domains. Consistent with this proposal, hydrogels of periodic low-complexity domains can bind

RNAPII CTD that itself is highly periodic[50], and we found that periodic TF IDRs recruit RNAPII CTD more efficiently than wild-type TF IDRs in the cell-based condensate tethering system. This model may help explain why minimal activation domains are typically embedded in large disordered sequences[6–10] and why some TF IDRs can be substituted with the periodic FUS prion-like domain[51,52]. This model predicts that important regulatory information may be encoded in sequences with weak or no activity.

Transcription factor-mediated differentiation and reprogramming are generally stochastic and inefficient, and the inefficiency is thought to be explained by chromatin barriers or lack of TF effector partners[4,5,53–58]. Our results suggest that an additional impediment to directed differentiation and reprogramming may be the suboptimal activity of native TFs, and that reprogramming efficiency may be improved by enhancing a prion-like phase separation 'grammar'

in native TFs. In summary, we propose that altering phase separation capacity may be a universal strategy to optimize any TF-dependent process.

## Online content

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

## Methods

### Ethics statement

The research complied with all relevant ethical regulations and was approved by the Max Planck Institute for Molecular Genetics and the Centre for Genomic Regulation.

### Cell culture

The cell lines HAP1, HEK293T, V6.5 mESCs, ZIP13K2 human iPSCs, Kelly, SH-SY5Y, C2C12 murine myoblasts and U2OS were cultured as per American Type Culture Collection guidelines. RCH-rtTA cells were derived from the RCH-ACV lymphoblastic leukaemia cell line[59]. RCH-rtTA cells and their derivates were cultured in RPMI medium (Gibco) containing 10% fetal bovine serum supplemented with 1% glutamine (Gibco), 1% penicillin–streptomycin (Thermo Fischer Scientific) and 550 μM β-mercaptoethanol (Gibco). Cells were maintained at a density of $0.1$–$6 × 10^6$ cells ml$^{-1}$. The cell lines were checked for mycoplasma contamination and tested negative.

### Genomic DNA extraction

Genomic DNA of cultured cells was extracted using a GeneJET genomic DNA purification kit (Thermo Fischer Scientific) following the manufacturer's instructions.

### Generation of HOXD4–GFP knock-in and knockout lines

For an endogenous knock-in of mEGFP-tagged HOXD4 variants, we cloned a synthesized, codon-optimized sequence for wild-type, AroPERFECT or AroPLUS *HOXD4* (Twist Bioscience) into a pUC19 backbone (Addgene, catalogue number 50005) that was linearized by restriction digest with BamHI (NEB) and HindIII (NEB). Besides the aforementioned HOXD4 coding sequences, the repair template contained N- and C-terminal homology regions for the *HOXD4* genomic locus amplified from HAP1 genomic DNA, a synthesized GS-linker sequence (Sigma) and a mEGFP fluorescent protein sequence amplified from a pET45 plasmid (Extended Data Fig. 6a). All plasmids were cloned via Gibson Assembly using a NEBuilder HiFi DNA assembly kit (NEB).

The endogenous *HOXD4* locus was targeted by two guide RNAs cutting the N- or C-terminus of the HOXD4 coding sequence, respectively (Extended Data Fig. 6a). Both guide RNA sequences (Supplementary Table 6) were cloned into the sgRNA-Cas9 vector px459 (Addgene, catalogue number 62988). Repair template and guide RNA vectors were cotransfected into HAP1 cells using Lipofectamine 3000 transfection reagent (Thermo Fischer Scientific) at a molar ratio of 5:1:1 following the manufacturer's instructions. To screen for functional integration, the transfected cells were sorted for mEGFP expression by flow cytometry after four days and a second time after an additional week. Positive cells were seeded into 96-well plates as single cells. After expansion, the clones were genotyped for correct integration by PCR on extracted genomic DNA (Extended Data Fig. 6c). Positive clones for every HOXD4-expressing line with similar mEGFP expression levels were selected. To generate a *HOXD4*-knockout cell line, HAP1 cells were transfected with both guide RNAs only. After four days, the cells were seeded as single cells by flow cytometry and genotyped for *HOXD4* deletion by PCR on extracted genomic DNA and quantitative real-time PCR on synthesized complementary DNA (Extended Data Fig. 6c,d).

### Generation of cells encoding DOX-inducible transgenes using the PiggyBac system

To generate a DOX-inducible overexpression system of HOXD4, we randomly integrated the coding sequences of wild-type, AroPERFECT and AroPLUS *HOXD4* into HAP1 cells using the PiggyBac transposon system. To generate a DOX-inducible overexpression system of NGN2, we randomly integrated the coding sequences of wild-type, AroLITE and AroPERFECT *NGN2* into ZIP13K2 cells using the PiggyBac transposon system. Similarly, to generate a DOX-inducible overexpression system of MYOD1, we randomly integrated the coding sequences of

wild-type, AroLITE, AroPERFECT C and AroLITE C *MYOD1* into C2C12 cells using the PiggyBac transposon system. The details are described in the Supplementary Information.

### Generation of DOX-inducible C/EBPα overexpression lines in RCH cells

TetO-C/EBPα–mEGFP plasmids were cloned via Gibson assembly using a pHAGE2-tetO backbone. HEK293T cells were cotransfected with vector plasmid and packaging plasmid using calcium phosphate transfection. Viral supernatants were collected 48 h later and concentrated by ultracentrifugation at 20,000$g$ and 20 °C for 2 h. The viral concentrates were resuspended in PBS. RCH cells were transduced by centrifugation with concentrated virus solution for 2 h at 32 °C and 1,000$g$ in culturing medium.

### MYOD1-mediated myogenic differentiation of C2C12 myoblasts

C2C12 myoblasts with an integrated MYOD1 overexpression cassette were seeded on chambered μ-Slide 8 well ibiTreat coverslips (Ibidi). Once 85–90% confluence was reached, 2 μg ml$^{-1}$ DOX was added to the culture medium to induce expression of the *MYOD1* transgene. The differentiation medium was changed every day for three days. For imaging, the cells were washed with PBS and fixed with 4% paraformaldehyde for 15 min at room temperature. The cells were counterstained with DAPI (Fig. 7c and Extended Data Fig. 10c).

### RNA isolation and quantitative real-time PCR

RNA from cultured cells was extracted using a Direct-zol RNA Micro-Prep kit (Zymo Research) following the manufacturer's instructions. Subsequently, 1 μg of extracted RNA was used as input material for cDNA synthesis with the RevertAid first strand cDNA synthesis kit (Thermo Fischer Scientific) using random hexamer primers as per the manufacturer's instructions. The synthesized cDNA was diluted 1:10 with water and stored at −20 °C. Quantitative real-time PCR was performed using 2×PowerUP SYBR green master mix (Applied Biosystems) and the primers listed in Supplementary Table 6.

### KAPA stranded messenger RNA-seq of HAP1 *HOXD4* knock-in cells

Six-well plates were seeded with HAP1 cells at a density of $1 × 10^5$ cells per well and cultured for three days until 80% confluency was reached. RNA was extracted using a Direct-zol RNA MicroPrep kit (Zymo Research) following the manufacturer's instructions. For each sample, 1 μg RNA was used as input for library preparation using the KAPA stranded mRNA-seq kit (Roche) according to the manufacturer's instructions. Unique dual-indexed set-B (UDI; Kapa Biosystems) adaptors were ligated and the library was amplified for eight cycles. The libraries were sequenced on a NovaSeq 6000 system as paired-end 100 with $50 × 10^6$ fragments per library (Fig. 3d,e and Extended Data Fig. 6e).

### Generation of DNA constructs for protein purification

For the purification of mEGFP- or mCherry-labelled fusion proteins, we amplified sequences from codon-optimized gene fragments (Twist Bioscience) for *HOXD4* wild type, AroLITE A, AroLITE G, AroLITE S, AroPLUS, AroPLUS patched, AroPLUS LITE, AroPLUS LITE patched and AroPERFECT; *HOXC4* wild type, AroLITE S and AroPERFECT; *HOXB1* wild type and AroLITE A; *NANOG* wild type and AroLITE A; *C/EBPα* wild type, AroLITE A, AroPERFECT IS15 and AroPERFECT IS10; and *NGN2* wild type, AroLITE A and AroPERFECT C IDRs. The primers used are listed in Supplementary Table 6. The amplified gene fragments were cloned into a pET45-mEGFP or pET45-mCherry backbone[21], linearized by restriction digest with AscI (NEB) and HindIII (NEB), via NEBuilder HiFi assembly. All sequences of interest were cloned C-terminally to the fluorescence marker.

## Protein purification

Overexpression of recombinant protein in BL21 (DE3) (NEB M0491S) was performed as described[20]. *Escherichia coli* pellets were resuspended in 25 ml of ice-cold Buffer A (50 mM Tris pH 7.5, 500 mM NaCl and 20 mM imidazole) supplemented with cOmplete protease inhibitors (Sigma, catalogue number 11697498001) and 0.1% Triton X-100 (Thermo Fischer Scientific, catalogue number 851110), and sonicated for ten cycles (15 s on, 45 s off) on a Qsonica Q700 sonicator. The bacterial lysate was cleared by centrifugation at 15,500$g$ and 4 °C for 30 min. For protein purification, we used the Äkta avant 25 chromatography system. All 25 ml of the cleared lysate was loaded onto a cOmplete His-Tag purification column (Merck, catalogue number 6781543001) pre-equilibrated in Buffer A. The loaded column was washed with 15×column volumes (CV) of Buffer A. Fusion protein was eluted in 10×CV of Elution Buffer (50 mM Tris pH 7.5, 500 mM NaCl and 250 mM imidazole) and diluted 1:1 in Storage Buffer (50 mM Tris pH 7.5, 125 mM NaCl, 1 mM dithiothreitol and 10% glycerol). The fractions enriched for GFP were pooled after His-affinity purification and manually loaded through an injection valve connected to a 500 µl capillary tube onto an equilibrated Superdex 200 increase 10/300 GL column (Cytiva, 28-9909-44). The loaded column was equilibrated with 0.15×CV of ice-cold Buffer A supplemented with cOmplete protease inhibitors. The fusion proteins were eluted with 1.1×CV of ice-cold Buffer A supplemented with cOmplete protease inhibitors. The elution fractions were pooled. The eluates were further concentrated by centrifugation at 10,000$g$ and 4 °C for 30 min using 3000 MWCO Amicon Ultra centrifugal filters (Merck, UFC803024). The concentrated fraction was diluted 1:100 in Storage Buffer, re-concentrated and stored at −80 °C.

## In vitro droplet fusion and surface wetting assay

For the in vitro fusion and surface wetting assays, we measured the concentration of purified mEGFP-tagged fusion proteins using a NanoDrop 2000 system (Thermo Fischer Scientific) and subsequently diluted the measured protein preparations to 50 µM in Storage Buffer. The protein preparations were mixed 1:1 with 5 µl of 20% PEG 8000 in de-ionized water (wt/vol). The resulting 10 µl was immediately pipetted on a chambered coverslip (Ibidi, 80826-90). Images of the contact interface between the drop and the slide were acquired using an LSM880 confocal microscope equipped with a plan-apochromat ×63, numerical aperture (NA) = 1.40 oil DIC objective with a ×5 zoom, resulting in a lateral pixel resolution of 0.04 µm. A total of 25 images were taken in a time series with 15 s intervals for each video (Supplementary Videos 1–16). C/EBPα droplet fusion and surface wetting assays were performed with different protein preparations as the in vitro droplet formation assay.

## In vitro droplet assay

For the in vitro droplet formation experiments (Figs. 1g, 2b, 4b and Extended Data Figs. 2c,g, 4h, 9b), we measured the concentration of purified mEGFP IDR fusion proteins using a NanoDrop 2000 system (Thermo Fischer Scientific) and subsequently diluted the protein preparations to the required concentration in Storage Buffer. The in vitro droplet formation assay was performed as previously described[21]. The protein preparations were mixed 1:1 with 5 µl of 20% PEG 8000 in de-ionized water (wt/vol) and equilibrated for 30 min at room temperature. The resulting 10 µl was pipetted on a chambered coverslip (Ibidi, 80826-90). After equilibration for 3 min, images of the drop on the slide were acquired with an LSM880 confocal microscope equipped with a Plan-Apochromat ×63, NA = 1.40 oil DIC objective with a ×2.5 zoom, resulting in a lateral pixel resolution of 0.04 µm, if indicated. Quantification of condensate formation was based on at least ten images acquired in at least two independent image series per condition.

## Image analysis of in vitro droplet formation

Protein droplets were detected using the ZEN blue 3.4 Image Analysis and Intellesis software packages. By use of a previously trained Intellesis model in spectral mode, we achieved image segmentation of individual pixels into objects (droplet area) or background (image background). A minimum cutoff of 120 nm in diameter was applied on the identified objects. Relative amounts of condensed protein were calculated by dividing the sum of mEGFP signal in objects defined as droplet area by the overall sum of mEGFP signal in the field of view. All values were calculated using RStudio. Plots were generated using GraphPad Prism 9. To fit data to a sigmoidal curve, we applied the in-built nonlinear regression function (Sigmoidal; $x$ is the concentration; Figs. 1h, 2c and Extended Data Figs. 2d,h, 4i, 7a, 9c).

## FRAP

FRAP experiments on droplets were formed as described above without 30 min of pre-assembly at room temperature and a protein concentration of 25 µM. The droplets were bleached immediately after pipetting the protein mixture onto the slide using ten iterations of 488 nm light at 70% laser power. Bleaching was performed on a central region of a settled single droplet. Fluorescence recovery was measured over a time course of 60 s at intervals of 2 s. Quantification of FRAP data was based on at least ten images acquired in at least two independent image series per condition. The resulting signal recovery was normalized to the background and fitted to a power law model in Microsoft Excel. All figures were generated using GraphPad Prism 9 (Figs. 2d, 4c, 6b and Extended Data Fig. 4a,j,k).

## Generation of DNA constructs for transactivation assays

To study the transactivation strength of TF IDRs, we amplified sequences from codon-optimized gene fragments (Twist Bioscience) using the primers listed in Supplementary Table 6. The amplified gene fragments were cloned into a pGAL4 (Addgene, catalogue number 145245) backbone, linearized with AsiSI (NEB) and BsiWI (NEB), via NEBuilder HiFi assembly.

## Generation of DNA constructs for TF-IDR tiling assays

To control for the potential creation of short linear motifs in TF-IDR mutants, we tiled up the *HOXD4* wild-type and AroPERFECT, C/EBPα wild-type and AroPERFECT IS15, *OCT4* wild-type C and AroPERFECT C, *MYOD1* wild-type C and AroPERFECT C, and *EGR1* wild-type and AroSCRAMBLED IDRs into 40-amino-acid segments with 20-amino-acid overlaps. We amplified all 40-amino-acid tiles in steps of 20 amino acids starting from the first amino acid of the sequence using the primers listed in Supplementary Table 6. The amplified gene fragments were cloned into a pGAL4 backbone, linearized with AsiSI (NEB) and BsiWI (NEB), via NEBuilder HiFi assembly (Figs. 2e,4f and Extended Data Figs. 5d,f,10b).

## Transactivation assay

The transactivation activity of TF IDRs was assayed using the Dual-Glo Luciferase Assay system (Promega). Mouse embryonic stem cells were seeded at a density of $1 \times 10^5$ cells cm$^{-2}$ on 24-well plates that had been pre-coated with gelatin. For feeder-free culture conditions, mESC medium was supplemented with 2× leukemia inhibitory factor (LIF). HEK293T, SH-SY5Y and Kelly cells as well as C2C12 mouse myoblasts were seeded on 24-well plates at a density of $1 \times 10^5$ cells cm$^{-2}$. After 24 h, every well was transfected with 200 ng pGal4 empty vector control or the equimolar amount of the expression construct carrying an IDR of interest, 250 ng of the firefly luciferase expression vector (Promega) and 15 ng of the *Renilla* luciferase expression vector (Promega) using FuGENE HD transfection reagent (Promega) according to the manufacturer's instructions. After another 24 h, the cells were washed once with PBS and lysed in 100 µl of 1×Lysis Passive Buffer (Promega) for 15 min on a shaker at room temperature. Subsequently, 10 µl of cell lysate was pipetted, in triplicate, onto a white-bottomed 96-microwell plate, followed by quantification of the firefly and *Renilla* genes using the Dual-Glo Luciferase Assay System Quick Protocol for 96-well plates

(Promega). Triplicate data were normalized to *Renilla* luminescence of the respective well and finally normalized to the empty vector control. Data are shown as the mean ± s.d. All data shown were generated from three independent transfections from at least two cell passages (Figs. 1i, 2a,e–g, 4a,f,g, 5a, 7a and Extended Data Figs. 2e,i–m, 4d,f, 5a,e,g, 7c, 9a) and were plotted using GraphPad Prism 9. Two-tailed Student's *t*-tests were performed to assess statistical significance.

### Western blots
Cultured cells were washed twice in PBS and lysed in RIPA buffer for 30 min on an orbital shaker at 4 °C. Subsequently, the cell lysate was centrifuged for 20 min at 20,000*g*. The cleared lysate was transferred to a new tube and total protein was quantified by BCA assay (Thermo Fischer Scientific). Extracted protein (20 µg) was run on a 4–12% NuPAGE SDS gel and transferred onto a polyvinylidene fluoride membrane using an iBlot2 dry gel transfer device (Invitrogen) following the manufacturer's instructions. For GAL4-DBD blots, 50 µg of extracted protein was used. The membranes were blocked with 5% skim milk in TBST and incubated overnight with primary antibodies at 4 °C. The primary antibodies used in this study include antibodies to IFI16 (Santa Cruz Biotechnology, sc-8023; 1;200), GFP (Invitrogen, A11122; 1:2,000), HSP90 (BD, 610419; 1:4,000), ARHGAP4 (Santa Cruz Biotechnology, sc-376251; 1:200), ESX1 (Santa Cruz Biotechnology, sc-365740; 1:200), GAL4-DBD (Santa Cruz Biotechnology, sc-510; 1:200) and FLAG (Merck, F1804; 1:2,000). Horseradish peroxidase-conjugated secondary antibodies to the host species were used at dilutions of 1:3,000–1:5,000 and visualized with HRP substrate SuperSignal West Dura (Thermo Fischer Scientific; Fig. 3f and Extended Data Figs. 4b,g, 5c, 6f, 7b, 10a).

### Generation of DNA constructs for locus reconstruction assays
To confirm mutant-specific regulation of C/EBPα target promoters and enhancers, we amplified promoter and enhancer regions of *GBP5*, *FAM98A* and *S100A* using the primers listed in Supplementary Table 6. The amplified fragments were cloned into a pGL3-Basic vector (Promega), linearized with BamHI (NEB) and SalI (NEB) in case of an enhancer region or with HindIII (NEB) and KpnI (NEB) in case of a promoter, via NEBuilder HiFi assembly. Full-length C/EBPα wild type and AroPERFECT IS15 sequences for overexpression were cloned into a pGAL4 backbone, linearized with EcoRI (NEB) and AsiSI (NEB), via NEBuilder HiFi assembly.

### Locus reconstruction with pGL3 reporter assays
Transcription factor activity at genomic loci was assayed using the Dual-Glo Luciferase Assay system (Promega). Mouse embryonic stem cells were seeded at a density of $1 \times 10^5$ cells cm$^{-2}$ on 24-well plates that had been pre-coated with gelatin. For feeder-free culture conditions, mESC medium was supplemented with 2× leukemia inhibitory factor (LIF). After 24 h, every well was transfected with 200 ng of plasmid containing a C/EBPα wild type or AroPERFECT IS15 overexpression cassette, 250 ng of pGL3-Basic control of an equimolar amount of the pGL3 construct carrying enhancer/promoter sequences of interest and 15 ng of the *Renilla* luciferase expression vector (Promega) using FuGENE HD transfection reagent (Promega) following the manufacturer's instructions. After a further 24 h, the cells were washed once with PBS and lysed in 100 µl 1×Lysis Passive Buffer (Promega) for 15 min on a shaker at room temperature. Subsequently, 10 µl of the cell lysate was pipetted, in triplicate, onto a white-bottomed 96-microwell plate, followed by quantification of the firefly and *Renilla* genes using the Dual-Glo luciferase assay system quick protocol for 96-well plates (Promega). Triplicate data were normalized to the *Renilla* luminescence of the respective well and then normalized to the pGL3-Basic vector control. Data are shown as the mean ± s.d. All data shown were generated from three independent transfections from at least two cell passages (Fig. 5l) and were plotted using GraphPad Prism 9. Two-tailed Student's *t*-tests were performed to assess statistical significance.

### LacO-LacI tethering assay
For the LacO-LacI tethering experiments (Figs. 3g and 4d), we used a vector containing CFP–LacI, followed by a previously published multiple cloning site[20]. The RNAPII-CTD plasmid was cloned via digestion with AsiSI (NEB) and BsiWI (NEB) using the NEBuilder HiFi assembly master mix.

The tethering experiments were adapted from a previous report[20]. Imaging was performed on live U2OS cells 48 h after transfection with 100 ng CFP–LacI-HOXD4 wild type, HOXD4 AroPERFECT, C/EBPα wild type or C/EBPα AroPERFECT IS15 plasmid and 100 ng RNAPII-CTD–YFP-NLS using the FuGENE HD transfection reagent. Images were acquired using an LSM880 confocal microscope equipped with a plan-apochromat ×63 NA = 1.40 oil DIC objective with a ×2 zoom. The laser intensities were adjusted before imaging to prevent possible channel bleed. Images were acquired across two experimental replicates.

### LacO-LacI tethering assay analysis
For the analysis of LacO-LacI images (Figs. 3h and 4e), regions of interest corresponding to CFP–LacI-IDR fusion proteins were detected manually based on the cyan channel using ImageJ v2.0.0. The mean intensities of these selected regions of interest were measured in both the YFP and CFP channels. The background intensity of the YFP channel was defined using a mean intensity measurement of a random nuclear region of the same size and shape as the primary region of interest. Enrichment of the YFP signal in the regions of interest (predefined by the CFP signal) was calculated by dividing the YFP mean signal intensity of the region of interest by the YFP mean signal intensity of the random nuclear region. Values were plotted as indicated using GraphPad Prism 9; *n*, number of observations.

### Generation of HAP1 cells expressing DOX-inducible *HOXD4* transgenes with the PiggyBac system
To generate a DOX-inducible overexpression system of HOXD4, we randomly integrated the coding sequences of wild-type, AroPERFECT and AroPLUS *HOXD4* into HAP1 cells using the PiggyBac transposon system.

N-terminally FLAG-tagged coding sequences of human wild-type, AroPERFECT or AroPLUS *HOXD4* (Twist Bioscience) with a downstream 5×GS-linker (Sigma) were cloned into a backbone of the inducible Caspex expression vector (Addgene, catalogue number 97421), linearized by restriction digest with NcoI (NEB) and KpnI (NEB). Carrier plasmids and PiggyBac transposase expression vector (SBI, PB210PA-1) were cotransfected at a molar ratio of 6:1 into wild-type HAP1 cells using Lipofectamine 3000 according to the manufacturer's instructions. The transfected bulk population was screened for integration by addition of 2 µg ml$^{-1}$ puromycin (Gibco) to the cell culture medium 24 h after transfection for a total of four days. Bulk populations of every condition were induced by addition of 2 µg ml$^{-1}$ DOX (Sigma) and screened for matching mEGFP expression levels across conditions using flow cytometry. For the generation of clonal HOXD4 overexpression lines, bulk cells were single-cell sorted by FACS. HAP1 *HOXD4* cells were directly sorted into wells of a 96-well plate. Wells without any cells or with more than two cells were discarded. The other clones were expanded and eventually tested for HOXD4 expression following DOX induction by FACS (Extended Data Fig. 6h). Cells with the most similar expression levels were selected for further experiments.

### Generation of DOX-inducible NGN2 overexpression systems in human iPSCs
To generate a DOX-inducible overexpression system of NGN2, we randomly integrated the coding sequences of wild-type, AroLITE and AroPERFECT *NGN2* into ZIP13K2 cells using the PiggyBac transposon system.

N-terminally FLAG-tagged coding sequences of human wild-type, AroLITE or AroPERFECT *NGN2* (Twist Bioscience) with a downstream T2A tag (Sigma) were cloned into a backbone of the inducible Caspex expression vector linearized by restriction digest with NcoI (NEB) and

KpnI (NEB). Carrier plasmids and PiggyBac transposase expression vector were cotransfected at a molar ratio of 6:1 into wild-type ZIP13K2 cells using Lipofectamine stem transfection reagent (Thermo Fischer Scientific) following the manufacturer's instructions. The transfected bulk population was screened for integration by addition of 2 µg ml⁻¹ puromycin (Gibco) to the cell culture medium 24 h after transfection for a total of four days. The surviving cells were seeded at low density with added 1×Y-27632 Rho-kinase inhibitor (biogems, 1293823) for the first 24 h and expanded for several days until colonies derived from single cells were big enough to be picked and cultured separately. Clones of every condition were induced by addition of 2 µg ml⁻¹ DOX (Sigma) and screened for matching mEGFP expression levels across conditions using flow cytometry.

### Generation of DOX-inducible MYOD1 overexpression lines in C2C12 cells

To generate a DOX-inducible overexpression system of MYOD1, we randomly integrated the coding sequences of wild-type, AroLITE, AroPERFECT C and AroLITE C *MYOD1* into C2C12 cells using the PiggyBac transposon system.

N-terminally FLAG-tagged coding sequences of human wild-type, AroLITE, AroPERFECT C or AroLITE C *MYOD1* (Twist Bioscience) with a downstream T2A tag (Sigma) were cloned into a backbone of the inducible Caspex expression vector linearized by restriction digest with NcoI (NEB) and KpnI (NEB). Carrier plasmids and PiggyBac transposase expression vector were cotransfected at a molar ratio of 6:1 into wild-type C2C12 cells using Lipofectamine 3000 transfection reagent following the manufacturer's instructions. The transfected bulk population was screened for integration by addition of 2 µg ml⁻¹ puromycin (Gibco) to the cell culture medium 24 h after transfection for a total of four days. Cells of every condition were induced by addition of 2 µg ml⁻¹ DOX (Sigma) and screened for matching mEGFP expression levels across conditions by flow cytometry.

### Imaging of HAP1 HOXD4 PiggyBac overexpression cells

For the subnuclear localization analysis of HOXD4 mutants, HAP1 cells with integrated HOXD4 overexpression cassettes were seeded onto chambered coverslips. After 24 h, the culture medium was substituted with 2 µg ml⁻¹ DOX to induce expression of *HOXD4* transgenes. The following day, the cells were washed with PBS and fixed with 4% paraformaldehyde for 15 min at room temperature. The cells were then stained with 0.25 µg ml⁻¹ DAPI (Invitrogen). Images were acquired using a Stellaris 8 confocal microscope and a plan-apochromat ×100 NA = 1.40 oil CS2 objective (Leica). For the analysis of subnuclear localization, a mosaic of at least 100 tile regions was imaged for each condition over two replicates. Object quantification was performed using the ZEN 3.4 software (Zeiss). Briefly, DAPI counterstain was used to segment objects after Gaussian smoothing. The mean mEGFP intensities were then individually calculated for each segmented nucleus and the granularity was calculated by dividing the s.d. of the mEGFP signal of each nucleus by the corresponding mean mEGFP signal using customer ImageJ/FIJI routines (Fig. 3b)[60].

### Imaging of HAP1 *HOXD4* knock-in cells

For imaging of HOXD4 knock-in cells, 2 × 10⁴ cells were seeded onto chambered coverslips. After 24 h, the cells were washed with PBS and fixed with 4% paraformaldehyde for 15 min at room temperature. The cells were permeabilized with PBS supplemented with 0.1% Tween-20 (Sigma) for 5 min and PBS supplemented with 0.25% Tween-20 for 15 min. The cells were then stained with primary (antibody-GFP; Invitrogen, A11122; 1:500) and secondary (goat anti-rabbit Alexa Fluor 594; Jackson ImmunoResearch, 2338059, 1:500) antibodies. Nuclei were stained with 0.25 µg ml⁻¹ DAPI. Images were acquired using a Stellaris 8 confocal microscope and a Plan-Apochromat ×100/1.40 oil CS2 objective (Leica). For the analysis of subnuclear localization,

a mosaic of at least 100 tile regions was imaged for each condition over two replicates. Object quantification was performed using the ZEN 3.4 software (Zeiss). Briefly, DAPI counterstain was used to segment objects after Gaussian smoothing. The mean mEGFP intensities were then individually calculated for each segmented nucleus and the granularity was calculated by dividing the s.d. of the mEGFP signal of each nucleus by the corresponding mean mEGFP signal using customer ImageJ/FIJI routines (Fig. 3a)[60].

### NGN2-mediated neural differentiation of human iPSCs

We adapted our protocol for the differentiation of human iPSCs into neurons by overexpression of NGN2 from a previous study[42]. ZIP13K2 cells with an integrated NGN2 overexpression cassette were cultured on 10 cm culture plates that had been pre-coated with Matrigel (Corning). When the cultures reached a confluency of approximately 80%, 2 µg ml⁻¹ DOX (Sigma) was added to the culture medium to induce expression of the *NGN2* transgene. After 24 h, the induced cultures were sorted for mEGFP-expressing cells by flow cytometry. Positive cells were seeded at a density of 2 × 10⁴ cells cm⁻² in mTeSR+ medium plus 1×Rho-kinase inhibitor on Matrigel-pre-coated 96-well microclear plates (Greiner bio-one). On day 2, the mTeSR+ medium was replaced with N2B27 neural cell culture medium supplemented with 5 µg ml⁻¹ human BDNF (Bio-Techne). The differentiation medium was changed every day for a total of four days. Living cells were stained with 0.25 µg ml⁻¹ Hoechst and Spy650-TUB (1:2,000; Spirochrome) and incubated in the microscope before image acquisition to equilibrate and thermalize all materials (Fig. 6d–f).

### KAPA stranded mRNA-seq of ZIP13K2 *NGN2* PiggyBac cells

On day 5 of NGN2-mediated neural differentiation, RNA was extracted from ZIP13K2 induced neurons following the Direct-zol RNA MicroPrep Kit (Zymo Research) standard protocol. Complementary DNA libraries were then prepared and sequenced as described earlier in the 'KAPA stranded messenger RNA-seq of HAP1 *HOXD4* knock-in cells' section (Fig. 6g,h and Extended Data Fig. 9f,g).

### Live-cell imaging of human iPSC-derived neurons

Living cells were imaged using the Celldiscoverer 7 imaging platform (Zeiss) in wide-field mode running under the ZEN Blue 3.1 imaging software and full environmental control (5% vol/vol CO₂, 100% humidity and 37 °C). The final experiments were performed using a plan-apochromat ×20, NA = 0.7 objective and a ×2 tube lens (Zeiss), and captured on an Axiocam 506 camera (Zeiss) with 3 × 3 binning, resulting in a lateral pixel resolution of 0.347 µm per pixel. The fully automated imaging approach typically captured 20–40% of individual well surfaces. Focus stabilization was achieved by surface method in each third tile region. All images were acquired with one or two additional transmitted light or contrasting method (brightfield, oblique or phase gradient contrast) channel. Each individual image position was acquired in consecutive sections of three slices surrounding the focus position with a *z*-spacing of 0.63 µm to ensure the acquisition of each and every neurite. All parameters were kept identical during the experimental time course. The resulting large overview tile scan underwent a maximum-intensity projection and subsequent channel stitching using the nuclear counterstain (Hoechst) as reference (Fig. 6d). We quantified cell numbers (Hoechst) and neurite density (SPY650) based on the respective channel.

### Image analysis of nuclei and neurite densities in differentiated neurons

Wide-field images were acquired using a ×20 air objective (NA = 0.7) with ×2 optical post magnification on a Celldiscoverer 7 microscope under the ZEN Blue 3.2 software (Zeiss). For each well and replicate, a mosaic of 201 tile regions was imaged. A definite hardware focus was defined as the centre for three slices of a consecutive *z*-stack with a slice

distance of 0.34 µm. Image acquisition was performed using a Zeiss Axiocam 506 camera in 3 × 3 binning mode, resulting in a lateral resolution of 0.34 µm per pixel. The resulting images were projected using maximum-intensity projection in a ZEN 3.4 on a dedicated Zeiss analysis workstation. Object quantification was performed in the image analysis module in ZEN 3.4 (Zeiss, Germany). Briefly, within maximum-intensity projections, nuclei were identified by nuclear counterstaining using Otsu intensity thresholds after faint smoothing (Gauss: 2,0) and nearby objects were segmented downstream by standard water shedding. Neurites were segmented by fixed intensity threshold on the respective staining without any water shedding (Fig. 6e,f).

### FLAG-NGN2 ChIP–Seq

To study the chromatin association of wild-type, AroLITE and AroP-ERFECT *NGN2*, we performed ChIP–Seq experiments in ZIP13K2 cells expressing the respective constructs 24 and 48 h after induction of NGN2-mediated neural differentiation (Fig. 6i,j and Extended Data Fig. 9h,i). The previously published ChIPmentation protocol was used[61].

The cells were detached using Accutase solution (Sigma), washed twice in PBS and fixed by incubation with 1% formaldehyde for 10 min at room temperature with rotation. Subsequently, the reaction was quenched by the addition of glycine to a final concentration of 125 mM. Per replicate, $3 \times 10^6$ cells were used as starting material. Briefly, we followed the ChIPmentation protocol version 3 for histone marks and TFs[62]. The cells were lysed in lysis buffer 3 (10 mM Tris–HCl pH 8.0, 100 mM NaCl, 1 mM EDTA pH 8.0, 0.5 mM EGTA, 0.1% sodium deoxycholate and 0.5% *N*-laurosylsarcosine) supplemented with 1×cOmplete protease inhibitor cocktail. The chromatin was then sonicated for 10 min using a Covaris E220 Evolution focused-ultrasonicator with 2% duty cycles, 105 W peak incident power and 200 cycles per burst. The lysates were clarified by centrifugation for 10 min at 20,000g and 10% of the clarified lysate was put aside as input control. The remaining lysate was mixed with 50 µl of equilibrated anti-FLAG (Merck, F1804; 1 µg total) coupled to Dynabeads Protein G magnetic beads (Invitrogen) and incubated on a 3D-shaker overnight at 4 °C. The next day, the samples were washed twice in TF-wash buffer I (20 mM Tris–HCl pH 7.4, 150 mM NaCl, 0.1% SDS, 1% Triton X-100 and 2 mM EDTA pH 8.0), followed by two washes in TF-wash buffer III (10 mM Tris–HCl pH 8.0, 250 mM LiCl, 1% Triton X-100, 0.7% sodium deoxycholate and 1 mM EDTA pH 8.0) and a final wash with 10 mM Tris–HCl pH 8.0. All samples were tagmented for 5 min at 37 °C using the Illumina Tagment DNA kit and immediately put on ice. The tagmented chromatin was washed twice in ice-cold wash buffer I and twice in TET buffer (10 mM Tris–HCl pH 8.0, 5 mM EDTA pH 8.0 and 0.2% Tween-20), and reverse-crosslinked for 1 h at 55 °C and 9 h at 65 °C in the presence of 300 mM NaCl and proteinase K (Ambion). Subsequently, DNA was purified using AMPureXP beads. Sequencing libraries were amplified using the Kapa HiFi HotStart ready mix (Roche) and Nextera custom primers (Illumina)[61] for a total of 12 cycles and paired-end sequenced on an NovaSeq 6000 system (Illumina) with a depth of approximately $50 \times 10^6$ fragments per library (Fig. 6i,j and Extended Data Fig. 9h, i).

### TT-SLAM-Seq

To study the immediate transcriptional effects of wild-type, AroLITE and AroPERFECT *NGN2* overexpression on ZIP13K2 human iPSCs, the cells were treated with DOX for 12 or 24 h and subjected to 15 min of 4-thiouridine labelling using 500 µM 4-thiouridine. TT-SLAM-Seq was performed as previously described[21].

### Image analysis of differentiated C2C12 myotubes

Wide-field images were acquired using a ×20 air objective (NA = 0.7) with ×2 optical post magnification on a Celldiscoverer 7 under the ZEN Blue 3.2 software (Zeiss). For each well and replicate, a mosaic of 49 tile regions was covered. We defined the definite hardware focus as the centre for three slices of a consecutive z-stack with a slice distance

of 0.34 µm. Image acquisition was performed using a Zeiss Axiocam 506 microscope, in 3 × 3 binning mode, resulting in a lateral resolution of 0.34 µm per pixel. The resulting images were projected using maximum-intensity projection in ZEN 3.4 (Zeiss) on a dedicated Zeiss analysis workstation. Quantification of fusion scores was conducted by implementation of a simple hierarchy order, which was built within the image analysis module in ZEN 3.4 (Zeiss). We designed two segregating parent classes by fixed intensity thresholds based on mEGFP signal resulting in fused myotubes and non-myotubes. Within these primary regions, nuclei were identified. Secondary objects were identified exclusively within primary objects (myotubes and non-myotubes) by applying Gaussian smoothing and fixed intensity thresholds on the nuclear counterstaining, followed by standard water shedding the respective fluorescence image. All nuclei objects were filtered according to an area between 30 and 300 µm² (Fig. 7d).

### C/EBPα-mediated transdifferentiation of B cells to macrophages

To induce C/EBPα-mediated B cell-to-macrophage transdifferentiation, infected RCH-rtTA cells were seeded at $0.3 \times 10^6$ cells ml⁻¹ in RCH culture medium supplemented with IL-2 (Preprotech, 200-03) and CSF-1 (Preprotech, 315-03B), both at 10 ng ml⁻¹, as well as 2 µg ml⁻¹ DOX. The macrophage transdifferentiation was monitored by flow cytometry. Briefly, blocking was carried out for 10 min at room temperature using a 1:20 dilution of human FcR binding inhibitor (eBiosciences, 16-9161-73). Subsequently, the cells were stained with antibodies to CD19 (APC–Cy7 mouse anti-human CD19; BD Pharmingen, catalogue number 557791) and Mac1 (APC mouse anti-human CD11b/Mac1; BD Pharmingen, catalogue number 550019) at 4 °C for 20 min in the dark. After washing, DAPI counterstaining was performed just before analyses. All analyses were performed using an LSR Fortessa instrument (BD Biosciences). Data analysis was completed using the FlowJo software (Fig. 5c and Extended Data Fig. 7e).

### FACS analysis of CD66a and FCGR2A

CD66 and FCGR2A expression levels were monitored by FACS analysis during C/EBPα-mediated transdifferentiation of B cells to macrophages. RCH-rtTA cells expressing DOX-inducible wild-type or AroPERFECT IS15 *CEBPA* were seeded at $0.5 \times 10^6$ cells ml⁻¹ in RCH culture medium supplemented with IL-2 and CSF-1, both at 10 ng ml⁻¹, as well as 2 µg ml⁻¹ DOX. The cells were collected at 24 and 48 h. Blocking was carried out for 10 min at room temperature using a 1:20 dilution of human FcR binding inhibitor. Subsequently, the cells were stained with antibodies to CD66a (Alexa Fluor 647 anti-human CD66a; BioLegend, catalogue number 398905) and FCGR2A (PE anti-human FCGR2A; BioLegend, catalogue number 305503) at 4 °C for 20 min in the dark. After washing, DAPI counterstaining was performed just before analysis. All analyses were performed using an LSR Fortessa instrument (BD Biosciences). Data analysis was completed using the FlowJo software (Extended Data Fig. 8n,q).

### Generation of scRNA-seq data

One week after induction of C/EBPα-mediated B cell-to-macrophage transdifferentiation, the cells were collected and washed twice in PBS to remove dead cells and debris. The cells were then resuspended in solution at a density of 700 cells µl⁻¹. We used the Chromium Next GEM Single Cell 3′ technology for generating gene expression libraries from single cells. Briefly, gel beads-in-emulsion (GEMs) are generated by the combination of barcoded Single Cell 3′ v3.1 Gel Beads, a master mix containing cells and partitioning oil on a Chromium Next GEM Chip G. To achieve single-cell resolution, the cells are delivered at a limiting dilution, such that the majority (approximately 90–99%) of generated GEMs contain no cell, whereas the remainder largely contain a single cell. Immediately following GEM generation, gel beads were dissolved, primers were released and any co-partitioned cell was lysed. Primers (containing an

Illumina TruSeq Read 1, 16 nucleotide 10X Barcode, 12 nucleotide unique molecular identifier and 30 nucleotide poly-dT sequence) were mixed with the cell lysate and a master mix containing reverse transcription reagents. Incubation of the GEMs produced barcoded full-length cDNA from poly-adenylated mRNA. After incubation, the GEMs were broken and pooled fractions were recovered. Silane magnetic beads were used to purify the first-strand cDNA from the post GEM-reverse transcription reaction mixture, which includes leftover biochemical reagents and primers. Barcoded full-length cDNA was amplified via PCR to generate sufficient mass for library construction. The cDNA was analysed using an Agilent Bioanalyzer assay (catalogue number 5067-4626) to check size distribution profile and for quantification. Only 25% of the cDNA was used for 3′ Gene Expression Library construction. Enzymatic fragmentation and size selection were used to optimize the cDNA amplicon size. TruSeq Read 1 (read 1 primer sequence) was added to the molecules during GEM incubation. P5, P7, a sample index and TruSeq Read 2 (read 2 primer sequence) were added via end repair, A-tailing, adaptor ligation and PCR. The final libraries contained the P5 and P7 primers used in Illumina bridge amplification. The final libraries were analysed using an Agilent Bioanalyzer assay to estimate the quantity and check size distribution, and were then quantified by quantitative PCR using a library quantification kit (Kapa Biosystems, catalogue number KK4835).

### C/EBPα–GFP ChIP–Seq

To study the chromatin association of C/EBPα wild type and AroPER-FECT IS15, we performed ChIP-Seq in C/EBPα wild type and AroPER-FECT RCH-rtTA cells 24 and 48 h after induction of C/EBPα-mediated macrophage transdifferentiation (Fig. 5f–h,j, Extended Data Fig. 8k,l,o and Supplementary Fig. 2a–c,e). The protocol was previously described[41]. The cells ($5 \times 10^6$) were collected, crosslinked for 10 min using 1% formaldehyde and quenched using a final concentration of 0.125 M glycine. After a wash in cold PBS and centrifugation, the pellets were lysed in 500 µl pre-cooled SDS lysis buffer (1% SDS, 10 mM EDTA, 50 mM Tris pH 8 and 1×protease inhibitor cocktail) and incubated on ice for 15 min. The chromatin was sheared using a Bioruptor Pico sonicator (Diagenode) at 4 °C for 18 cycles of 30 s on and 30 s off. After sonication, the solution was clarified by centrifugation at 1,000$g$ and 4 °C for 5 min; the supernatant was transferred to a low-bind tube and mixed with 900 µl ChIP dilution buffer (0.01% SDS, 1.1% Triton X-100, 1.2 mM EDTA, 16.7 mM Tris–HCl pH 8.0, 167 mM NaCl and 1×protease inhibitor cocktail) containing antibody-coupled beads (10 µl anti-GFP; clone 3E6, Thermo Fischer Scientific, A-11120, and 35 µl of protein G magnetic beads; Thermo Fischer Scientific, 10003D). Five per cent were saved as input and the samples were incubated overnight at 4 °C under rotation. The beads were then collected and washed with 500 µl low salt buffer (0.1% SDS, 1% Triton X-100, 2 mM EDTA, 20 mM Tris–HCl pH 8.0 and 150 mM NaCl), high salt buffer (0.1% SDS, 1% Triton X-100, 2 mM EDTA, 20 mM Tris–HCl pH 8.0 and 500 mM NaCl), RIPA-LiCl buffer (10 mM Tris–HCl pH 8.0, 1 mM EDTA, 250 mM LiCl, 0.5% NP-40 and 0.5% sodium deoxycholate) and twice with TE buffer (10 mM Tris–HCl pH 8.0 and 1 mM EDTA). The beads were then collected and eluted in 70 µl Elution buffer (10 mM Tris–HCl pH 8.0, 5 mM EDTA, 300 mM NaCl and 0.5% SDS), followed by incubation with proteinase K for 1 h at 55 °C and then overnight at 65 °C to reverse the crosslinking. The beads were collected and transferred to a new tube and a second step of elution was performed with 30 µl Elution buffer. Finally, DNA was purified using a Qiagen MinElute column and 3 ng DNA was used to construct sequencing libraries with a NEBNext ultra DNA library prep kit for Illumina (E7370L). The libraries were sequenced on Illumina NextSeq 2000 instruments using the 50 nucleotides single-end mode to obtain around $50 \times 10^6$ reads per sample.

### Identification of periodic blocks in TF IDRs

We used 1,392 full-length TF protein sequences from Animal Transcription Factor DataBase (AnimalTFDB) v3.0 (ref. 63) and determined the positions of all aromatic residues F, Y and W (stickers) within them. Next, we identified spacers—stretches of non-aromatic residues between the stickers. A periodic block of aromatic residues was defined as a region that comprises at least four aromatic amino acids. We considered spacer lengths of 4–9, 10–20 or 21–30 amino acids. The ranges of different spacer lengths used for the analysis were chosen based on previous modelling studies on biopolymers using the stickers-and-spacers formalism[64–66]. Next, we identified periodic blocks that overlap IDR regions using the Metapredict v2 IDR prediction network[67]. This resulted in the identification of periodic blocks of aromatic residues in 531 TF IDRs (Extended Data Fig. 1a,b, Supplementary Table 1). For an internal ranking of periodic TF IDRs, we calculated a periodicity score comprising the number of periodic blocks that overlapped with the protein IDRs. The three spacer subgroups were weighed by 1, 1.1 and 1.2 for the lengths of 4–9, 10–20 and 21–30 residues in a single spacer, respectively. The weighing values were arbitrarily chosen with the assumption that uniform aromatic dispersion with long spacers may be less likely to occur randomly (Extended Data Fig. 1a,b and Supplementary Table 1).

### Prion-like domain analysis

For all predictions, if not stated otherwise, the total human proteome was used from the GRCh38.p13 assembly. For this, we filtered all non-canonical proteins using Ensembl v104 annotation. For genes that did not have any isoform classified as 'Ensembl canonical', the longest 'Genecode basic' isoform was considered. The AnimalTFDB v3.0 database[63] was used as the reference set for annotating TFs and TF families (Fig. 1k,l and Extended Data Fig. 3d,e). Prion-like domains were identified using the PLAAC web application with default settings[68]. From the above described set of human proteins, aromatic-rich prion-like domains were defined as those with 10% of more aromatic content.

### Identification of intrinsically disordered protein regions

Intrinsically disordered protein regions were predicted using Metapredict with default settings using the Metapredict v2 network[67].

### Identification of regions with significant periodicity in the human proteome

We developed an in-house method to identify regions with significant, albeit not necessarily perfect, periodicity. Briefly, the number of residues between adjacent aromatic residues (that is, spacer length) was calculated for each protein and the observed distribution of spacer lengths within a sequence was compared with the expected geometric distribution using a K–S test. The mean of the geometric distribution was then extrapolated from the proportion of aromatic residues, implicitly modelling their occurrence by a Poisson process. Next, the method was applied to every 100-amino-acid-long region using a sliding window approach and the $P$ value of the K–S test was plotted against the position of each window in every protein. After plotting the $P$ value of every 100-amino-acid-long region of each protein, the consecutive points below a $P$-value threshold (0.5 × average $P$ value) were identified as periodic regions. Those regions were compared with the Metapredict IDRs and InterPro domain regions (https://www.ebi.ac.uk/interpro/), and overlap was defined as the overlap between regions of at least one amino acid. Only regions that contained at least five aromatic residues in the 100-amino-acid-window with the lowest $P$ value were included. Regions with significant periodicity were defined by the minimum $P$-value cutoff of 0.01. (Fig. 1k,l and Extended Data Fig. 3a,b). All regions are listed in Supplementary Table 2.

### Omega score calculation

The $\Omega_{Aro}$ score was calculated using a modified localCIDER version[69]. Given that the omega score function is not length normalized, we adapted the Python code to allow for variable interspace size referred in the package as the so-called blob size. This parameter is now calculated

by dividing the sequence length by the fraction of aromatic residues. For this analysis, only IDRs with a minimum of three aromatic residues were included. The mean random score was defined as the mean of 1,000 κ-score calculations of randomly shuffled sequence from the original sequence. The ggplot2 program (ref. [70]) was used for plotting violin plots and custom R to generate a distribution plot for the mean of random (Figs. 1f, 2a, 4a, 6a, 7a and Extended Data Figs. 2b, 4c,f, 5g, 7c, 9a). One-way analysis of variance with a post-Tukey test was used to compare IDR sets (Fig. 1l).

## Bulk RNA-seq analysis

RNA-seq raw data were filtered and trimmed using cutadapt[71] with default settings. Filtered data from HAP1 and ZIP13K2 cells were mapped to a custom human genome hg38 including the cloned mEGFP sequence using STAR aligner[72]. Count read tables were generated by the same program. C2C12 RNA-seq data were mapped to the mm10 mouse genome using the abovementioned programs. Differential expression analysis was performed using the DEseq2 package[73] in R version 4.2 (ref. [74]). Differentially expressed genes were defined as having a fold change ≥ 1.5, Benjamini–Hochberg $P ≤ 0.01$ and a minimum mean read count across the experiment samples of 50 reads. For the HAP1 dataset, knockout samples were compared with the parental lines, and AroPERFECT and AroPLUS were compared with the *HOXD4* wild-type line. For the ZIP13K2 datasets, the *NGN2* wild-type line was compared with the parental ZIP13K2 line. AroLITE and AroPERFECT *NGN2* were compared with the wild-type *NGN2* line. Genes were considered as NGN2 targets if they were differentially expressed in the parental ZIP13K2 versus wild-type *NGN2* comparison and had a peak assigned in the wild-type *NGN2* ChIP–Seq analysis. For the C2C12 experiments, we compared the gene expression in the wild-type *MYOD1* line with parental C2C12 cell gene expression and AroLITE, Aro-LITE C, AroPERFECT and AroPERFECT C variants with wild-type *MYOD1*. The differentially expressed genes are listed in Supplementary Table 4.

Principal component analysis was carried out using the PCAPlot function from the DEseq2 package on the normalized read matrix that was transformed using the variance stabilizing transformation function from the DEseq2 package and plotted using ggplot2 (Figs. 3d, 6g, 7e and Extended Data Fig. 10d). Volcano plots were plotted using ggplot2 (Figs. 3e, 6h, 7f and Extended Data Figs. 8i, 10e). Heatmaps were plotted with the aid of the ComplexHeatmap package[75] in R and cluster analysis was done by k-means clustering using the cluster[76] package in R (Extended Data Figs. 6e, 9f and 10f).

Gene-set-enrichment analysis of the *MYOD1* RNA-seq was conducted using GSEAPreranked v6.0.12 (ref. [77]) with 1,000 permutations on the ranked list of gene sets from the comparisons of AroPERFECT C versus wild type and wild type versus parental sorted according to the Wald statistic (stat)[73] against the Wikipathways cell adhesion gene set in *Mus musculus*[78] (Extended Data Fig. 10g). Empirical P values were used for the plots. Highest-ranking genes in the AroPERFECT-C versus wild type comparison that are *MYOD1* targets were highlighted in the volcano plots (Fig. 7f and Extended Data Fig. 10e).

The marker genes shown in Extended Data Fig. 9g were identified as single-cell cluster markers in *NGN2*-induced neural differentiation in previous studies[79,80].

## ScRNA-seq analysis

**Data pre-processing.** The scRNA-seq datasets were processed using 10X Genomics' Cell Ranger pipeline v3.1.0 (ref. [81]) and mapped to a custom human genome hg38 including mEGFP and codon-optimized wild-type, AroPERFECT IS15 and AroPERFECT IS10 C/EBPα sequences. The Cell Ranger hdf5 files were processed using the Seurat package v4.0.6 (ref. [82]).

**Filtering and normalization.** We kept cells with more than 2,000 expressed genes, and genes with >5 reads across the samples were considered for analysis. Further filtering was done by removing cells with >20% mitochondrial genes and <5% ribosomal gene expression. The top ten genes associated with PCA components were then checked for mitochondrial and ribosomal genes. Next, cells were scored for cell cycle and gene expression on S and G2M genes was regressed to eliminate any dependence on cell cycle to clustering. Doublets were also identified and filtered out. mEGFP and C/EBPα wild-type, AroPERFECT IS15 and AroPERFECT IS10 reads were then used to identify mEGFP+ cells, and their expression was then transposed to the metadata so it would not affect clustering. Finally, the Harmony package was used to batch correct the three libraries.

**Cluster identification.** Cluster identification was then carried out using Seurat's built-in functions FindvariableGenes, RunPCA, RunUMAP and FindClusters by first identifying the genes with the highest variation across all samples and cell types, building a shared-nearest-neighbour graph and then running the Louvain algorithm on it. The number of clusters was determined by the optimum of the modularity function from the Louvain algorithm. The number of mEGFP+ cells was then calculated for each cluster and this was used to filter untransformed cell clusters, mainly cluster 0 and cluster 2.

**Assignment of cell types to clusters.** Cell-type cluster assignment was based on the comparison of marker sets from a published bulk RNA-seq experiment[83] and augmented by both RNA velocity analysis and known markers for both B cell and macrophage cell types. Briefly, RNA-seq data and marker sets were retrieved from ref. [83] and raw FASTQ files, aligned and reads were counted using STAR aligner against the human genome v38. Raw count data were then processed in DESeq2 and normalized to the variance stabilizing transformation. Marker set variance stabilizing transformation data were then retrieved and clustered according to the methods described previously[83] and each gene was assigned a gene cluster for Early, Early–inter, Inter1, Inter2, Inter–late, Late1 and Late2 as described in the publication. This assignment was designated 'Choi et al. differentiation clusters' in Extended Data Fig. 8a,b. To quantify the number of genes that are highly expressed in each single-cell cluster, single-cell gene expression was averaged within the single-cell cluster and normalized to the z-score. Normalized gene expression for the abovementioned marker set was then clustered by k-means clustering with $k = 8$ in an effort to separate each single-cell cluster by expression profile and a heatmap was generated using complexHeatmap to visualize the expression profile (Extended Data Fig. 8a). For each k-means cluster, the gene list was retrieved and the number of terms of Choi et al. differentiation clusters was quantified for each cluster (Extended Data Fig. 8b). This analysis helped define the B cell and macrophage population and assigned them to differentiation stages. Pseudotime and PAGA graph analysis also was used to aid in the trajectory of by giving temporal context to the single-cell clusters. Based on the differentiation term quantification, the expression pattern of the marker set, pseudotime and PAGA graph, we manually assigned each single-cell cluster to a differentiation state as follows: clusters 0, 2 and 3 were considered as the earlier cell stage as they showed the least amount of marker cell induction and also the lowest pseudotime score. As mentioned earlier, clusters 0 and 2 were excluded based on mEGFP quantification (Extended Data Fig. 8d) and were considered as untransduced B cells. Cluster 3 cells were assigned as initial B cells. Cluster 4 was assigned to Early based on quantification high amount of Early and Early–Inter terms and based on difference in proportion of Early–Inter was higher for that cluster. Cluster 1 had similar term quantification but was assigned as Early–Inter based on PAGA analysis. Finally, clusters 5, 6 and 7 had the highest quantify of Inter2, Late1 and Late2 macrophage markers. Clusters 5 and 6 had very similar quantifications and were thus assigned Differentiating macrophage 2 and 1, respectively, based on PAGA analysis. Late macrophage assignment was based on the unique expression signature by having the highest pseudotime score (Extended Data Fig. 8c). To confirm this assignment,

we also used cell-type markers and visualized the normalized expression in a UMAP graph. Markers for B cells—CD19—and macrophage cell types—*ITGAM*, *CD14*, *CD68* and *PTPRC*—as well as *CEACAM1*, *CEACAM4*, *CEACAM6*, *CEACAM8*, *FCGR2A*, *FCGR2B* and *FCGR3A* were used (Fig. 5i,k, Extended Data Fig. 8f,m,p and Supplementary Fig. 2d,f).

**Differential expression analysis.** Inter-cluster differential expression analysis was performed using the Wilcoxon test using the FindMarkers function, with default settings, and inter-sample cluster differential expression analysis between wild-type and IS15 cells in cluster 7 was performed using the FindMarkers and DESeq2 functions. The differentially expressed genes within the clusters are listed in Supplementary Table 5. A *q*-value cutoff of 0.05 was used to define differentially expressed genes for the Wilcoxon test and an adjusted Benjamini–Hochberg *P* value of 0.05 was used for the inter-sample test (Supplementary Table 5). Volcano and bar plots were generated in ggplot2; and violin, UMAP and feature plots were generated using Seurat's VlnPlot, FeaturePlot and DimPlot functions. The dot plot was made using a custom function to modify the output of the complexHeatmap package (Extended Data Fig. 8i,j).

**RNA velocity.** We generated loop files necessary for RNA velocity using velocyto[84] and exported barcodes, expression matrix, metadata and UMAP coordinates from Seurat to CSV files. scVelo[85] was used to build the manifold, calculate and visualize the RNA velocity using generalized dynamical model to solve the full transcriptional dynamics. PAGA graph[86] was calculated from this model to visualize the cell trajectory. Pseudotime was calculated using the Markov diffusion process and plotted by the scVelo bult-in function (Extended Data Fig. 8c).

### ChIP–Seq analysis
ChIP–Seq data from C/EBPα and NGN2 were mapped to a custom human genome hg38 using BWA v0.7.17 (ref. 87). SAMtools[88] was used for SAM to BAM file conversion, sorting and indexing, and Genome Analysis Toolkit v4 (ref. 89) was used to remove duplicate reads. Peak calling was then performed using MACS3 v3.0.0 b1 (ref. 90) using the input of the respective sample. Analysis and differential peak calling were done with DiffBind v3.6.5 (ref. 91). Normalization was done with the native method and background input. Differential calling was done using the DESeq2 method; the false-detection-rate threshold was set to 0.01. Peak visualization was performed using the DiffBind 'plotprofile' function with default settings for general profiles, unless otherwise stated. Set of overlapping sites was done using bedtools v2.6.0 and the intersect function. The profiles in Supplementary Fig. 2b were plotted using 'percentOfRegion' with 27 windows and 300% extension. The regions plotted correspond to a merged set of promoters, a merged set of enhancers and separate sets for B cell and macrophage superenhancers from a previous study[83]. Principal component analysis was done on normalized count samples and plotted with DiffBind (Extended Data Figs. 8k and 9h).

### TT-SLAM-Seq analysis
Raw reads were filtered and trimmed as described earlier for bulk RNA-seq samples. Filtered reads were aligned to the SILVA database69 (downloaded 6 March 2020) using STAR v2.7.9a with the parameters '–outFilterMultimapNmax 50 –outReadsUnmapped Fastx' to remove ribosomal RNA content. Unaligned reads were then reverse-complemented using the seqtk 'seq' v1.3-r106 using the '-r' parameter (https://github.com/lh3/seqtk). Reverse-complemented reads were processed using SLAM-DUNK[92] with the 'all' pipeline v0.4.1 using the '-rl 100 -5 0' parameters with the GENCODE gene annotation v39 as '-b' option. Reads with a 'T>C' conversion representing nascent transcription were filtered from the BAM files using alleyoop (provided together with SLAM-DUNK) with the 'read-separator' command. Counts per gene were quantified based on the 'T>C'-converted reads using featureCounts v2.0.6 (ref. 93) with the -s 1 and -t gene parameters for stranded and gene body counting. Samples were then submitted to

differential expression using the method described above. Heatmap representation was plotted as described earlier (Extended Data Fig. 9k). For genome-wide coverage tracks, technical replicates were merged using SAMtools 'merge'. BigWig files for single and merged replicates were obtained as described above. DeepTools2 v3.5.1 (ref. 94) was used to generate a metaplot using two separate BED files containing separate stranded genes in each file (Fig. 6k).

### Sequence disorder and pLDDT calculation for HNRNPA1
Disorder and pLDDT scores were calculated using Metapredict v2, and score plots were made using the built-in Metapredict graph plotting function (Extended Data Fig. 3a).

### AlphaFold predicted models
AlphaFold models were computed by an in-house implementation of AlphaFold[95] using version 2.0.0 (16 July 2021). The preset parameter was set to '–preset = casp14', matching the CASP14 prediction pipeline. In addition, templates were restricted to those available before the CASP14 predictions using the parameter –max_template_date = 2020-05-14. Models were rendered using UCSF ChimeraX, colouring the structure for aromatic residues (Extended Data Figs. 3c and 5a).

### Spacer analysis
The IDR composition was measured by calculating the frequency of each amino acid as a probability with the 'alphabetFrequency' function from Biostrings package v2.40.2 divided by the frequency of the amino acid calculated over the full human proteome in R. Quantification was performed for IDRs with and without periodic blocks. The frequency bar chart was plotted using ggplot (Extended Data Fig. 1d). To calculate the amino acid composition around the aromatic residues, we extracted the sequence, in FASTA format, of every periodic block for positions −2, −1, 0, +1 and +2 around the aromatic residue (0 represents the aromatic residue) using custom Python script. The FASTA file was then submitted to GLAM2 analysis to calculate the frequency of amino acids and to output a position weigth matrix. The cumulative bar plot was plotted using ggplot masking the position weigth matrix table into disorder promoting, order-promoting and neutral residues (Extended Data Fig. 1e). Periodic block motif analysis was performed by extracting sequences of the periodic blocks in TF IDRs described in this study, and charged blocks from a previous study[96], in FASTA format and submitting them to GLAM2 analysis. The top three position weigth matrices were plotted (Extended Data Fig. 1f).

### Gene-set-enrichment analysis
Gene ontology enrichment analyses for proteins that contain a regions with significant periodicity and the TFs that contain a periodic block were done using gProfiler[97]. Gene ontology categories for biological process were filtered for term size of >1,000 genes to remove general categories. An adjusted *P*-value cutoff of 0.001 was used. For periodic block containing TFs analysis REAC and WikiPathways enrichment was also done with gProfiler. Gene-set-enrichment analysis was done using clusterProfiler[98,99] (Extended Data Fig. 3d,e).

### UCSC track visualization
For track visualization, MACS3 backgroup-subtracted bigWig files from each replicate were merged using the UCSC bigWigMerge tool and then converted from big bedGraph format back into bigWig using the UCSC bedGraphToBigWig tool. Visualization was done using the pygenometracks tool set[100].

### Statistics and reproducibility
All experiments were repeated as stated in the figures, legends and methods. Statistical details are presented in the figure legends and as detailed below. Comparisons were performed in GraphPad Prism 9.0. No statistical method was used to pre-determine sample size. Data distribution was assumed to be normal but this was not formally tested.

The experiments were not randomized. Data collection and analysis were not performed blind to the conditions of the experiments. For the neural reprogramming experiments, wells were excluded in case of wash-off or out-of-focus events. Investigators were not blinded to allocation during experiments and outcome assessment.

In the box plots in Fig. 1l, the centre line shows the median, the bounds of the box correspond to interquartile (25th–75th) percentile, and whiskers extend to Q3 + 1.5× the interquartile range and Q1 – 1.5× the interquartile range. Dots beyond the whiskers show Tukey's fences outliers.

Exact $P$ values were as follows: Fig. 2a, $P_{\text{(WT versus AroPLUS)}} = 0.03172$, $P_{\text{(AroPLUS versus AroPLUS LITE)}} = 0.07727$, $P_{\text{(AroPLUS versus AroPLUS patched)}} = 0.00729$, $P_{\text{(AroPLUS versus AroPLUS patched LITE)}} = 0.03433$, $P_{\text{(WT versus AroPERFECT)}} = 0.00006$, $P_{\text{(AroLITE versus AroPERFECT)}} = 0.00004$, $P_{\text{(AroPLUS versus AroPERFECT)}} = 0.000461$, $P_{\text{(AroPLUS LITE versus AroPERFECT)}} = 0.00252$, $P_{\text{(AroPLUS patched versus AroPERFECT)}} = 0.00014$ and $P_{\text{(AroPLUS patched LITE versus AroPERFECT)}} = 0.00008$; Fig. 2f, $P_{\text{(WT(N)-FUSNxs versus wild type)}} = 0.00942$, $P_{\text{(WT(N)-FUSNxs versus wild type (N))}} = 0.01837$ and $P_{\text{(WT(N)-FUSNxs versus FUSNxs)}} = 0.01054$; Extended Data Fig. 5b(top), $P_{\text{(wild-type N-IDR versus AroPERFECT N-IDR)}} = 0.00121$, $P_{\text{(AroLITE N-IDR versus AroPERFECT N-IDR)}} = 0.000003$, $P_{\text{(wild-type C-IDR versus AroPERFECT C-IDR)}} = 0.01711$, $P_{\text{(AroLITE C-IDR versus AroPERFECT C-IDR)}} = 0.000005$; Extended Data Fig. 5b(middle), $P_{\text{(wild type versus AroPERFECT)}} = 0.02946$ and $P_{\text{(AroLITE versus AroPERFECT)}} = 0.00069$ (middle); Extended Data Fig. 5b(bottom), $P_{\text{(wild type versus AroPERFECT)}} = 0.02079$, $P_{\text{(AroLITE versus AroPERFECT)}} = 0.02087$ (bottom); Extended Data Fig. 6j, $P_{\text{(HOXD4 wild-type YFP versus HOXD4 wild-type YFP-RNAPII CTD)}} = 0.9999$, $P_{\text{(HOXD4 wild-type YFP versus HOXD4 AroPERFECT YFP)}} = 0.0509$, $P_{\text{(HOXD4 AroPERFECT YFP versus HOXD4 AroPERFECT YFP-RNAPII CTD)}} = 0.0325$, $P_{\text{(HOXD4 wild-type YFP-RNAPII CTD versus HOXD4 AroPERFECT YFP-RNAPII CTD)}} = 0.9999$, $P_{\text{(C/EBP}\alpha\text{ wild-type YFP versus C/EBP}\alpha\text{ AroPERFECT YFP)}} = 0.9999$, $P_{\text{(C/EBP}\alpha\text{ wild-type YFP versus C/EBP}\alpha\text{ wild-type YFP-RNAPII CTD)}} = 0.9999$, $P_{\text{(C/EBP}\alpha\text{ AroPERFECT YFP versus C/EBP}\alpha\text{ AroPERFECT YFP-RNAPII CTD)}} = 0.1524$ and $P_{\text{(C/EBP}\alpha\text{ wild-type YFP-RNAPII CTD versus C/EBP}\alpha\text{ AroPERFECT YFP-RNAPII CTD)}} = 0.2275$.

The imaging experiments in Figs. 3a, 4b, Extended Data Fig. 6g and Supplementary Fig. 1a,c were performed twice independently with similar results. The imaging experiments in Extended Data Fig. 10c were performed three times independently with similar results. The western blot experiments in Fig. 3f and Extended Data Figs. 4b,g, 5c, 6f, 7b, 10a were performed twice independently with similar results. The genotyping experiments in Extended Data Fig. 6c were performed once.

### Reporting summary

Further information on research design is available in the Nature Portfolio Reporting Summary linked to this article.

## Data availability

Sequencing data were deposited at the Gene Expression Omnibus (GEO) under the accession number GSE201655. Plasmids were deposited at Addgene (accession numbers 215570–215644). Raw data, except for large wide-view microscopy images, were deposited at Zenodo under https://doi.org/10.5281/zenodo.10628753 (ref. 101). The complete set of raw and processed data are available at https://owww.molgen.mpg.de/~TFsuboptimization/. Source data are provided with this paper. All other data supporting the findings of this study are available from the corresponding author on reasonable request.

## Code availability

All relevant code is made available on GitHub under the link https://github.com/hniszlab/TFsubopt. The Periodic Block finder code is made available in the link https://github.com/alexpmagalhaes/PeriodicBlock_finder. The QuasiIDRfinder code is made available in the link https://github.com/gozdekibar/QuasiIDRfinder.

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

## Acknowledgements

We thank the members of the Hnisz laboratory for helpful discussions. We thank S. Stricker for advice on the myotube differentiation experiments, G. Masserdotti and M. Götz for advice on the neural differentiation experiments, I. Stöppelkamp for help with the LacO-LacI experiments and H. Niskanen for help with the TT-SLAM-Seq experiments. We thank the Max Planck Institute for Molecular Genetics (MPI-MG) Sequencing Core and the Centre for Genomic Regulation Genomics unit for sequencing, the MPI-MG FACS facility for help with FACS, and the MPI-MG imaging facility for help with imaging. This work was funded by the Max Planck Society and partially supported by grants from the Deutsche Forschungsgemeinschaft (DFG) SPP2202 Priority Program Grant HN 4/3-1 (to D.H.), the Spanish Ministry of Science and Innovation (EMBL Partnership, Severo Ochoa Centre of Excellence) and the CERCA Program/Generalitat de Catalunya (to T.G.). G.K. was partially supported by a BMBF grant (grant number FKZ 031L0169A to M.V.). M.C.K. was supported by Grant PCI2021-122032-2B funded by MICIU/AEI/ 10.13039/501100011033 by the European Union NextGenerationEU/PRTR.

## Author contributions

Conceptualization: J.N., A.P.M., G.K., G.S., Y.Z., H.M.W., F.R., M.C.-K., S.D.M., T.G., M.V. and D.H. Investigation: J.N., G.S., Y.Z., H.M.W., F.R. and M.A.-B. Methodology: J.N., A.P.M., G.K., R.B. and S.D.M. Formal analysis: J.N., A.P.M., G.K., G.S. and Y.Z. Resources: T.G., M.V. and D.H. Visualization: J.N., A.P.M., G.K., G.S., Y.Z., H.M.W., R.B. and S.D.M. Writing (original draft): J.N. and D.H. Software: A.P.M., G.K. and S.D.M. Supervision: T.G., M.V. and D.H. Funding acquisition: T.G., M.V. and D.H. G.K. and G.S. contributed equally to the manuscript.

## Funding

## Competing interests

The Max Planck Society has filed a patent application (EP23215195) based on the study. D.H. is a founder and scientific advisor of Nuage Therapeutics. The other authors declare no competing interests.

## Additional information

**Extended data** is available for this paper at https://doi.org/10.1038/s41556-024-01411-0.

**Correspondence and requests for materials** should be addressed to Denes Hnisz.

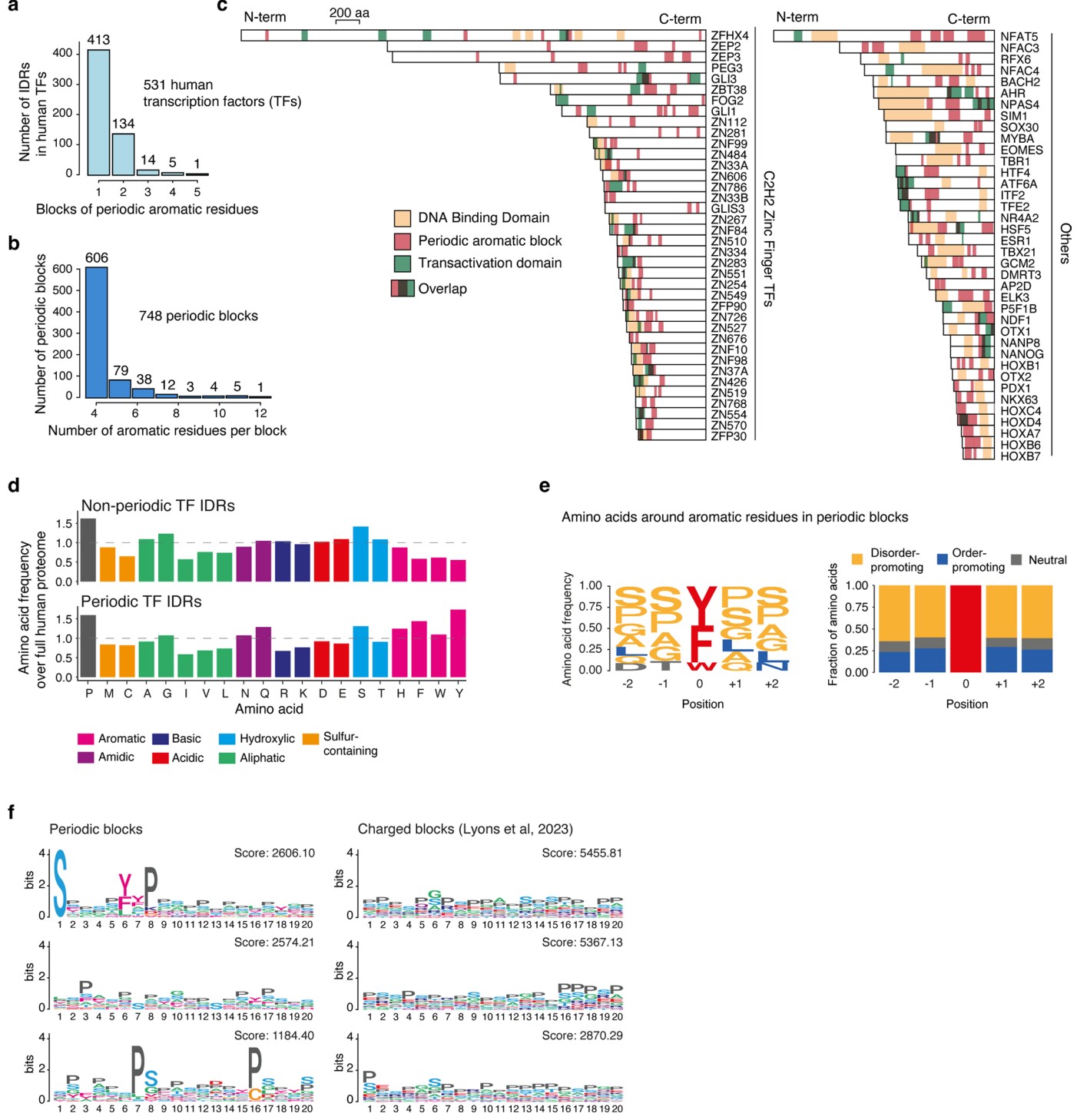

**Extended Data Fig. 1 | Characterization of periodic blocks in human TF IDRs. a.** Distribution plot of the 531 human TFs that contain short periodic blocks overlapping their intrinsically disordered regions (IDRs). Most TF IDRs overlap one short periodic block. **b.** Distribution plot of the 748 periodic blocks of aromatic amino acids in human TF IDRs. Most periodic blocks consist of 4 aromatic residues. **c.** Domain annotation of the 80 human TFs with the highest IDR periodicity score. Zinc finger TFs are shown on the left, members of all other TF families on the right. The majority of periodic blocks do not overlap 'minimal' activation domains. **d.** Frequency of amino acids in non-periodic, and periodic TF IDRs, relative to their frequencies in the full proteome. Note that periodic TF IDRs are relatively enriched for aromatic residues, depleted for charged residues, and enriched for neutral residues. **e.** Amino acid PWM and cumulative bar frequency plot around aromatic residues in periodic blocks. Colours represent disorder promoting (yellow), order promoting (blue) and neutral residues (grey). **f.** Variable length gapped or un-gapped motif analysis of periodic blocks and charged blocks from Lyons et. al, represented as PWM plot. Note that no motif could be found.

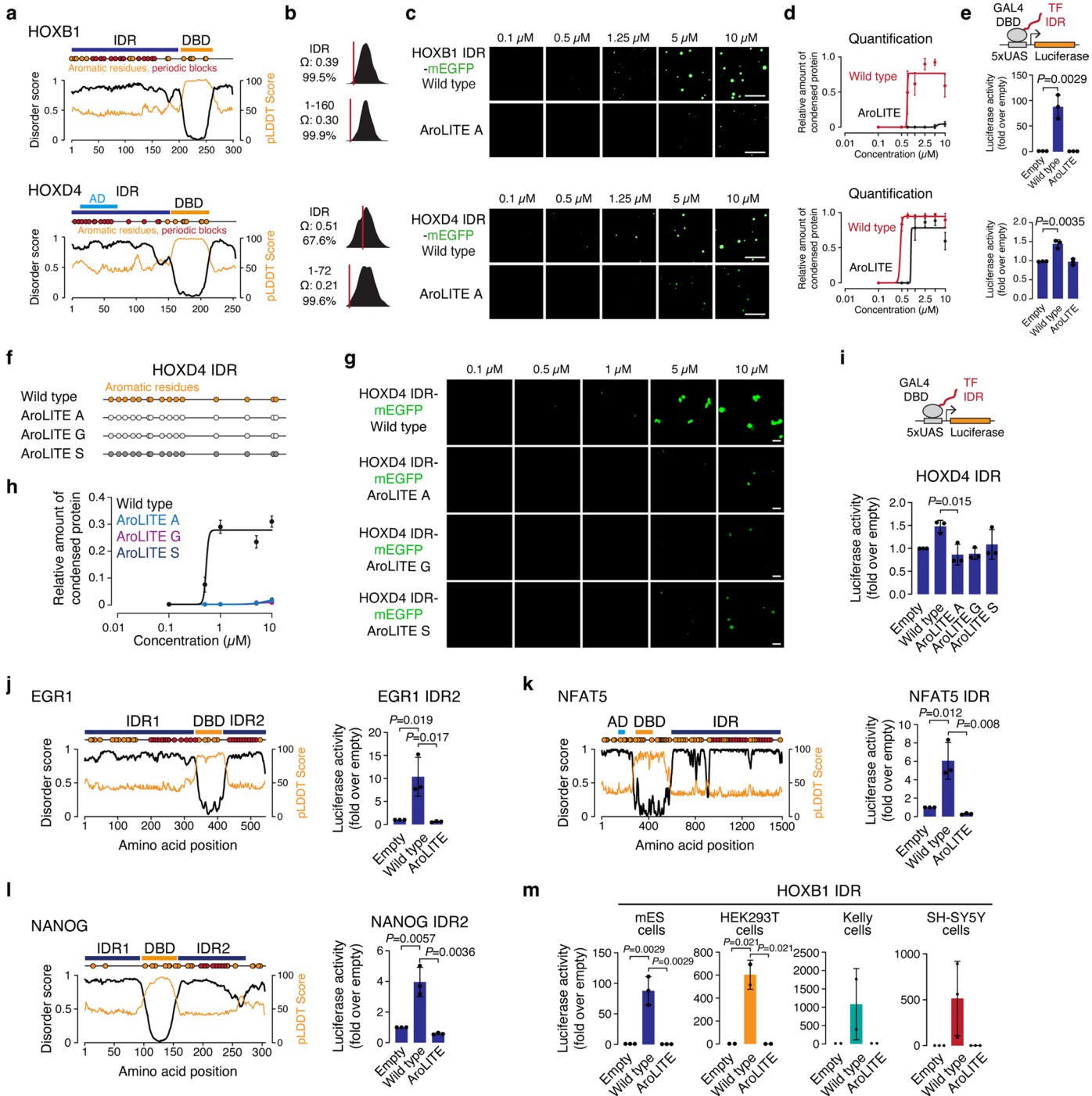

**Extended Data Fig. 2 | Aromatic residues in periodic TF IDRs are necessary for in vitro phase separation and transactivation. a.** Disorder plots (Metapredict) of HOXB1 and HOXD4 in black, AlphaFold2 pLDDT score plots in yellow. Predicted activation domains are annotated with light blue. **b.** Omega plots of HOXB1 and HOXD4 for full IDR regions (top) and portions encoding periodic aromatic blocks (bottom). Shown are the coordinates of the regions, $\Omega_{Aro}$ scores and the percentage of randomly generated sequences that have a lower $\Omega_{Aro}$ score than the actual sequence. **c.** Representative images of droplet formation of purified, recombinant TF IDR−mEGFP proteins. Scale bar: 5 μm. **d.** The relative amount of condensed protein per concentration quantified in the droplet formation assays. Data are displayed as mean ± SD. N = 10 images per condition pooled from two independent replicates. **e.** Schematic and results of luciferase reporter assays. **f.** Schematic model of HOXD4 IDRs. **g.** Representative images of droplet formation of purified HOXD4 IDR−mEGFP proteins. Scale bar: 5 μm. **h.** The relative amount

of condensed protein per concentration quantified in the droplet formation assays. Data are displayed as mean ± SD. N = 10 images per condition pooled from two independent replicates. **i.** Schematic and results of luciferase reporter assays. **j.** (left) Disorder plot (Metapredict) in black and AlphaFold2 pLDDT score plots in yellow for EGR1. (right) Results of luciferase reporter assays of the EGR1 C-IDR. **k.** (left) Disorder plot for NFAT5. (right) Results of luciferase reporter assays. **l.** (left) Disorder plot for NANOG. (right) Results of luciferase reporter assays. **m.** Results of luciferase reporter assays in the indicated cell types. In **e.**, **i.**, **j.**, **k.**, **l.**, **m.** the luciferase values were normalized against an internal *Renilla* control, and the values are displayed as percentages normalized to the activity measured using an empty vector. Data are displayed as mean ± SD. Data are from three biological replicates. *P* values are from two-sided unpaired t-tests. In **d.**, **h.** the curves were generated as a nonlinear regression to a sigmoidal curve function. IDR: intrinsically disordered region, DBD: DNA-binding domain.

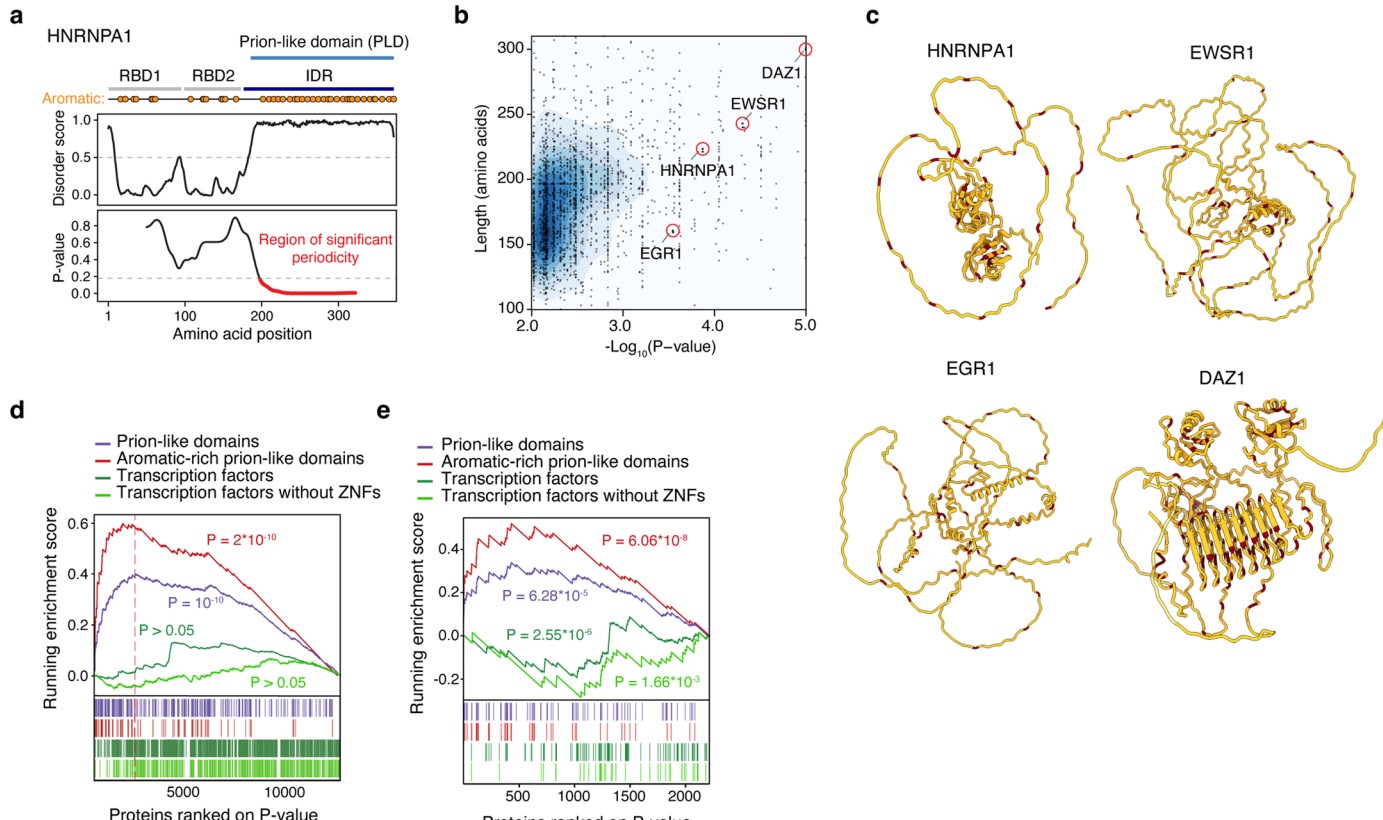

**Extended Data Fig. 3 | Proteins that contain regions with significant periodicity. a**. Region of significant periodicity in HNRNPA1. Plotted is the disorder score (Metapredict) on the top, and the *P* values (from K–S test) of the periodicity algorithm on the bottom against the position of amino acids. The positions of the two RNA binding domains (RBD1, RBD2) are noted as grey boxes. The position of the intrinsically disordered region (IDR) is noted with a dark blue bar. The position of the prion-like domain (PLD) is noted with a light blue bar. **b**. Density plot of all proteins that contain a region of significant periodicity. For each region of significant periodicity, the length of the region is plotted against the lowest *P* value (from K–S test) within the region. A *P* value cutoff of 0.01 was used to identify 2,202 regions. Each black dot represents one region, and the depth of the colour of the cloud is proportional to the density of the dots in the area. The positions of the DAZ1, EWSR1, HNRNPA1 and EGR1 are highlighted

with red circles. **c**. AlphaFold models of four proteins. Aromatic residues are coloured in red, and all other residues are coloured in yellow. Note that in DAZ1, the periodic aromatic residues are in a structure of beta-sheets. EGR1 is the transcription factor with the highest ranked region of significant periodicity. **d, e**. Gene set enrichment analysis (GSEA) of the 2,202 human proteins that contain a region with significant periodicity. The GSEA revealed an enrichment of prion-like domains and depletion of transcription factors. The 2,202 proteins were ranked according to the lowest *P* value of their most periodic 100 amino acid window. The tick marks indicate the position of prion-like domains, aromatic rich prion-like domains (>10% aromatic content) and transcription factors on the ranked gene list. Since Zn-finger transcription factors (ZNFs) contain repetitive sequences, the transcription factors excluding ZNFs is also shown. Empirical *P* value is reported.

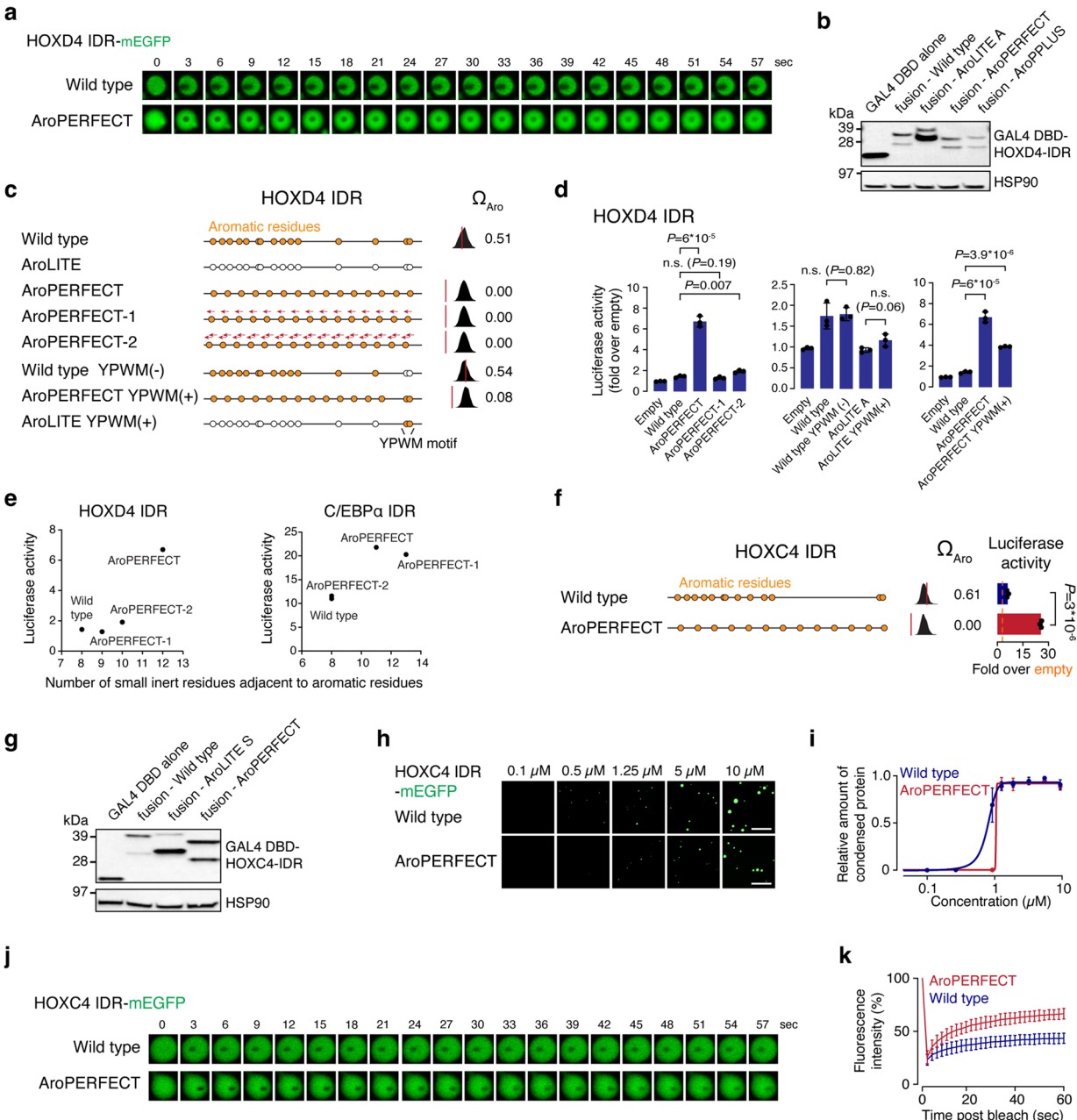

**Extended Data Fig. 4 | Characterization of periodic TF IDR mutants.**
**a**. Representative images of fluorescence recovery after photobleaching (FRAP) experiments with HOXD4 IDR−mEGFP droplets. **b**. Western blot of GAL4-DBD and GAL4-DBD-HOXD4-IDR-fusion proteins in HEK293T cells 24 hours after transfection using a GAL4-DBD specific antibody. HSP90: loading control. Except for AroLITE A, GAL4-DBD-HOXD4-IDR fusion proteins are expressed at comparable levels. **c**. Schematic models of HOXD4 wild type and mutant IDRs. Omega plots of the HOXD4 IDRs and $\Omega_{Aro}$ scores are shown next to the schematic models. **d**. Results of luciferase reporter assays. The YPWM motif does not contribute to the transactivation potential of the HOXD4 IDR. **e**. The activity of HOXD4 IDRs (left) and C/EBPα IDRs (right) scales with the number of small inert residues adjacent to aromatic residues in the IDR constructs. **f**. (left) Schematic models of wild type and AroPERFECT HOXC4 IDRs. (middle) Omega plots and $\Omega_{Aro}$ scores of the IDRs. IDR: intrinsically disordered region (right). Results of luciferase reporter assays. **g**. Western blot of GAL4-DBD and GAL4-DBD-HOXC4-IDR fusion proteins in HEK293T cells 24 hours after transfection using a GAL4-DBD specific antibody. HSP90: loading control. **h**. Representative images of droplet formation of purified HOXC4 IDR−mEGFP proteins. Scale bar: 5 µm. For the wild type IDR, the exact same images are displayed in Fig. 1g. **i**. The relative amount of condensed protein per concentration quantified in the droplet formation assays. Data are displayed as mean ± SD. N = 10 images per condition pooled from two independent replicates. The curve was generated as a nonlinear regression to a sigmoidal curve function. **j**. Representative images FRAP experiments with HOXC4 IDR−mEGFP droplets. **k**. Fluorescence intensity of HOXC4 wild type IDR and HOXC4 AroPERFECT IDR in vitro droplets before, during and after photobleaching. Data displayed as mean ± SD. N = 20 images from two replicates. In **d**., **f**. luciferase values were normalized against an internal *Renilla* control, and the values are displayed as percentages normalized to the activity measured using an empty vector. Data are displayed as mean ± SD from three biological replicates. *P* values are from two-sided unpaired t-tests.

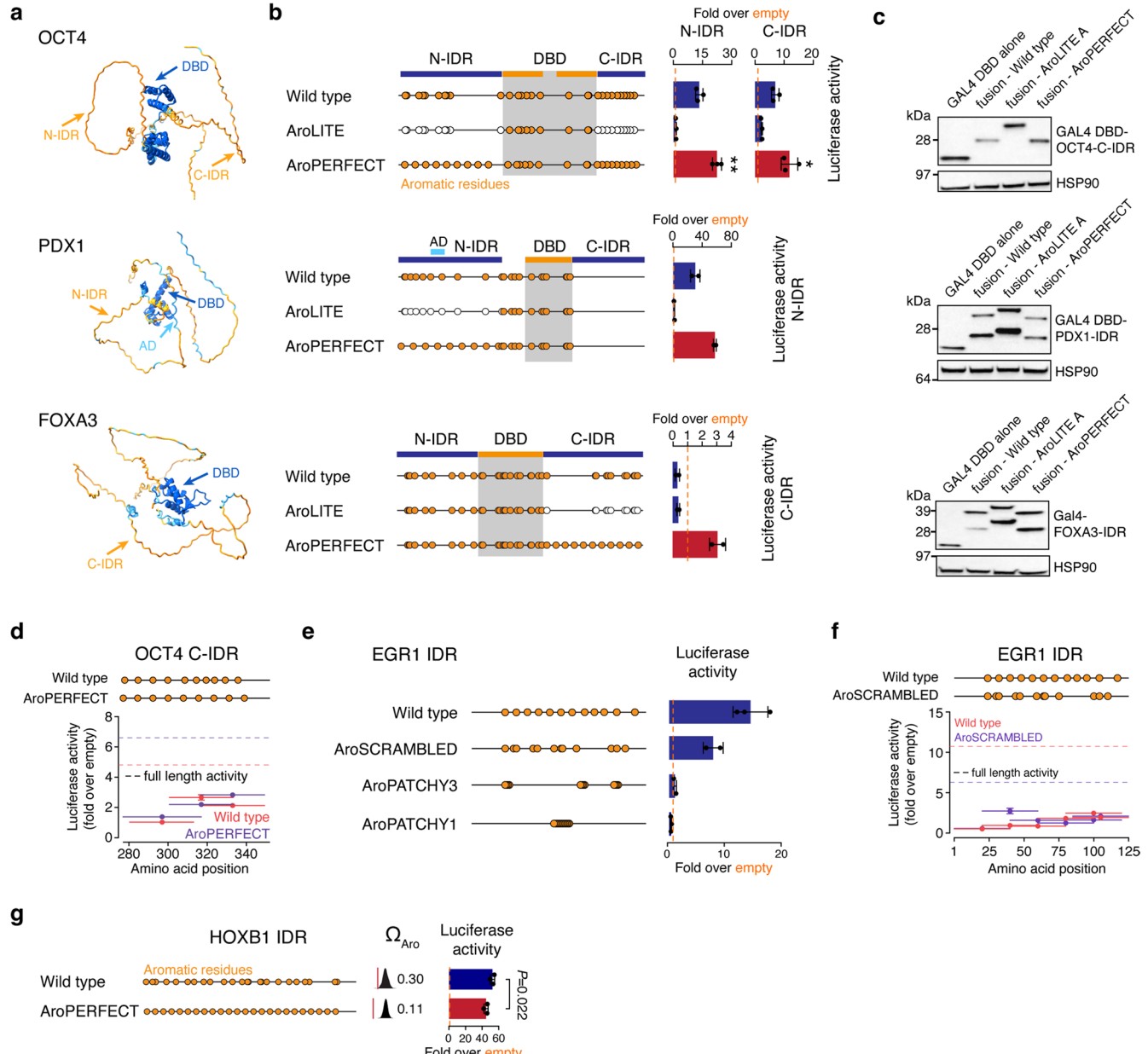

**Extended Data Fig. 5 | Optimizing aromatic dispersion enhances the activity of multiple TF IDRs. a**. AlphaFold models of OCT4, PDX1 and FOXA3. **b**. (left) Schematic models of OCT4 (top), PDX1 (middle) and FOXA3 (bottom) wild type and mutant sequences. (right) Results of luciferase reporter assays. Note that shown AroPERFECT IDRs have stronger transactivation capacity as their respective wild type sequences. **c**. Western blot of GAL4-DBD and GAL4-DBD-OCT4-IDR- (top), GAL4-DBD-PDX1-IDR- (middle) and GAL4-DBD-FOXA3-IDR-(bottom) fusion proteins in HEK293T cells 24 hours after transfection using a GAL4-DBD specific antibody. HSP90: loading control. Wild type and AroPERFECT mutants are expressed at comparable levels. **d**. Results of a OCT4 C-IDR tiling experiment by using luciferase reporter assays. Sequences were tiled into fragments of 40 amino acids with 20 amino acid overlaps. The activities of the full-length IDRs are indicated with dashed horizontal lines. **e**. (left) Schematic model of EGR1 IDR wild type and mutant sequences. Aromatic amino acids are highlighted as orange dots. (right) Results of luciferase reporter assays. **f**. Results of a EGR1 IDR tiling experiment by using luciferase reporter assays. Sequences were tiled into fragments of 40 amino acids with 20 amino acid overlaps. The activities of the full-length IDRs are indicated with dashed horizontal lines. **g**. (left) Schematic model of HOXB1 IDR wild type and AroPERFECT sequences. Aromatic amino acids are highlighted as orange dots. (middle) Omega plots and $\Omega_{Aro}$ scores of the IDRs. (right) Results of luciferase reporter assays. In **b**., **e**., **g**. luciferase values were normalized against an internal *Renilla* control, and the values are displayed as percentages normalized to the activity measured using an empty vector. Data are displayed as mean ± SD. N = 3 for OCT4, N = 2 for FOXA3 and N = 2 for PDX1 from independent replicates. *P*-values are from two-sided unpaired t-tests. *: $P < 0.05$, ***: $P < 10^{-3}$. DBD: DNA-binding domain; IDR: intrinsically disordered region; AD: activation domain.

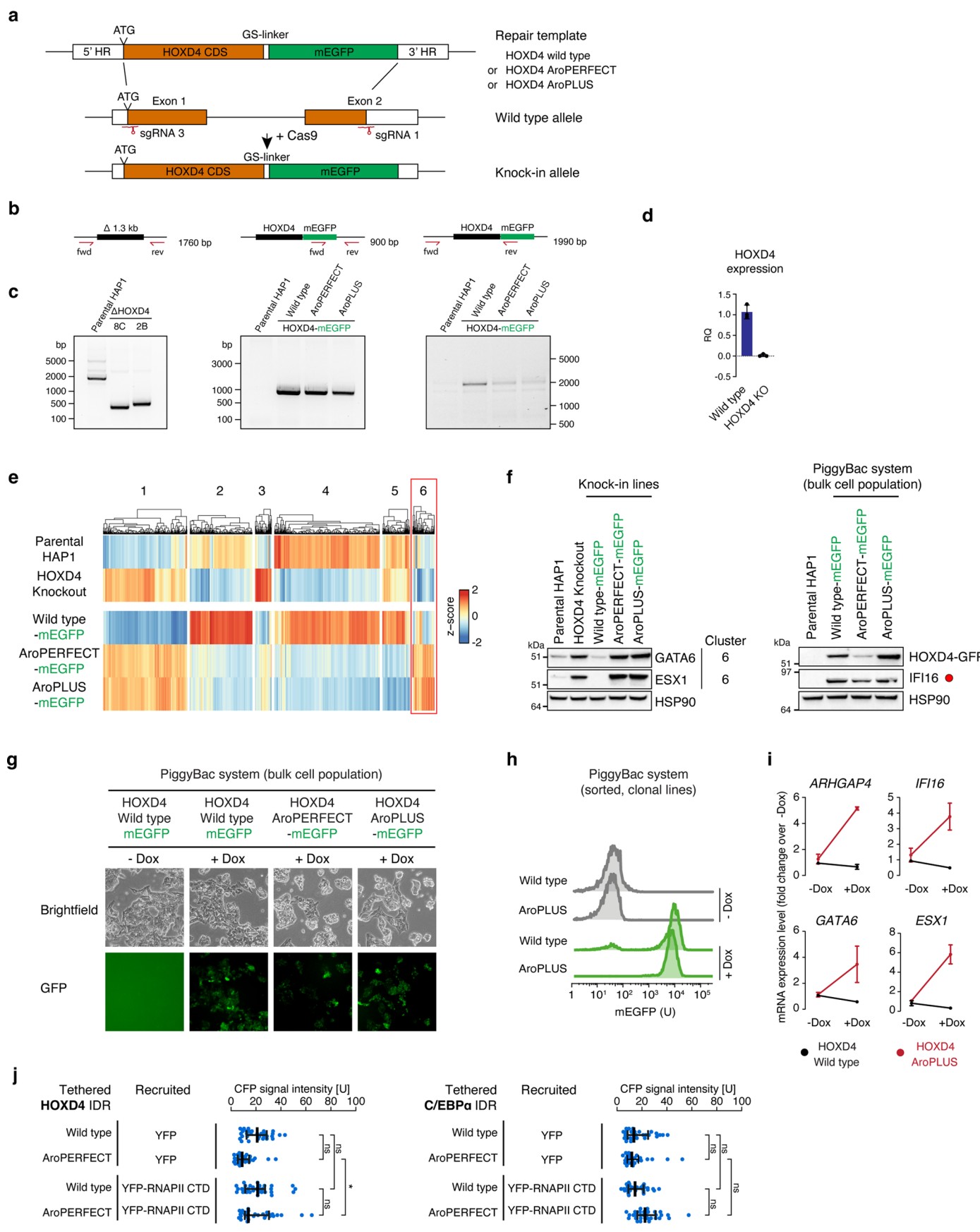

**Extended Data Fig. 6 | See next page for caption.**

**Extended Data Fig. 6 | Characterization of HAP1 HOXD4 knock-in and HOXD4 overexpression cells. a**. Scheme of mEGFP knock-in strategy at the *HOXD4* locus. **b**. Scheme of the PCR genotyping strategy of the HAP1 cell lines. **c**. PCR genotyping of HAP1 cell lines. **d**. *HOXD4* gene expression levels quantified as RQ value in HAP1 wild type and HAP1 HOXD4 knockout cells by quantitative real-time PCR. Data represented as mean ± SD from three technical replicates. **e**. Heatmap analysis of RNA-Seq data in the five cell lines. Cluster 1: Upregulated in knockout and AroPERFECT/AroPLUS. Cluster 2 and 4: downregulated in knockout and AroPERFECT/AroPLUS. Note that Cluster 4 is enriched in PBX targets. Cluster 3: expressed in knockout with minimal upregulation in AroPERFECT/AroPLUS (largely similar to Cluster 1). Cluster 5: slight reduction in knockout, more pronounced repression in AroPERFECT and AroPLUS. Clusters 1–5 comprise genes that respond similarly in the knockout, AroPERFECT and AroPLUS compared to wild type cells. Cluster 6: HOXD4-targets (that is, downregulated in the knockout compared to wild type) that are upregulated in AroPERFECT AroPLUS cells. Genes in this cluster are consistent with a partial gain-of-function effect of AroPERFECT AroPLUS HOXD4. Expression values are represented by scaling and centering VST transformed read count normalized values (z-score). K-means clustering was used to define the clusters. **f**. Western blot analysis in

the indicated HAP1 cell lines (left), and bulk cell populations encoding the PiggyBac overexpression system (right). HSP90: loading control. **g**. (top) Differential interference contrast microscopy of the indicated cell lines. Scale bar is 0.4 mm. (bottom) Fluorescence microscopy images. Cells were imaged 14 days after constant doxycycline induction. **h**. Flow cytometry analysis of mEGFP expression in HAP1 HOXD4–mEGFP PiggyBac cell lines after 14 days of Dox induction. A representative quantification is shown. Data normalized to mode. **i**. Gene expression levels quantified as fold change in HAP1 PiggyBac clones, measured by quantitative real-time PCR after 14 days of constant doxycycline induction. Data represented as mean ± SD from two biological replicates. **j**. Control quantification of CFP fluorescence intensity in the tethered foci from the experiments shown in Figs. 3h and 4e. Data displayed as mean ± SD. (left) For YFP, N = 50 and 51 nuclei for WT and AroPERFECT, respectively, and for YFP–RNAPII CTD, N = 50 and 53 nuclei for WT and AroPERFECT respectively. (right) For YFP, N = 51 and 51 nuclei for WT and AroPERFECT respectively, and for YFP–RNAPII CTD, N = 51 and 56 nuclei for WT and AroPERFECT respectively. All pooled from two independent replicates. *P* values are from 2-way ANOVA multiple comparisons tests. Exact *P* values reported in 'Statistics and Reproducibility'. *:$P < 0.05$.

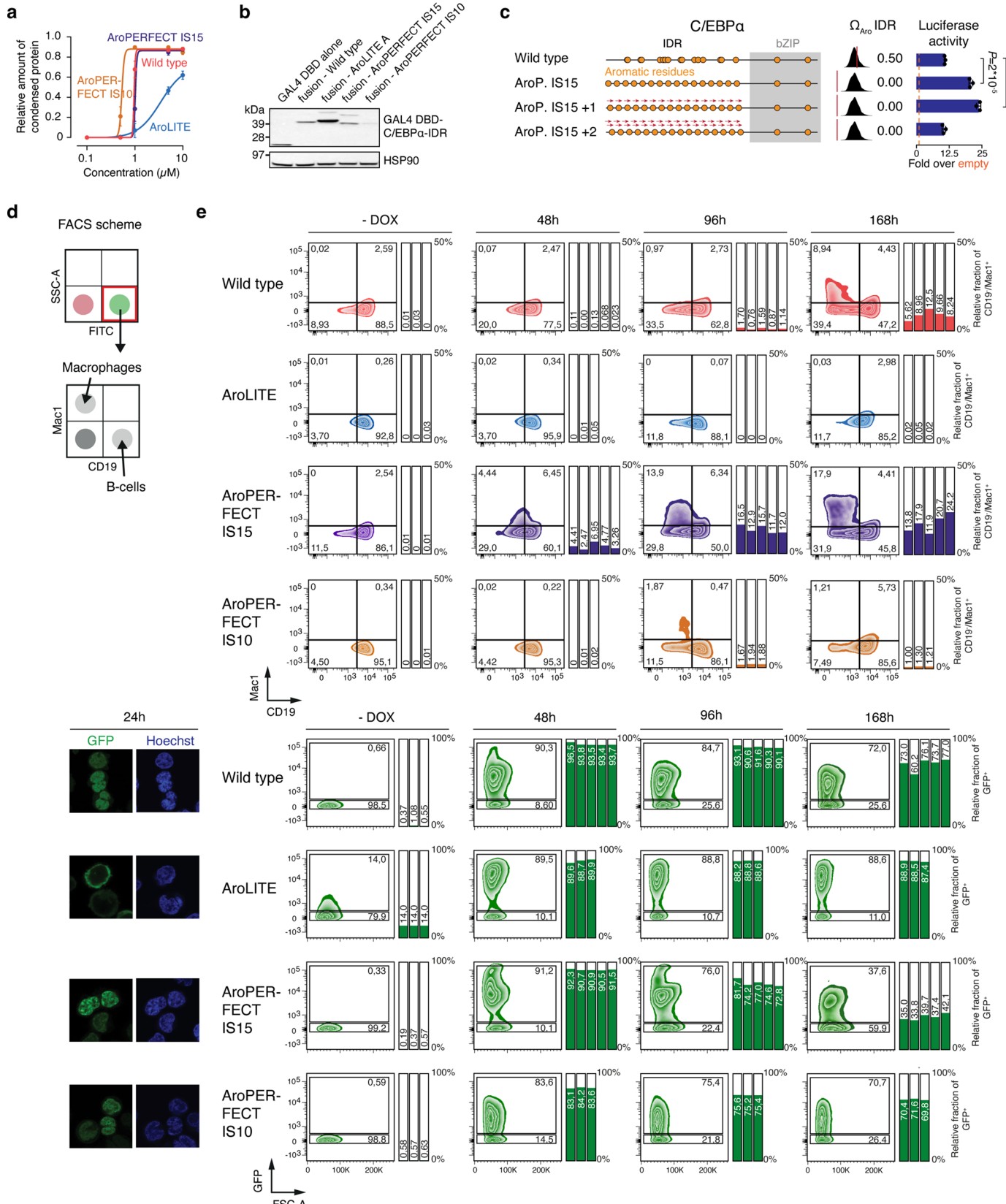

**Extended Data Fig. 7 | See next page for caption.**

**Extended Data Fig. 7 | C/EBPα supporting data. a**. The relative amount of condensed protein per concentration quantified in the droplet formation assays. Data are displayed as mean ± SD. N = 10 images from 2 replicates. The curve was generated as a nonlinear regression to a sigmoidal curve function. **b**. Western blot of GAL4-DBD and GAL4-DBD-C/EBPα-IDR fusion proteins in HEK293T cells 24 hours after transfection using a GAL4-DBD specific antibody. HSP90 is shown as loading control. Wild type and AroPERFECT IS15 mutants are expressed at comparable levels. **c**. (left) Schematic models of wild type and mutant C/EBPα proteins. The position of the bZIP DNA-binding domain is highlighted with a grey box and aromatic amino acids are highlighted as orange dots. (middle) Omega plots and $\Omega_{Aro}$ scores in the IDR. IDR: intrinsically disordered region. (right) Results of luciferase reporter assays in V6.5 mouse embryonic stem cells. Luciferase values were normalized against an internal *Renilla* control, and the values are displayed as percentages normalized to the activity measured using an empty vector (dashed orange line). Data are displayed as mean ± SD from three biological replicates per condition. *P* values are from two-sided unpaired t-tests. **d**. Scheme of FACS analysis strategy for quantification of macrophage differentiation efficiency. **e**. Flow cytometry analysis of Mac1 and CD19 expression in differentiating RCH-rtTA cells after induction of C/EBPα constructs with doxycycline. The lines separating the quadrants of the plot indicate the gating strategy to categorize the population into Mac1/CD19 positive or negative. The bar plots show the percentage of Mac1[+] CD19[−] cells among the mEGFP[+] cell population in every replicate that corresponds to each condition. Concatenated data is shown (top sub-panel). Flow cytometry analysis of mEGFP expression in differentiating RCH-rtTA cells. Gates indicate cell populations considered as mEGFP[+] or mEGFP[−]. The bar plots on the right depict the percentage of the mEGFP[+] cell population in every replicate that correspond to each condition. Concatenated data is shown (bottom sub-panel). In the bottom sub-panel, Fluorescence microscopy images of differentiating RCH-rtTA cells expressing GFP-tagged C/EBPα proteins are displayed 24 h after transgene induction. Scale bar is 10 μm. Replicates are shown on the plot.

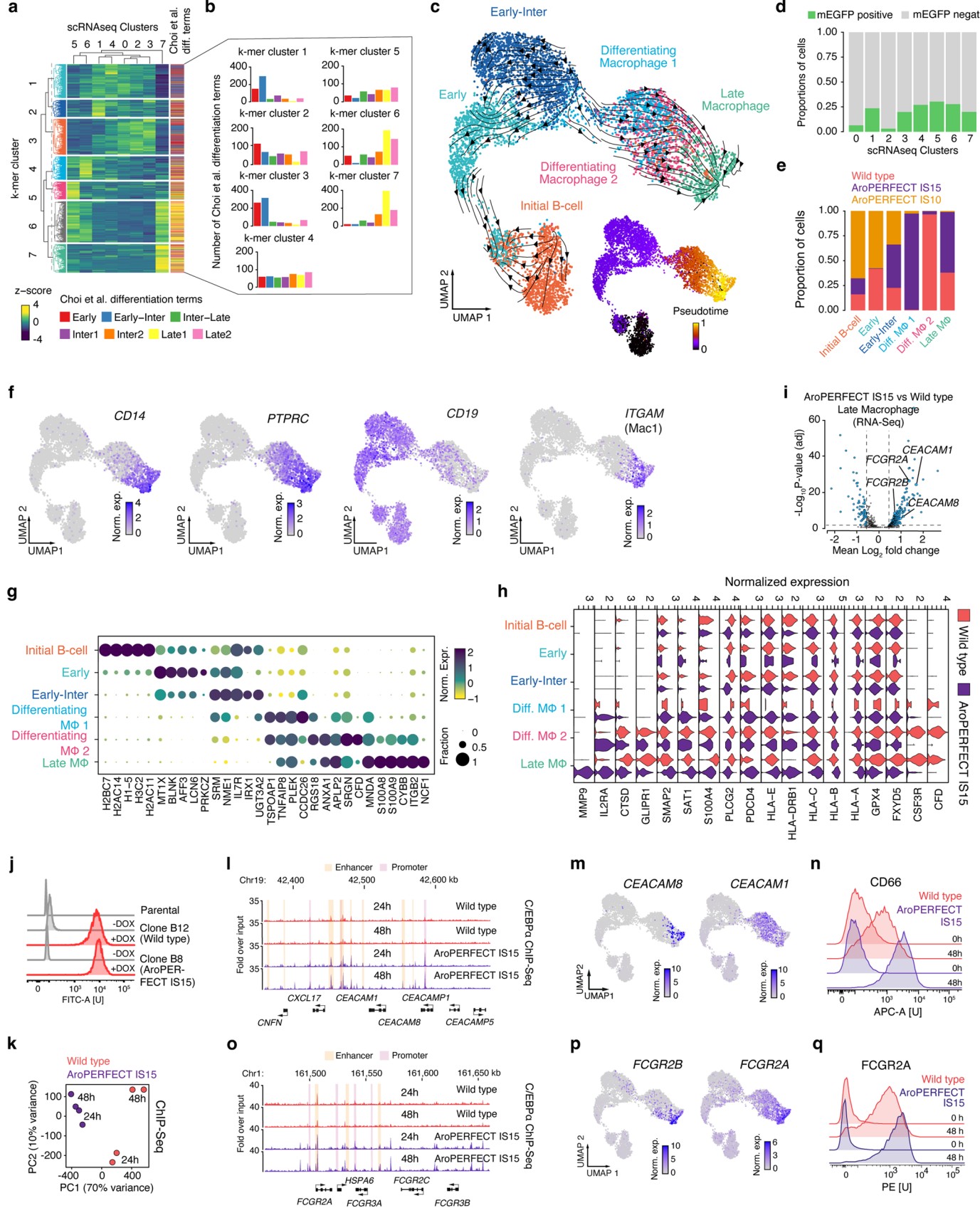

**Extended Data Fig. 8 | See next page for caption.**

**Extended Data Fig. 8 | C/EBPα single-cell RNA-seq supporting data.**
**a**. Characterization of scRNA-seq clusters using the data for various stages of B cell macrophage differentiation from a previous study. Average expression for each cluster was normalized by vst and centered (z-score). K-means clustering was used to define the heatmap clusters. **b**. Quantification of the cluster's genes for each k-cluster of the heatmap. Based on the quantification and expression profile of the heatmap the single-cell clusters were manually assigned. **c**. RNA velocity stream plot was embedded to pre-computed UMAP plot. The streamlines represent velocity vector field. The pseudotime plot (bottom right) illustrates the relative time relationship between the cells. **d**. Quantification of mEGFP-positive cells in the initial clusters. Cluster 0 and 2 contain virtually no mEGFP-positive cells, and were therefore removed from downstream analyses. **e**. Sample proportions for each cluster. Differentiating macrophage 1 is wild type-specific and Differentiating macrophage 2 is AroPERFECT IS15-specific. AroPERFECT IS10 cells are absent from the macrophage clusters. **f**. (left to right) Combined UMAP coloured *CD14* and *PTPRC, CD19* and *ITGAM (MAC1)* gene expression. These markers are associated with macrophage differentiation. **g**. Top 5 differentially expressed genes per cluster. These gene show specific expression signatures associated with each cluster and could be used as differentiation stage markers. **h**. Stacked violin plots for select DEG genes for Late macrophage cluster between AroPERFECT IS15 and wild type. Most genes seem to be expressed in other cluster with the exceptions of *MMP9*. *CSF3R* and *CFD* which seem to be macrophage and C/EBPα wild type specific while *IL2RA* is macrophage and C/EBPα AroPERFECT IS15 specific. **i**. Volcano plot of differentially expressed genes in the Late Macrophage cluster for wild type vs AroPERFECT IS15 samples. Differentially expressed target genes (Benjamini–Hochberg method, $P < 0.05$) are highlighted in blue. **j**. Flow cytometry analysis of GFP expression in RCH-rtTA clonal cell lines expressing GFP-tagged versions of C/EBPα. Data normalized to mode. **k**. Principal component analysis of the ChIP–Seq peak profiles for wild type and AroPERFECT IS15 C/EBPα-expressing cells 24 h and 48 h after induction of C/EBPα expression (PC1 vs. PC2). **l, n**. C/EBPα AroPERFECT IS15 shows enhanced binding at the *CEACAM* gene cluster (**l**) and at the *FCGR2A* locus (**n**). Displayed are genome browser tracks of ChIP–Seq data of C/EBPα wild type and AroPERFECT IS15 in RCH-rtTA cells, 24 and 48 hours after C/EBPα expression. Coordinates are hg38 genome assembly coordinates. **m, p**. Combined UMAP coloured on *CEACAM8* and *CEACAM1* (**m**) and *FCGR2B* and *FCGR2A* (**p**) expression. **n, q**. Flow cytometry analysis of CD66 (**n**) and FCGR2A (**q**) expression in differentiating GFP + RCH-rtTA cells 0 h and 48 h after induction of C/EBPα overexpression. Data normalized to mode.

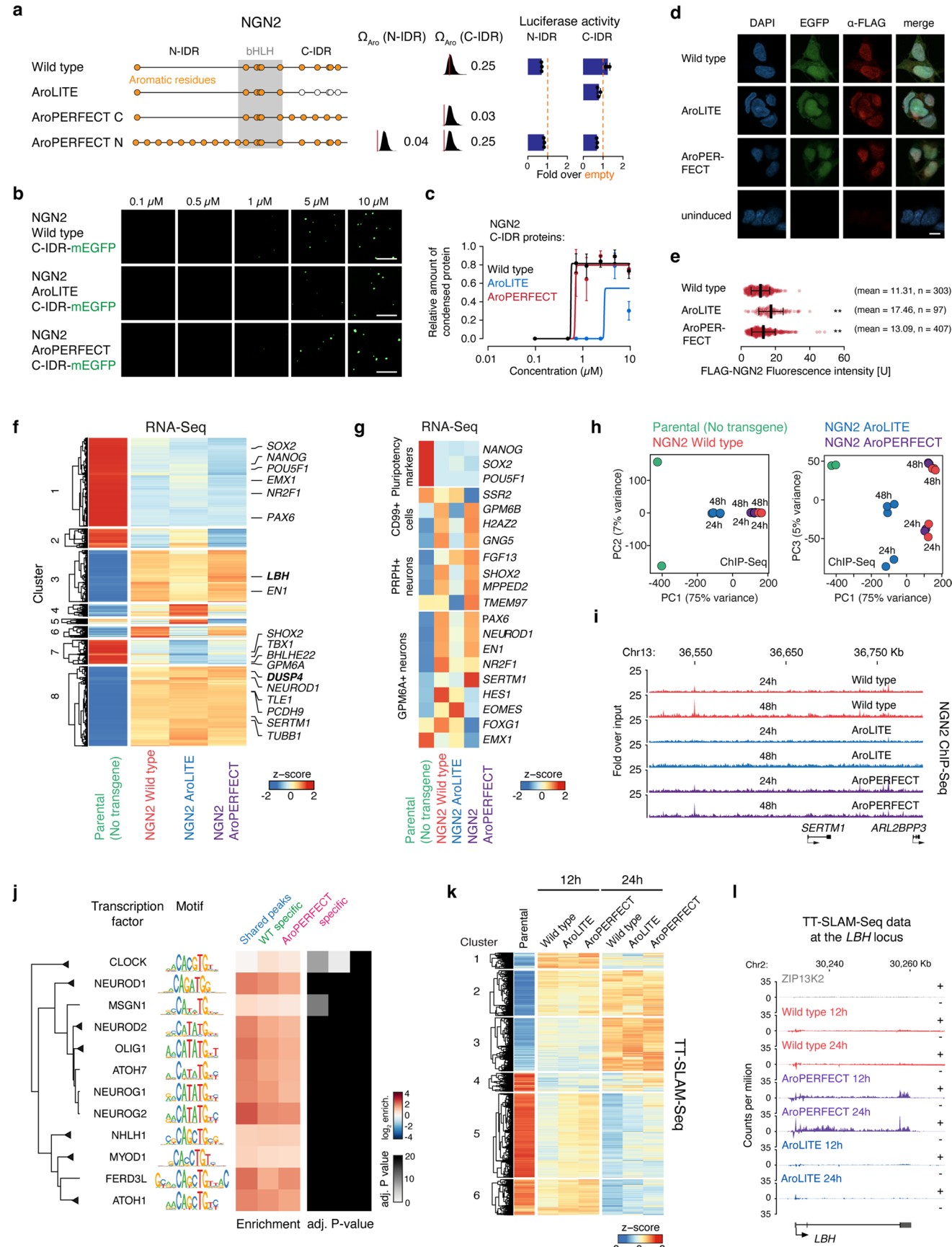

**Extended Data Fig. 9 | See next page for caption.**

**Extended Data Fig. 9 | NGN2 supporting data. a**. (left) Schematic models of NGN2 proteins. (middle) Omega plots and $\Omega_{Aro}$ scores of the IDRs. (right) Results of luciferase reporter assays. Luciferase values were normalized against an internal *Renilla* control, and the values are displayed as percentages normalized to the activity measured using an empty vector (dashed orange line). Data are displayed as mean ± SD from three biological replicates. **b**. Representative images of droplet formation of purified NGN2 C-terminal IDR−mEGFP proteins. Scale bar: 5 μm. **c**. The relative amount of condensed protein per concentration quantified in the droplet formation assays. Data are displayed as mean ± SD. N = 10 images per condition pooled from two independent replicates. The curve was generated as a nonlinear regression to a sigmoidal curve function. **d**. Fluorescence microscopy images of differentiating ZIP13K2 cells expressing FLAG-tagged versions of NGN2 at 48 h. NGN2-FLAG was visualized with an α-FLAG antibody. GFP signal is the endogenous mEGFP fluorescence signal of mEGFP. Scale bar: 5 μm. **e**. Quantification of FLAG-NGN2 signal. Data displayed as mean ± SD. N = number of cells from one biological replicate. *P* values are from two-sided unpaired t-test. $P_{(Wild\ type\ vs.\ AroLITE)}$=0.00001, $P_{(Wild\ type\ vs.\ AroPERFECT)}$=0.00019. **f**. Heatmap analysis of RNA-Seq data in the four cell lines. Genes were clustered using k-means clustering on expression values. Expression values are represented by scaling and centering VST transformed read count normalized values (z-score). **g**. Marker gene analysis from selected genes from single-cell cluster markers in NGN2 induced neural differentiation. **h**. Principal component analysis of the NGN2 ChIP−Seq peak profiles. **i**. NGN2 AroLITE loss of binding at the *SERTM1* locus. Displayed are genome browser tracks of ChIP−Seq data of NGN2 wild type, AroLITE and AroPERFECT in ZIP13K2 cells, 24 and 48 hours after NGN2 overexpression. Coordinates are hg38 genome assembly coordinates. **j**. Enrichment scores of bHLH TF motifs, and adjusted *P* values. *P* values from Benjamini−Hochberg method. **k**. Heatmap analysis of TT-SLAM-seq data in the four cell lines 12 h and 24 h after transgene induction. Genes were clustered using k-means clustering on expression values. Expression values are represented by scaling and centering VST transformed read count normalized values (z-score). **l**. TT-SLAM-Seq data at the *LBH* locus.

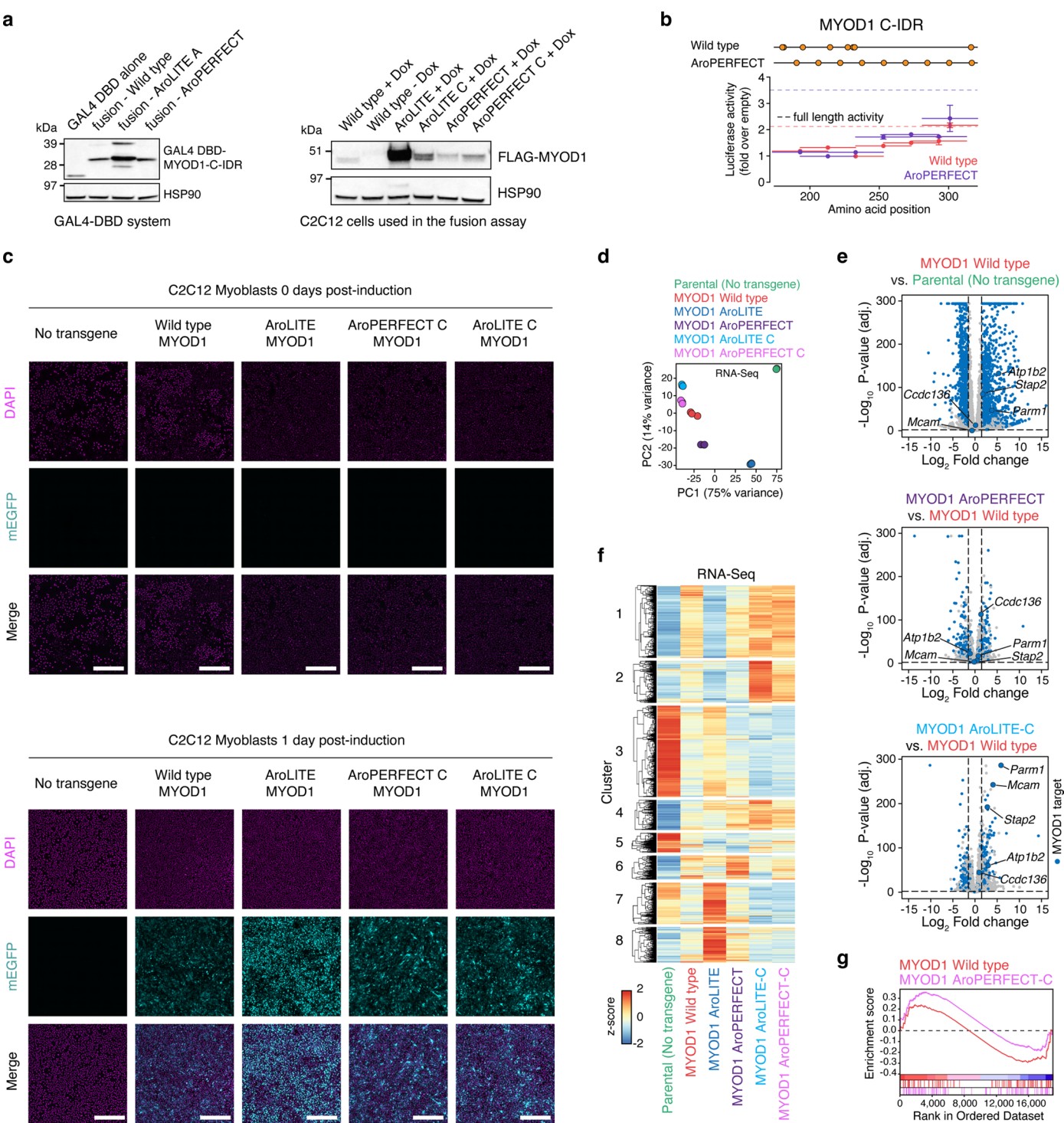

**Extended Data Fig. 10 | MYOD1 supporting data. a.** (left) Western blot of GAL4-DBD and GAL4-DBD-MYOD1 C-IDR-fusion proteins in HEK293T cells 24 hours after transfection using a GAL4-DBD specific antibody. (left). Western blot of FLAG-MYOD1 fusion proteins in differentiating C2C12 cells 24 hours after transgene induction. Wild type and AroPERFECT mutants are expressed at comparable levels. HSP90: loading control. Wild type and AroPERFECT mutants are expressed at comparable levels. **b**. Results of a MYOD1 C-IDR tiling experiment by using luciferase reporter assays. Sequences were tiled into fragments of 40 amino acids with 20 amino acid overlaps. Data displayed as mean ± SD. N = 2 biological replicates. The activities of the full-length IDRs are indicated with dashed horizontal lines. **c**. Fluorescence images of C2C12 myoblasts at day 0 and 1 after induction of MYOD1 wild type, MYOD1 AroLITE, MYOD1 AroPERFECT C or MYOD1 AroLITE C transgene with doxycycline. DAPI was used as DNA counterstain (magenta). Co-expressed mEGFP of the MYOD1-

T2A-mEGFP fusion protein was used as cytoplasmic marker (cyan). Scale bar 0.5 mm. **d**. Principal component analysis of the RNA-Seq expression profiles of Parental C2C12, C2C12 MYOD1 wild type, C2C12 MYOD1 AroLITE, C2C12 MYOD1 AroPERFECT, C2C12 MYOD1 AroLITE C and C2C12 MYOD1AroPERFECT-C cells (PC1 vs. PC2). **e**. Differential expression analysis of Parental C2C12 (top), C2C12 MYOD1 AroPERFECT (centre) and C2C12 MYOD1 AroLITE C (bottom) cells versus C2C12 MYOD1 wild type cells. MYOD1 target genes are highlighted in blue. *P*-values from Benjamini–Hochberg method. **f**. Heatmap analysis of RNA-Seq data in the six cell lines. Genes were clustered using k-means clustering on expression values. Expression values are represented by scaling and centering VST transformed read count normalized values (z-score). K-means clustering was used to define the clusters. **g**. Gene set enrichment analysis (GSEA) of differentially expressed genes in the MYOD1 AroPERFECT C RNA-Seq sample. Empirical *P* value is reported.

# Reporting Summary

## Statistics

For all statistical analyses, confirm that the following items are present in the figure legend, table legend, main text, or Methods section.

| n/a | Confirmed | |
|---|---|---|
| ☐ | ☒ | The exact sample size (*n*) for each experimental group/condition, given as a discrete number and unit of measurement |
| ☒ | ☐ | A statement on whether measurements were taken from distinct samples or whether the same sample was measured repeatedly |
| ☐ | ☒ | The statistical test(s) used AND whether they are one- or two-sided<br>*Only common tests should be described solely by name; describe more complex techniques in the Methods section.* |
| ☒ | ☐ | A description of all covariates tested |
| ☐ | ☒ | A description of any assumptions or corrections, such as tests of normality and adjustment for multiple comparisons |
| ☐ | ☒ | A full description of the statistical parameters including central tendency (e.g. means) or other basic estimates (e.g. regression coefficient) AND variation (e.g. standard deviation) or associated estimates of uncertainty (e.g. confidence intervals) |
| ☐ | ☒ | For null hypothesis testing, the test statistic (e.g. *F*, *t*, *r*) with confidence intervals, effect sizes, degrees of freedom and *P* value noted<br>*Give P values as exact values whenever suitable.* |
| ☒ | ☐ | For Bayesian analysis, information on the choice of priors and Markov chain Monte Carlo settings |
| ☒ | ☐ | For hierarchical and complex designs, identification of the appropriate level for tests and full reporting of outcomes |
| ☒ | ☐ | Estimates of effect sizes (e.g. Cohen's *d*, Pearson's *r*), indicating how they were calculated |

*Our web collection on statistics for biologists contains articles on many of the points above.*

## Software and code

Policy information about availability of computer code

| Data collection | - Fluorescence images were collected with widefield and confocal microscopes using Zen Black 2.3. software (Zeiss).<br><br>- Western blot images were collected using Image Lab software (version 6.1.0 buildt 7) (Bio-Rad).<br><br>- FACS data was collected with FACS Diva software (BD Biosciences) and BD Aria II and BD Celesta instruments. |
|---|---|
| Data analysis | Fluorescence images were analyzed using ZenBlue 3.1, 3.2, 3.4 (Zeiss), Fiji/ImageJ (2.1.0/1.53i). Data was plotted using GraphPad PRISM 9.<br><br>FACS data was analyzed using FlowJo (v10.7)<br><br>GraphPad PRISM (v9.2.0) was used for statistical analysis and barplot generation.<br><br>For single cell RNA-seq analysis, the following software were used: Cell ranger v3.1.0, Seurat 4.0.6, R v4.2., velocyto v1.0, scvelo v0.3.1, DESeq2 v1.41.0.<br><br>For Bulk RNA-seq data analysis, the following software were used: cutadapt 4.7, STAR aligner 2.7.9a, DESeq2 v1.41.0, R v4.2, GSEAPreranked v6.0.12.<br><br>For Chip-seq cutadapt 4.7, bwa v0.7.17, samtools v1.19.1, gatk v4.4, MACs v3.0b1, bamCoverage v3.5.1, MEME v5.1.1, deepTools2 v3.5.1, bamCoverage v3.5.1, deepTools2 v3.5.1, Bigwigmerge v377, bedGraphToBigWig v377, pygenometracks 3.7, DiffBind v3.6.5 bedtools v2.6.0.<br><br>For TT-SLAM-seq analysis, the following software were used: STAR aligner 2.7.9a, seqtk 1.3-r106, SLAM-DUNK v0.4.1, samtools v1.19.1, featureCounts v2.0.6, bamCoverage v3.5.1, deepTools2 v3.5.1, Bigwigmerge v377, bedGraphToBigWig v377, pygenometracks 3.7. |

Protein sequence analysis was performed with the following software: Biostrings v2.40.2, Metapredict v2, UCSF ChimeraX v1.6, Alphafold v2.0, MEME v5.1.1, PLAAC v1, localCIDER 0.1.20,

All software versions and parameters are listed in the Methods.

Custom Python and R code is available https://github.com/hniszlab/TFsubopt.

The Periodic Block finder code is made available in the link https://github.com/alexpmagalhaes/PeriodicBlock_finder.

The QuasiIDRfinder code is made available in the link https://github.com/gozdekibar/QuasiIDRFinder

Custom code available under : https://doi.org/10.5281/zenodo.10628753

For manuscripts utilizing custom algorithms or software that are central to the research but not yet described in published literature, software must be made available to editors and reviewers. We strongly encourage code deposition in a community repository (e.g. GitHub). See the Nature Portfolio guidelines for submitting code & software for further information.

## Data

Policy information about availability of data

All manuscripts must include a data availability statement. This statement should provide the following information, where applicable:
- Accession codes, unique identifiers, or web links for publicly available datasets
- A description of any restrictions on data availability
- For clinical datasets or third party data, please ensure that the statement adheres to our policy

Sequencing data was deposited at the Gene Expression Omnibus (GEO), under the accession ID: GSE201655.

RAW data was deposited at: https://owww.molgen.mpg.de/~TFsuboptimization/

For NGS experiments of human samples we used human genome hg38 and annotations from GENCODE GRCh38.p13.

For NGS experiments of mouse samples we used mouse mm10 genome and annotations from GENCODE GRCm38.p6.

For annotation of proteins and IDRs we used GENCODE gene annotation v39, GENCODE GRCh38.p13 and IDs from Ensembl v104.

Transcription factor sequences and annotations were from AnimalTFDB3.0.

# Field-specific reporting

Please select the one below that is the best fit for your research. If you are not sure, read the appropriate sections before making your selection.

☒ Life sciences        ☐ Behavioural & social sciences        ☐ Ecological, evolutionary & environmental sciences

For a reference copy of the document with all sections, see nature.com/documents/nr-reporting-summary-flat.pdf

# Life sciences study design

All studies must disclose on these points even when the disclosure is negative.

| Sample size | No statistical methods were used to predetermine sample sizes. Sample sizes are indicated in the figure panels or legends or in the Methods. For droplet experiments we imaged at least 10 independent fields of a view for each experimental condition based on current methodology in the field ( Sabari et al. Science. 2018; Boija et al. Cell. 2018). For imaging experiments multiple replicate experiments were performed indiciated in the legends, according to current standards in the field (Sabari et al., Science 2018; Boija et al. Cell 2018). For transactivation experiments, we collected luminescence data from at least four independent transfections of two biological replicates for each experimental condition. For fluorescence imaging of differentiating neurons and muscle cells, we imaged 3-5 randomly selected fields of view per biological replicate and experimental condition in three independent rounds of differentiation. For Supplemental Figure 3 we imaged two clonal lines with 3-5 randomly selected fields of view and displayed them separately to highlight possible clonal heterogeneity. |
|---|---|
| Data exclusions | In rare instances, out of focus images or images with the majority of cells washed off the slide by mechanical force (e.g. pipetting) were excluded in differentiation experiments. |
| Replication | For droplet experiments we imaged at least 3-5 fields of view in at least 2 independent replicate series. For transactivation experiments, we collected luminescence data from at least four independent transfections of two biological replicates for each experimental condition. For fluorescence imaging of differentiating neurons and muscle cells, we imaged 3-5 randomly selected fields of view per biological replicate and experimental condition in three independent rounds of differentiation. For Supplemental Data 3 we imaged two clonal lines with 3-5 randomly selected fields of view and displayed them separately to highlight possible clonal heterogeneity. For this experiment, replication attempts were successful for 3/5 cases, two attempts failed on technical reasons. For all experiments attempts at replication were generally successful, unless technical issues arose. The number of replicates are reported in figures and legends. |

| Randomization | Not relevant for the study. |
| Blinding | Blinding was not relevant for the experiments. |

# Reporting for specific materials, systems and methods

We require information from authors about some types of materials, experimental systems and methods used in many studies. Here, indicate whether each material, system or method listed is relevant to your study. If you are not sure if a list item applies to your research, read the appropriate section before selecting a response.

### Materials & experimental systems

| n/a | Involved in the study |
|---|---|
| ☐ | ☒ Antibodies |
| ☐ | ☒ Eukaryotic cell lines |
| ☒ | ☐ Palaeontology and archaeology |
| ☒ | ☐ Animals and other organisms |
| ☒ | ☐ Human research participants |
| ☒ | ☐ Clinical data |
| ☒ | ☐ Dual use research of concern |

### Methods

| n/a | Involved in the study |
|---|---|
| ☐ | ☒ ChIP-seq |
| ☐ | ☒ Flow cytometry |
| ☒ | ☐ MRI-based neuroimaging |

## Antibodies

| Antibodies used | Immunofluorescence experiments:<br><br>GFP (A11122, 1:500, Invitrogen)<br>FLAG (F1804, 1:500, Sigma-Aldrich)<br>Donkey anti-Rabbit-Alexa647 (711-605-152, 1:1000,JacksonImmuno)<br>Donkey anti-Mouse-Alexa647 (715-605-150, 1:1000,JacksonImmuno)<br><br>Western blotting experiments:<br><br>HSP90 (BD610419; 1:4000, BD)<br>IFI16 (sc-8023, 1:200, Santa Cruz)<br>GFP (A11122, 1:2000, Invitrogen)<br>ARHGAP4 (sc-376251, 1:200, Santa Cruz)<br>ESX1 (sc-365740, 1:200, Santa Cruz)<br>FLAG (F1804, 1:2000, Sigma-Aldrich)<br>GATA6 (AF1700, 1:1000, RnD)<br>Peroxidase-AffiniPure Donkey Anti-Goat IgG (705-035-147, 1:5000, JacksonImmuno)<br>Peroxidase IgG Fraction Monoclonal Mouse Anti-Rabbit IgG (211-032-171, 1:5000, JacksonImmuno)<br>Peroxidase AffiniPure Goat Anti-Mouse IgG (115-035-174, 1:1000, JacksonImmuno)<br>Gal4 (sc-510, 1:200, Santa Cruz)<br><br>FACS experiments:<br><br>CD19 APC-Cy7 Mouse anti-Human CD19 (557791, 1:200, BD)<br>APC Mouse Anti-Human CD11b/Mac-1 (550019, 1:200, BD)<br>CD66a Alexa Fluor 647 anti-human CD66a (398905, 1:250, BioLegend)<br>FCGR2A PE anti-human FCGR2A (305503, 1:200, BioLegend)<br><br>ChIP-seq experiments:<br><br>GFP clone 3E6 (A-11120, 1:500, Invitrogen)<br>FLAG (F1804, 1:250, Sigma-Aldrich) |
| Validation | Antibodies in Immunofluorescence, ChIP-seq and Western blot experiments were validated by comparing to parental cell lines without transgene expression.<br><br>All antibodies are validated by the provider and cited in numerous publications:<br><br>Immunofluorescence and Western blot experiments:<br><br>GFP (A11122, Invitrogen) – rabbit<br>https://www.thermofisher.com/antibody/product/A-11122.html?CID=AFLCA-A-11122<br><br>FLAG (F1804, Sigma-Aldrich) – mouse<br>https://www.sigmaaldrich.com/deepweb/assets/sigmaaldrich/product/documents/119/160/f1804bul-mk.pdf<br><br>HSP90 (BD610419, BD) – mouse |

https://wwwfishersci.com/shop/products/anti-hsp90-clone-68-bd-2/BDB610419

IFI16 (sc-8023, Santa Cruz) – mouse
https://datasheets.scbt.com/sc-8023.pdf

ARHGAP4 (sc-376251, Santa Cruz) – mouse
https://datasheets.scbt.com/sc-376251.pdf

ESX1 (sc-365740, Santa Cruz) – mouse
https://datasheets.scbt.com/sc-365740.pdf

GATA6 (AF1700, RnD) – mouse
https://resources.rndsystems.com/pdfs/datasheets/af1700.pdf?v=20240206

Gal4 (sc-510, Santa Cruz) – mouse
https://datasheets.scbt.com/sc-510.pdf

FACS experiments:

CD19 APC-Cy7 Mouse anti-Human CD19 (557791, BD)
https://www.bdbiosciences.com/content/bdb/paths/generate-tds-document.us.557791.pdf

APC Mouse Anti-Human CD11b/Mac-1 (550019, BD)
https://www.bdbiosciences.com/content/bdb/paths/generate-tds-document.de.550019.pdf

CD66a Alexa Fluor 647 anti-human CD66a (398905, BioLegend)
https://www.biolegend.com/en-us/products/alexa-fluor-647-anti-human-cd66a-b-c-e-antibody-20073

FCGR2A PE anti-human FCGR2A (305503, BioLegend)
https://www.biolegend.com/de-at/products/pe-anti-human-fcgr2a-cd32a-antibody-21510?GroupID=GROUP28

ChiP-seq experiments:

GFP clone 3E6 (A-11120, Invitrogen)
https://www.thermofisher.com/antibody/product/GFP-Antibody-clone-3E6-Monoclonal/A-11120

# Eukaryotic cell lines

Policy information about cell lines

| Cell line source(s) | General information provided in methods under: "Cell culture"

- V6.5 mouse embryonic stem cells (mESCs), source: Konrad Hochedlinger lab

- HEK293T, source: ATCC, Identifier: CRL-3216

-SH-SY5Y, source: DSMZ, Identifier: ACC-209

-Kelly, source: DSMZ, Identifier: ACC-355

-HAP1, source: Aktas Lab (MPI-MG)

-HAP1-HOXD4-mEGFP lines, source: This paper (see Methods "Generation of HOXD4 GFP knock-in and knockout lines" and "Generation of Doxycycline-inducible HOXD4 overexpression lines in HAP1 cells")

-ZIP13K2, source: Müller Lab (MPI-MG)

-ZiP13K2-NGN2-T2A-mEGFP lines, source: This paper (see Methods "Generation of Doxycycline-inducible NGN2 overexpression systems in human iPS cells")

-C2C12, source: Stricker Lab (Freie Universität Berlin)

-C2C12-MYOD1-T2A-mEGFP lines, source: This paper (see Methods "Generation of Doxycycline-inducible MYOD1 overexpression lines in C2C12 cells")

-RCH-rtTA, source: Graf Lab (CRG Barcelona)

-RCH-rtTA-CEBPa-GFP lines, source: This paper (see Methods "Generation of Doxycycline-inducible C/EBPα overexpression lines in RCH cells")

-U2OS, source: Kinkley Lab (MPI-MG) |

| Authentication | The identity of HEK293T, SH-SY5Y, Kelly, HAP1, RCH-rtTA, U2Os, C2C12,parental V6.5 mESCs and ZIP13K2 iPSCs, and all cell |

| Authentication | lines derived from them has been validated using |
|---|---|
| | morphological characteristics, qPCRs, FACS, immunofluorescence, RNA-seq, and marker gene expression (where applicable) but have not been authenticated. |
| | Expression of HOXD4, NGN2, MYOD1, CEBPa transgene expression was validated by FACS and RNA-sequencing. |

| Mycoplasma contamination | All cell lines tested negative for mycoplasma contamination. |
|---|---|

| Commonly misidentified lines (See ICLAC register) | None used. |
|---|---|

# ChIP-seq

## Data deposition

☒ Confirm that both raw and final processed data have been deposited in a public database such as GEO.

☐ Confirm that you have deposited or provided access to graph files (e.g. BED files) for the called peaks.

| Data access links *May remain private before publication.* | Sequencing data was deposited at the Gene Expression Omnibus (GEO), under the accession ID: GSE201655 |
|---|---|

| Files in database submission | Processed Files |
|---|---|

GSM6069079_DH-RNA-043barcodes.tsv.gz
GSM6069079_DH-RNA-043features.tsv.gz
GSM6069079_DH-RNA-043matrix.mtx.gz
GSM6069080_DH-RNA-044barcodes.tsv.gz
GSM6069080_DH-RNA-044features.tsv.gz
GSM6069080_DH-RNA-044matrix.mtx.gz
GSM6069081_DH-RNA-045barcodes.tsv.gz
GSM6069081_DH-RNA-045features.tsv.gz
GSM6069081_DH-RNA-045matrix.mtx.gz
GSM6710791_DH-RNA-065_hg38.star.ReadsPerGene.out.tab.gz
GSM6710792_DH-RNA-066_hg38.star.ReadsPerGene.out.tab.gz
GSM6710793_DH-RNA-067_hg38.star.ReadsPerGene.out.tab.gz
GSM6710794_DH-RNA-068_hg38.star.ReadsPerGene.out.tab.gz
GSM6710795_DH-RNA-069_hg38.star.ReadsPerGene.out.tab.gz
GSM6710796_DH-RNA-070_hg38.star.ReadsPerGene.out.tab.gz
GSM6710797_DH-RNA-071_hg38.star.ReadsPerGene.out.tab.gz
GSM6710798_DH-RNA-072_hg38.star.ReadsPerGene.out.tab.gz
GSM6710799_DH-RNA-073_hg38.star.ReadsPerGene.out.tab.gz
GSM6710800_DH-RNA-074_hg38.star.ReadsPerGene.out.tab.gz
GSM6710801_DH-RNA-075_hg38.star.ReadsPerGene.out.tab.gz
GSM6710802_DH-RNA-076_hg38.star.ReadsPerGene.out.tab.gz
GSM6710803_DH-RNA-077_hg38.star.ReadsPerGene.out.tab.gz
GSM6710804_DH-RNA-078_hg38.star.ReadsPerGene.out.tab.gz
GSM6710805_DH-RNA-079_hg38.star.ReadsPerGene.out.tab.gz
GSM6710806_ZIP13K2_WT_1_HG38_NGN2.star.ReadsPerGene.out.tab.gz
GSM6710807_ZIP13K2_WT_2_HG38_NGN2.star.ReadsPerGene.out.tab.gz
GSM6710808_ZIP13K2_WT_3_HG38_NGN2.star.ReadsPerGene.out.tab.gz
GSM6710809_NGN2_WT_1_HG38_NGN2.star.ReadsPerGene.out.tab.gz
GSM6710810_NGN2_WT_2_HG38_NGN2.star.ReadsPerGene.out.tab.gz
GSM6710811_NGN2_WT_3_HG38_NGN2.star.ReadsPerGene.out.tab.gz
GSM6710812_NGN2_AroLITE_1_HG38_NGN2.star.ReadsPerGene.out.tab.gz
GSM6710813_NGN2_AroLITE_2_HG38_NGN2.star.ReadsPerGene.out.tab.gz
GSM6710814_NGN2_AroLITE_3_HG38_NGN2.star.ReadsPerGene.out.tab.gz
GSM6710815_NGN2_AroPERFECT_1_HG38_NGN2.star.ReadsPerGene.out.tab.gz
GSM6710816_NGN2_AroPERFECT_2_HG38_NGN2.star.ReadsPerGene.out.tab.gz
GSM6710817_NGN2_AroPERFECT_3_HG38_NGN2.star.ReadsPerGene.out.tab.gz
GSM6710818_C2C12_WT_1_mm39_myod1.star.ReadsPerGene.out.tab.gz
GSM6710819_C2C12_WT_2_mm39_myod1.star.ReadsPerGene.out.tab.gz
GSM6710820_C2C12_WT_3_mm39_myod1.star.ReadsPerGene.out.tab.gz
GSM6710821_MYOD1_AroLite_1_mm39_myod1.star.ReadsPerGene.out.tab.gz
GSM6710822_MYOD1_AroLite_2_mm39_myod1.star.ReadsPerGene.out.tab.gz
GSM6710823_MYOD1_AroLite_3_mm39_myod1.star.ReadsPerGene.out.tab.gz
GSM6710824_MYOD1_AroLiteC_1_mm39_myod1.star.ReadsPerGene.out.tab.gz
GSM6710825_MYOD1_AroLiteC_2_mm39_myod1.star.ReadsPerGene.out.tab.gz
GSM6710826_MYOD1_AroLiteC_3_mm39_myod1.star.ReadsPerGene.out.tab.gz
GSM6710827_MYOD1_AroPerfect_1_mm39_myod1.star.ReadsPerGene.out.tab.gz
GSM6710828_MYOD1_AroPerfect_2_mm39_myod1.star.ReadsPerGene.out.tab.gz
GSM6710829_MYOD1_AroPerfect_3_mm39_myod1.star.ReadsPerGene.out.tab.gz
GSM6710830_MYOD1_AroPerfectC_1_mm39_myod1.star.ReadsPerGene.out.tab.gz
GSM6710831_MYOD1_AroPerfectC_2_mm39_myod1.star.ReadsPerGene.out.tab.gz
GSM6710832_MYOD1_AroPerfectC_3_mm39_myod1.star.ReadsPerGene.out.tab.gz

GSM6710833_MYOD1_WT_1_mm39_myod1.star.ReadsPerGene.out.tab.gz
GSM6710834_MYOD1_WT_2_mm39_myod1.star.ReadsPerGene.out.tab.gz
GSM6710835_MYOD1_WT_3_mm39_myod1.star.ReadsPerGene.out.tab.gz
GSM6710836_WT_24_ChIP_rep1_peaks.narrowPeak.gz
GSM6710837_WT_24_ChIP_rep2_peaks.narrowPeak.gz
GSM6710838_WT_48_ChIP_rep1_peaks.narrowPeak.gz
GSM6710839_WT_48_ChIP_rep2_peaks.narrowPeak.gz
GSM6710840_IS15_24_ChIP_rep1_peaks.narrowPeak.gz
GSM6710841_IS15_24_ChIP_rep2_peaks.narrowPeak.gz
GSM6710842_IS15_48_ChIP_rep1_peaks.narrowPeak.gz
GSM6710843_IS15_48_ChIP_rep2_peaks.narrowPeak.gz
GSM6710852_ZIP13K2_NGN2_24h_AroLITE_2_peaks.narrowPeak.gz
GSM6710853_ZIP13K2_NGN2_24h_AroLITE_3_peaks.narrowPeak.gz
GSM6710854_ZIP13K2_NGN2_24h_AroPERFECT_2_peaks.narrowPeak.gz
GSM6710855_ZIP13K2_NGN2_24h_AroPERFECT_3_peaks.narrowPeak.gz
GSM6710856_ZIP13K2_NGN2_24h_WT_1_peaks.narrowPeak.gz
GSM6710857_ZIP13K2_NGN2_24h_WT_2_peaks.narrowPeak.gz
GSM6710858_ZIP13K2_NGN2_48h_AroLITE_1_peaks.narrowPeak.gz
GSM6710859_ZIP13K2_NGN2_48h_AroLITE_3_peaks.narrowPeak.gz
GSM6710860_ZIP13K2_NGN2_48h_AroLITE_2_peaks.narrowPeak.gz
GSM6710861_ZIP13K2_NGN2_48h_AroPERFECT_1_peaks.narrowPeak.gz
GSM6710862_ZIP13K2_NGN2_48h_AroPERFECT_3_peaks.narrowPeak.gz
GSM6710863_ZIP13K2_NGN2_48h_AroPERFECT_2_peaks.narrowPeak.gz
GSM6710864_ZIP13K2_NGN2_48h_WT_1_peaks.narrowPeak.gz
GSM6710865_ZIP13K2_NGN2_48h_WT_2_peaks.narrowPeak.gz
GSM6710866_ZIP13K2_NGN2_48h_WT_3_peaks.narrowPeak.gz

RAW Files

mpimg_L23394-1_DH-RNA-043_S1_L001_R1_001.fastq.gz
mpimg_L23394-1_DH-RNA-043_S1_L002_I1_001.fastq.gz
mpimg_L23395-1_DH-RNA-044_S2_L002_I1_001.fastq.gz
mpimg_L23394-1_DH-RNA-043_S1_L002_R1_001.fastq.gz
mpimg_L23396-1_DH-RNA-045_S3_L001_R1_001.fastq.gz
mpimg_L23394-1_DH-RNA-043_S1_L002_R2_001.fastq.gz
mpimg_L23396-1_DH-RNA-045_S3_L001_I1_001.fastq.gz
mpimg_L23395-1_DH-RNA-044_S2_L001_R1_001.fastq.gz
mpimg_L23396-1_DH-RNA-045_S3_L002_I1_001.fastq.gz
mpimg_L23394-1_DH-RNA-043_S1_L001_R2_001.fastq.gz
mpimg_L23396-1_DH-RNA-045_S3_L001_R2_001.fastq.gz
mpimg_L23395-1_DH-RNA-044_S2_L001_I1_001.fastq.gz
mpimg_L23395-1_DH-RNA-044_S2_L001_R2_001.fastq.gz
mpimg_L23394-1_DH-RNA-043_S1_L001_I1_001.fastq.gz
mpimg_L23395-1_DH-RNA-044_S2_L002_R1_001.fastq.gz
mpimg_L23395-1_DH-RNA-044_S2_L002_R2_001.fastq.gz
mpimg_L23396-1_DH-RNA-045_S3_L002_R1_001.fastq.gz
mpimg_L23396-1_DH-RNA-045_S3_L002_R2_001.fastq.gz

mpimg_L24588-1_DH-RNA-065_S65_R1_001.fastq.gz
mpimg_L24589-1_DH-RNA-066_S66_R1_001.fastq.gz
mpimg_L24590-1_DH-RNA-067_S67_R1_001.fastq.gz
mpimg_L24591-1_DH-RNA-068_S68_R1_001.fastq.gz
mpimg_L24592-1_DH-RNA-069_S69_R1_001.fastq.gz
mpimg_L24593-1_DH-RNA-070_S70_R1_001.fastq.gz
mpimg_L24594-1_DH-RNA-071_S71_R1_001.fastq.gz
mpimg_L24595-1_DH-RNA-072_S72_R1_001.fastq.gz
mpimg_L24596-1_DH-RNA-073_S73_R1_001.fastq.gz
mpimg_L24597-1_DH-RNA-074_S74_R1_001.fastq.gz
mpimg_L24598-1_DH-RNA-075_S75_R1_001.fastq.gz
mpimg_L24599-1_DH-RNA-076_S76_R1_001.fastq.gz
mpimg_L24600-1_DH-RNA-077_S77_R1_001.fastq.gz
mpimg_L24601-1_DH-RNA-078_S78_R1_001.fastq.gz
mpimg_L24602-1_DH-RNA-079_S79_R1_001.fastq.gz
mpimg_L24588-1_DH-RNA-065_S65_R2_001.fastq.gz
mpimg_L24589-1_DH-RNA-066_S66_R2_001.fastq.gz
mpimg_L24590-1_DH-RNA-067_S67_R2_001.fastq.gz
mpimg_L24591-1_DH-RNA-068_S68_R2_001.fastq.gz
mpimg_L24592-1_DH-RNA-069_S69_R2_001.fastq.gz
mpimg_L24593-1_DH-RNA-070_S70_R2_001.fastq.gz
mpimg_L24594-1_DH-RNA-071_S71_R2_001.fastq.gz
mpimg_L24595-1_DH-RNA-072_S72_R2_001.fastq.gz
mpimg_L24596-1_DH-RNA-073_S73_R2_001.fastq.gz
mpimg_L24597-1_DH-RNA-074_S74_R2_001.fastq.gz
mpimg_L24598-1_DH-RNA-075_S75_R2_001.fastq.gz
mpimg_L24599-1_DH-RNA-076_S76_R2_001.fastq.gz
mpimg_L24600-1_DH-RNA-077_S77_R2_001.fastq.gz

```
mpimg_L24601-1_DH-RNA-078_S78_R2_001.fastq.gz
mpimg_L24602-1_DH-RNA-079_S79_R2_001.fastq.gz

mpimg_L26725-1_DH-RNA-093_S416_R1_001.fastq.gz
mpimg_L26726-1_DH-RNA-094_S417_R1_001.fastq.gz
mpimg_L26727-1_DH-RNA-095_S418_R1_001.fastq.gz
mpimg_L26728-1_DH-RNA-096_S419_R1_001.fastq.gz
mpimg_L26729-1_DH-RNA-097_S420_R1_001.fastq.gz
mpimg_L26730-1_DH-RNA-098_S421_R1_001.fastq.gz
mpimg_L26731-1_DH-RNA-099_S422_R1_001.fastq.gz
mpimg_L26732-1_DH-RNA-100_S423_R1_001.fastq.gz
mpimg_L26733-1_DH-RNA-101_S424_R1_001.fastq.gz
mpimg_L26734-1_DH-RNA-102_S425_R1_001.fastq.gz
mpimg_L26735-1_DH-RNA-103_S426_R1_001.fastq.gz
mpimg_L26736-1_DH-RNA-104_S427_R1_001.fastq.gz
mpimg_L26725-1_DH-RNA-093_S416_R2_001.fastq.gz
mpimg_L26726-1_DH-RNA-094_S417_R2_001.fastq.gz
mpimg_L26727-1_DH-RNA-095_S418_R2_001.fastq.gz
mpimg_L26728-1_DH-RNA-096_S419_R2_001.fastq.gz
mpimg_L26729-1_DH-RNA-097_S420_R2_001.fastq.gz
mpimg_L26730-1_DH-RNA-098_S421_R2_001.fastq.gz
mpimg_L26731-1_DH-RNA-099_S422_R2_001.fastq.gz
mpimg_L26732-1_DH-RNA-100_S423_R2_001.fastq.gz
mpimg_L26733-1_DH-RNA-101_S424_R2_001.fastq.gz
mpimg_L26734-1_DH-RNA-102_S425_R2_001.fastq.gz
mpimg_L26735-1_DH-RNA-103_S426_R2_001.fastq.gz
mpimg_L26736-1_DH-RNA-104_S427_R2_001.fastq.gz

mpimg_L26564-1_DH-RNA-063_S22_R1_001.fastq.gz
mpimg_L26565-1_DH-RNA-064_S23_R1_001.fastq.gz
mpimg_L26566-1_DH-RNA-080_S24_R1_001.fastq.gz
mpimg_L26567-1_DH-RNA-081_S25_R1_001.fastq.gz
mpimg_L26568-1_DH-RNA-082_S26_R1_001.fastq.gz
mpimg_L26569-1_DH-RNA-083_S27_R1_001.fastq.gz
mpimg_L26570-1_DH-RNA-084_S28_R1_001.fastq.gz
mpimg_L26571-1_DH-RNA-085_S44_R1_001.fastq.gz
mpimg_L26572-1_DH-RNA-XXX_S46_R1_001.fastq.gz
mpimg_L27772-1_DH-RNA-108_S4_R1_001.fastq.gz
mpimg_L27773-1_DH-RNA-109_S5_R1_001.fastq.gz
mpimg_L27774-1_DH-RNA-110_S6_R1_001.fastq.gz
mpimg_L26573-1_DH-RNA-087_S47_R1_001.fastq.gz
mpimg_L26574-1_DH-RNA-088_S48_R1_001.fastq.gz
mpimg_L26575-1_DH-RNA-089_S49_R1_001.fastq.gz
mpimg_L26576-1_DH-RNA-090_S50_R1_001.fastq.gz
mpimg_L26577-1_DH-RNA-091_S51_R1_001.fastq.gz
mpimg_L26578-1_DH-RNA-092_S52_R1_001.fastq.gz
mpimg_L26564-1_DH-RNA-063_S22_R2_001.fastq.gz
mpimg_L26565-1_DH-RNA-064_S23_R2_001.fastq.gz
mpimg_L26566-1_DH-RNA-080_S24_R2_001.fastq.gz
mpimg_L26567-1_DH-RNA-081_S25_R2_001.fastq.gz
mpimg_L26568-1_DH-RNA-082_S26_R2_001.fastq.gz
mpimg_L26569-1_DH-RNA-083_S27_R2_001.fastq.gz
mpimg_L26570-1_DH-RNA-084_S28_R2_001.fastq.gz
mpimg_L26571-1_DH-RNA-085_S44_R2_001.fastq.gz
mpimg_L26572-1_DH-RNA-XXX_S46_R2_001.fastq.gz
mpimg_L27772-1_DH-RNA-108_S4_R2_001.fastq.gz
mpimg_L27773-1_DH-RNA-109_S5_R2_001.fastq.gz
mpimg_L27774-1_DH-RNA-110_S6_R2_001.fastq.gz
mpimg_L26573-1_DH-RNA-087_S47_R2_001.fastq.gz
mpimg_L26574-1_DH-RNA-088_S48_R2_001.fastq.gz
mpimg_L26575-1_DH-RNA-089_S49_R2_001.fastq.gz
mpimg_L26576-1_DH-RNA-090_S50_R2_001.fastq.gz
mpimg_L26577-1_DH-RNA-091_S51_R2_001.fastq.gz
mpimg_L26578-1_DH-RNA-092_S52_R2_001.fastq.gz

D6105_lib_07222AAD_ACTTCGTT-GATGCGTT_R1_001.fastq.gz
W24C_2_10106AAD_CTGCCAAG-TCCATATA_R1_001.fastq.gz
D6107_lib_07224AAD_ATGAGAGG-TCGTCTTG_R1_001.fastq.gz
W48C_2_10108AAD_CGCCAGTC-CCAAGACG_R1_001.fastq.gz
D6106_lib_07223AAD_CGGTTGGT-TGCAGCGT_R1_001.fastq.gz
P24C_2_10107AAD_ACGCCGCA-ATGTTAAC_R1_001.fastq.gz
D6108_lib_07225AAD_CTCGCAAG-GATCTACG_R1_001.fastq.gz
P48C_2_10109AAD_CTAAACAA-TCGCTACG_R1_001.fastq.gz
D6109_lib_07226AAD_GATCTTGC-CCAATTCC_R1_001.fastq.gz
W24I_2_10110AAD_TATACCTC-TGTGACTA_R1_001.fastq.gz
D6111_lib_07228AAD_TAACGCCA-AGGCAAGA_R1_001.fastq.gz
```

W48I_2_10112AAD_ACTCTTAG-AATCCACG_R1_001.fastq.gz
D6110_lib_07227AAD_TCAGATAC-CGCGAGAC_R1_001.fastq.gz
P24I_2_10111AAD_CTCTTGAT-CCACTTCT_R1_001.fastq.gz
D6112_lib_07229AAD_GTCAACCA-ATATGCAA_R1_001.fastq.gz
P48I_2_10113AAD_GAGCAACA-GCATCTAC_R1_001.fastq.gz

mpimg_L27298-1_DH-Other-170_S158_R1_001.fastq.gz
mpimg_L27299-1_DH-Other-171_S255_R1_001.fastq.gz
mpimg_L27301-1_DH-Other-173_S257_R1_001.fastq.gz
mpimg_L27302-1_DH-Other-174_S258_R1_001.fastq.gz
mpimg_L27296-1_DH-Other-168_S156_R1_001.fastq.gz
mpimg_L27297-1_DH-Other-169_S157_R1_001.fastq.gz
mpimg_L27290-1_DH-Other-162_S150_R1_001.fastq.gz
mpimg_L27291-1_DH-Other-163_S151_R1_001.fastq.gz
mpimg_L27292-1_DH-Other-164_S152_R1_001.fastq.gz
mpimg_L27293-1_DH-Other-165_S153_R1_001.fastq.gz
mpimg_L27294-1_DH-Other-166_S154_R1_001.fastq.gz
mpimg_L27295-1_DH-Other-167_S155_R1_001.fastq.gz
mpimg_L27287-1_DH-Other-159_S147_R1_001.fastq.gz
mpimg_L27288-1_DH-Other-160_S148_R1_001.fastq.gz
mpimg_L27289-1_DH-Other-161_S149_R1_001.fastq.gz
mpimg_L27305-1_DH-Other-177_S261_R1_001.fastq.gz
mpimg_L27306-1_DH-Other-178_S262_R1_001.fastq.gz
mpimg_L27304-1_DH-Other-176_S260_R1_001.fastq.gz
mpimg_L27298-1_DH-Other-170_S158_R2_001.fastq.gz
mpimg_L27299-1_DH-Other-171_S255_R2_001.fastq.gz
mpimg_L27301-1_DH-Other-173_S257_R2_001.fastq.gz
mpimg_L27302-1_DH-Other-174_S258_R2_001.fastq.gz
mpimg_L27296-1_DH-Other-168_S156_R2_001.fastq.gz
mpimg_L27297-1_DH-Other-169_S157_R2_001.fastq.gz
mpimg_L27290-1_DH-Other-162_S150_R2_001.fastq.gz
mpimg_L27291-1_DH-Other-163_S151_R2_001.fastq.gz
mpimg_L27292-1_DH-Other-164_S152_R2_001.fastq.gz
mpimg_L27293-1_DH-Other-165_S153_R2_001.fastq.gz
mpimg_L27294-1_DH-Other-166_S154_R2_001.fastq.gz
mpimg_L27295-1_DH-Other-167_S155_R2_001.fastq.gz
mpimg_L27287-1_DH-Other-159_S147_R2_001.fastq.gz
mpimg_L27288-1_DH-Other-160_S148_R2_001.fastq.gz
mpimg_L27289-1_DH-Other-161_S149_R2_001.fastq.gz
mpimg_L27305-1_DH-Other-177_S261_R2_001.fastq.gz
mpimg_L27306-1_DH-Other-178_S262_R2_001.fastq.gz
mpimg_L27304-1_DH-Other-176_S260_R2_001.fastq.gz

**Genome browser session**
(e.g. UCSC)

Reviewers can view the ChIP-Seq data at:
https://genome-euro.ucsc.edu/s/apmagalhaes/SubOpt_ChIP

## Methodology

**Replicates**

ChIP-Seq experiments were performed with 3 replicates for FLAG-NGN2 and CEBP/a. Bulk RNA-seq and TTSLAMseq experiments were performed with 3 biological replicates.

**Sequencing depth**

Total reads Uniquely mapped reads Length of reads Type Library
83642894 54949626 100bp Pair-end HAP1_Parental_WT_rep1
71027423 47610277 100bp Pair-end HAP1_Parental_WT_rep2
56951635 38986064 100bp Pair-end HAP1_Parental_WT_rep3
56955224 39329753 100bp Pair-end HAP1_HOXD4_WT_GFP_rep1
62124406 41271109 100bp Pair-end HAP1_HOXD4_WT_GFP_rep2
65732072 44132421 100bp Pair-end HAP1_HOXD4_WT_GFP_rep3
72989824 49331533 100bp Pair-end HAP1_HOXD4_AroPERFECT_GFP_rep1
75091733 48798291 100bp Pair-end HAP1_HOXD4_AroPERFECT_GFP_rep2
64293354 43258227 100bp Pair-end HAP1_HOXD4_AroPERFECT_GFP_rep3
63446798 43938397 100bp Pair-end HAP1_HOXD4_AroPLUS_GFP_rep1
84882022 53853668 100bp Pair-end HAP1_HOXD4_AroPLUS_GFP_rep2
67722437 40444092 100bp Pair-end HAP1_HOXD4_AroPLUS_GFP_rep3
104794729 53935807 100bp Pair-end HAP1_HOXD4_KO_rep1
72708520 42280338 100bp Pair-end HAP1_HOXD4_KO_rep2
85083118 37090405 100bp Pair-end HAP1_HOXD4_KO_rep3

78214706 61657120 100bp Pair-end ZIP13K2_WT_1
71899108 56656736 100bp Pair-end ZIP13K2_WT_2
59160286 46943262 100bp Pair-end ZIP13K2_WT_3
84741214 59010761 100bp Pair-end NGN2_WT_1
85396044 58477095 100bp Pair-end NGN2_WT_2
84726168 60119354 100bp Pair-end NGN2_WT_3

72071817 58259440 100bp Pair-end NGN2_AroLITE_1
67907275 53515875 100bp Pair-end NGN2_AroLITE_2
67771801 55279375 100bp Pair-end NGN2_AroLITE_3
88930215 70855293 100bp Pair-end NGN2_AroPERFECT_1
93952187 75443215 100bp Pair-end NGN2_AroPERFECT_2
84778537 68363391 100bp Pair-end NGN2_AroPERFECT_3

42321075 29857940 100bp Pair-end C2C12_WT_1
64598444 46267749 100bp Pair-end C2C12_WT_2
82306985 59149573 100bp Pair-end C2C12_WT_3
80946669 63282013 100bp Pair-end MYOD1_AroLite_1
74033673 56197975 100bp Pair-end MYOD1_AroLite_2
58269310 44063566 100bp Pair-end MYOD1_AroLite_3
46052693 36169911 100bp Pair-end MYOD1_AroLiteC_1
49238600 37820987 100bp Pair-end MYOD1_AroLiteC_2
33461352 25298292 100bp Pair-end MYOD1_AroLiteC_3
59090530 55449892 100bp Pair-end MYOD1_AroPerfect_1
59553741 57216936 100bp Pair-end MYOD1_AroPerfect_2
67867623 64948740 100bp Pair-end MYOD1_AroPerfect_3
30402772 20460464 100bp Pair-end MYOD1_AroPerfectC_1
32400056 21473188 100bp Pair-end MYOD1_AroPerfectC_2
37670706 26130992 100bp Pair-end MYOD1_AroPerfectC_3
30345574 21932837 100bp Pair-end MYOD1_WT_1
32627684 24050399 100bp Pair-end MYOD1_WT_2
23852795 16502519 100bp Pair-end MYOD1_WT_3

45083745 44443621 50 bp Single end WT_24_ChIP_rep1
47757874 47104678 50 bp Single end WT_24_ChIP_rep2
46055986 45487287 50 bp Single end WT_48_ChIP_rep1
45561595 45349108 50 bp Single end WT_48_ChIP_rep2
45467504 45044928 50 bp Single end IS15_24_ChIP_rep1
47081785 46445180 50 bp Single end IS15_24_ChIP_rep2
50391760 49809562 50 bp Single end IS15_48_ChIP_rep1
47302784 46575423 50 bp Single end IS15_48_ChIP_rep2
45972769 45491878 50 bp Single end WT_24_ChIP_input_rep1
48567463 48270072 50 bp Single end WT_24_ChIP_input_rep2
46889096 46423177 50 bp Single end WT_48_ChIP_input_rep1
48057746 47646638 50 bp Single end WT_48_ChIP_input_rep2
46780833 46170463 50 bp Single end IS15_24_ChIP_input_rep1
49335393 49039314 50 bp Single end IS15_24_ChIP_input_rep2
45899181 45056772 50 bp Single end IS15_48_ChIP_input_rep1
46974517 46887338 50 bp Single end IS15_48_ChIP_input_rep2

51712478 51678156 100bp Pair-end AroLITE_24_2
61068607 61002674 100bp Pair-end AroLITE_24_3
74767419 74688631 100bp Pair-end AroPERFECT_24_2
66344050 66286721 100bp Pair-end AroPERFECT_24_3
65551957 65496639 100bp Pair-end NGN2_24_1
69964691 69909977 100bp Pair-end NGN2_24_2
45990628 45958665 100bp Pair-end AroLITE_48_1
52756925 52715022 100bp Pair-end AroLITE_48_2
56012770 55971775 100bp Pair-end AroLITE_48_3
47400492 47361689 100bp Pair-end AroPERFECT_48_1
40986292 40953444 100bp Pair-end AroPERFECT_48_2
48033066 47995766 100bp Pair-end AroPERFECT_48_3
59224278 59179536 100bp Pair-end NGN2_48_1
51833944 51790572 100bp Pair-end NGN2_48_2
55064698 55025027 100bp Pair-end NGN2_48_3
60332885 60260788 100bp Pair-end NGN2_AroLITE_input
52295769 52226140 100bp Pair-end NGN2_AroPERFECT_input
72991162 72903923 100bp Pair-end NGN2_WT_input

22813448 2066101 100bp Single-end ZIP13K2_r1
23694568 2680532 100bp Single-end ZIP13K2_r2
95313445 15277183 100bp Single-end NGN2_WT_12h_r1
63274105 9573508 100bp Single-end NGN2_WT_12h_r2
63464230 9373107 100bp Single-end NGN2_WT_12h_r3
84398829 14917632 100bp Single-end NGN2_WT_24h_r1
76779317 15055873 100bp Single-end NGN2_WT_24h_r2
87319181 16552717 100bp Single-end NGN2_WT_24h_r3
77479680 10457921 100bp Single-end NGN2_AroPERFECT_12h_r1
63477115 9794869 100bp Single-end NGN2_AroPERFECT_12h_r2
75570606 11082568 100bp Single-end NGN2_AroPERFECT_12h_r3
72830923 13159432 100bp Single-end NGN2_AroPERFECT_24h_r2
95041489 16849381 100bp Single-end NGN2_AroPERFECT_24h_r3
89364578 12395003 100bp Single-end NGN2_AroLITE_12h_r1

75578998 11261735 100bp Single-end NGN2_AroLITE_12h_r2
89349138 12255078 100bp Single-end NGN2_AroLITE_12h_r3
83998939 14418922 100bp Single-end NGN2_AroLITE_24h_r2
66904097 12572224 100bp Single-end NGN2_AroLITE_24h_r3

**Antibodies**

For FLAG-NGN2 ChIP-seq in ZIP13K2 cells FLAG (F1804, 1:2000). For C/EBPa ChIP-seq in RCH-rtTA cells GFP clone 3E6 (A-11120).

**Peak calling parameters**

Raw reads of treatment and input samples were subjected to adapter and quality trimming with cutadapt (version 2.4; parameters: --nextseq-trim 20 --overlap 5 --minimum-length 25 --adapter AGATCGGAAGAGC -A AGATCGGAAGAGC). Reads were aligned separately to the mouse genome (mm10) or human genome (hg38) using bwa with the 'mem' command (version v0.7.17, default parameters). A sorted BAM file was obtained and indexed using samtools with the 'sort' and 'index' commands (version 1.10). Duplicate reads were identified and removed using gatk (version 4.1.4.1) with the 'MarkDuplicates' command and default parameters. Technical replicates of treatment and input samples were merged respectively using samtools 'merge'.
Peaks were called with reads aligning to the mouse genome only using MACS3 'callpeak' (version 3.0.8 b1; parameters --bdg --SPMR) using the input samples as control samples.
Genome-wide coverage tracks for single and merged replicates normalized by library size and input signal was subtracted using MACS3 output.

**Data quality**

Quality of raw reads was assessed using FastQC. Reads were trimmed using cutadapt in order to remove low-quality bases and adapter content.

% total deduplicated  percentage

62.83 WT_24_ChIP_rep1
73.07 WT_24_ChIP_rep2
74.94 WT_48_ChIP_rep1
81.54 WT_48_ChIP_rep2
75.95 IS15_24_ChIP_rep1
76.73 IS15_24_ChIP_rep2
72.00 IS15_48_ChIP_rep1
72.56 IS15_48_ChIP_rep2

80.22 WT_24_ChIP_rep1_Input
64.69 WT_24_ChIP_rep2_Input
77.28 WT_48_ChIP_rep1_Input
63.18 WT_48_ChIP_rep2_Input
79.23 IS15_24_ChIP_rep1_Input
63.96 IS15_24_ChIP_rep2_Input
78.34 IS15_48_ChIP_rep1_Input
66.27 IS15_48_ChIP_rep2_Input

66.29 ZIP13K2_NGN2_24h_AroLITE_2_Input
67.99 ZIP13K2_NGN2_24h_AroLITE_2
66.29 ZIP13K2_NGN2_24h_AroLITE_3_Input
67.02 ZIP13K2_NGN2_24h_AroLITE_3
62.07 ZIP13K2_NGN2_24h_AroPERFECT_1_Input
58.45 ZIP13K2_NGN2_24h_AroPERFECT_1
62.07 ZIP13K2_NGN2_24h_AroPERFECT_2_Input
67.34 ZIP13K2_NGN2_24h_AroPERFECT_2
62.07 ZIP13K2_NGN2_24h_AroPERFECT_3_Input
69.38 ZIP13K2_NGN2_24h_AroPERFECT_3
55.76 ZIP13K2_NGN2_24h_WT_1_Input
72.08 ZIP13K2_NGN2_24h_WT_1
55.76 ZIP13K2_NGN2_24h_WT_2_Input
63.96 ZIP13K2_NGN2_24h_WT_2
66.29 ZIP13K2_NGN2_48h_AroLITE_1_Input
61.48 ZIP13K2_NGN2_48h_AroLITE_1
66.29 ZIP13K2_NGN2_48h_AroLITE_2_Input
53.67 ZIP13K2_NGN2_48h_AroLITE_2
66.29 ZIP13K2_NGN2_48h_AroLITE_3_Input
60.54 ZIP13K2_NGN2_48h_AroLITE_3
62.07 ZIP13K2_NGN2_48h_AroPERFECT_1_Input
67.40 ZIP13K2_NGN2_48h_AroPERFECT_1
62.07 ZIP13K2_NGN2_48h_AroPERFECT_2_Input
69.76 ZIP13K2_NGN2_48h_AroPERFECT_2
62.07 ZIP13K2_NGN2_48h_AroPERFECT_3_Input
65.80 ZIP13K2_NGN2_48h_AroPERFECT_3
55.76 ZIP13K2_NGN2_48h_WT_1_Input
72.81 ZIP13K2_NGN2_48h_WT_1
55.76 ZIP13K2_NGN2_48h_WT_2_Input
73.15 ZIP13K2_NGN2_48h_WT_2
55.76 ZIP13K2_NGN2_48h_WT_3_Input
71.64 ZIP13K2_NGN2_48h_WT_3
61.25 ZIP13K2_WT_1_Input
40.92 ZIP13K2_WT_1

61.25 ZIP13K2_WT_2_Input
27.87 ZIP13K2_WT_2
61.25 ZIP13K2_WT_3_Input
18.67 ZIP13K2_WT_3

| | |
|---|---|
| Software | cutadapt<br>bwa mem<br>Star Aligner<br>samtools<br>gatk<br>MACS3<br>bamCoverage<br>SLAM-DUNK<br>featureCounts<br>seqtk |

# Flow Cytometry

## Plots

Confirm that:

☒ The axis labels state the marker and fluorochrome used (e.g. CD4-FITC).

☒ The axis scales are clearly visible. Include numbers along axes only for bottom left plot of group (a 'group' is an analysis of identical markers).

☒ All plots are contour plots with outliers or pseudocolor plots.

☐ A numerical value for number of cells or percentage (with statistics) is provided.

## Methodology

| | |
|---|---|
| Sample preparation | Cells were fixed for 15 minutes in 4% PFA at room temperature. This was followed by two washes in PBS. Flow cytometry workflow for RCH-rtTA cells provided in methods under "C/EBPα mediated B-cell to macrophage transdifferentiation" and "FACS analysis of CD66a and FCGR2A during C/EBPα-mediated B-cell to macrophage differentiation" |
| Instrument | BD FACS Celesta |
| Software | FACS Diva for collection and FlowJo for analysis |
| Cell population abundance | Cell population abundance is represented as normalized mode |
| Gating strategy | Gating for negative and positive population was determined with untreated or isotype controls. |

☒ Tick this box to confirm that a figure exemplifying the gating strategy is provided in the Supplementary Information.

