## [Peer Review File · Nature Cell Biology]

Peer Review Information

Journal: Nature Cell Biology

Manuscript Title: Dr Denes Hnisz

Corresponding author name(s): An activity-specificity trade-off encoded in human transcription factors

Reviewer Comments & Decisions:

Decision Letter, initial version:
--

*Please delete the link to your author homepage if you wish to forward this email to co-authors.

Dear Dr Hnisz,

Your manuscript, "Transcription factors as suboptimal prion-like sequences", has now been seen by 3 referees, who are experts in transcriptional regulation and lineage commitment (referee 1); biomolecular condensation (referee 2); and chromatin remodelling and sequencing (referee 3). I apologize once again for the delay.

As you will see from their comments (attached below) they find this work of potential interest, but have raised substantial concerns, which in our view would need to be addressed with considerable revisions before we can consider publication in Nature Cell Biology.

Nature Cell Biology editors discuss the referee reports in detail within the editorial team, including the chief editor, to identify key referee points that should be addressed with priority, and requests that are overruled as being beyond the scope of the current study. To guide the scope of the revisions, I have listed these points below. We are committed to providing a fair and constructive peer-review process, so please feel free to contact me if you would like to discuss any of the referee comments further.

I should stress that the referees' concerns point to unclear effects of aromatic residue spacing and unclear mechanistic links between this spacing, biomolecular condensation, and transcriptional activation activity, which would need to be addressed with experiments and data, and reconsideration of the study for this journal and re-engagement of referees would depend on strength of these revisions.

In particular, it would be essential to:

- A) Experimentally address concerns from all reviewers regarding the generalizability of your proposed mechanism of aromatic residue spacing on TFs (all Reviewers; this would entail further strengthening of data not only of HOXD4 but also on C/EBPa, NFAT5, and EGR1)
- B) Address with new data whether alterations in aromatic residue spacing affect biomolecular condensation as opposed to disrupting activation domains (Reviewers #1 and #2), protein stability (all Reviewers), or altering protein-protein interactions independently of biomolecular condensation (Reviewers #2 and #3).
- C) Experimentally test potential links between transcriptional factor activity and biomolecular condensation in line with current literature (PMID: 35537448 DOI: 10.1016/j.molcel.2022.04.017)

(reviewer #3)

D) Further characterize biomolecular condensation behaviour (Reviewers #2 and #3).

E) All other referee concerns pertaining to strengthening existing data, providing controls, methodological details, clarifications and textual changes, should also be addressed.

F) Finally please pay close attention to our guidelines on statistical and methodological reporting (listed below) as failure to do so may delay the reconsideration of the revised manuscript. In particular please provide:

We would be happy to consider a revised manuscript that would satisfactorily address these points, unless a similar paper is published elsewhere, or is accepted for publication in Nature Cell Biology in the meantime.

In contrast, although we agree with referee #2 that assess potential degran-like activity would provide valuable insights, we consider this point to be beyond the scope of the present study. Thus, addressing it experimentally will not be necessary for reconsideration of the manuscript at this journal.

- ensure that it conforms to our format instructions and publication policies (see below and www.nature.com/nature/authors/).

- provide a point-by-point rebuttal to the full referee reports verbatim, as provided at the end of this letter.

- provide the completed Editorial Policy Checklist (found here <https://www.nature.com/authors/policies/Policy.pdf>), and Reporting Summary (found here <https://www.nature.com/authors/policies/ReportingSummary.pdf>). This is essential for reconsideration of the manuscript and these documents will be available to editors and referees in the event of peer review. For more information see <http://www.nature.com/authors/policies/availability.html> or contact me.

Nature Cell Biology is committed to improving transparency in authorship. As part of our efforts in this direction, we are now requesting that all authors identified as 'corresponding author' on published papers create and link their Open Researcher and Contributor Identifier (ORCID) with their account on the Manuscript Tracking System (MTS), prior to acceptance. ORCID helps the scientific community

achieve unambiguous attribution of all scholarly contributions. You can create and link your ORCID from the home page of the MTS by clicking on 'Modify my Springer Nature account'. For more information please visit www.springernature.com/orcid.

[Redacted]

We would like to receive a revised submission within six months. We would be happy to consider a revision even after this timeframe, however if the resubmission deadline is missed and the paper is eventually published, the submission date will be the date when the revised manuscript was received.

We hope that you will find our referees' comments, and editorial guidance helpful. Please do not hesitate to contact me if there is anything you would like to discuss.

Best wishes,

Daryl Jason David

Daryl Jason Verzosa David, PhD

Senior Editor, Nature Cell Biology
Nature Portfolio

Heidelberger Platz 3, 14197 Berlin, Germany
Email: daryl.david@nature.com
ORCID: <https://orcid.org/0000-0002-9253-4805>

Reviewers' Comments:

Reviewer #1:

Remarks to the Author:

Naderi et al. report the finding of periodic blocks of aromatic residuals as a common feature of transcription factors, and that submaximal periodicity is frequently used to engender cell-type specificity of transcription factor function, much like the sub-optimal binding of transcription factors to enhancers. The authors systematically examined the 1500 annotated human transcription factors, and found that nearly 1/3 contain aromatic residue blocks that are periodically distributed. They found that altering the periodicity, through site-directed mutagenesis, can modulate the activation potential of the TFs. Further, they demonstrated that enhancing the periodicity of aromatic residue blocks in

several TFs, including HOXD4, C/EBP α in the context of macrophage transdifferentiating, NGN2 in IPS neural differentiation, and MYOD1 in myotube differentiation, led to alteration of transactivation capacity of the domain, phase separation capacity, and cellular phenotype. The authors further demonstrate that periodicity of aromatic residue blocks enhances TF IDR's ability to form in liquid-liquid phase separation, providing a mechanism by which such feature may promote transcriptional activation.

The current study offers several novel and potentially important conceptual advances. First, the periodicity of aromatic residue blocks is related to liquid-liquid phase condensate formation and transcriptional activation is novel, and provides new insights into the grammar of transcription factor IDR. Second, suboptimal arrangement of aromatic residues is a common feature of TF that could encode cell-type specificity of enhancers is interesting and novel. The experimental design and analyses are generally rigorous, thorough, and well done. On the other hands, the manuscript needs further improvement in clarity, and better articulation of the conclusions. More statistics should be performed for certain analyses. See below for details:

1. The method for calculating the periodicity score states that "The three spacer sub-groups were weighed by 1, 1.1 and 1.2 for the lengths of 4-9, 10-20 and 21-30 residues in a single spacer, respectively." What is the rationale for the specific weight for each group? Were those number arbitrarily defined or deduced from certain equations? Does the scoring method indicate that the number of periodic blocks weights much more than the spacer distance? Moreover, this algorithm does not seem to account for the uniformity, which later seems to be an important factor, represented by Ω_{Aro} .
2. Do the authors suggest that out of the 2,022 regions of significant periodicity across human proteins, 63 were located inside TF IDR regions, 333 were located in non-TF IDR regions, and 98% of the rest were overlapping a domain annotated in InterPro? I found this part very confusing for how it demonstrate that periodicity of TF may be submaximal, instead of being just normal.
 - a) There are several concepts, TF vs. Non-TF, IDR vs. the interPro domain vs. non-IDR. Can the authors delineate those relationships in Venn Diagram or other helpful formats?
 - b) Statistically, TF doesn't seem to differ from all proteins (Fig. 1g), so it's unclear to me what is defined as "submaximal" in Fig. 1 and why is TF special.
3. It's not surprising that increasing the number of aromatic residues will maximize the phase separation, shown by AroPLUS. It seems that with the same number of aromatic residues, with increased periodicity alone (or lower Ω_{Aro}) could enhance the transactivation (Fig. 2a) of HOXD4 but not the phase separation in vitro (Fig. 2b), which is partially contradictory to the sub-title.
4. In the knock-in assay, there was higher expression of HOXD4 (Fig. 2h and 2i). Was the increased granularity and intense cluster formation reflecting this or other functional change of HOXD4? Can the authors experimentally distinguished the "gain of function" of the protein per se from gain of expression?
5. One missing experiment is the ChIP-seq of AroLITE C/EBP α . If the periodicity of aromatic residuals within IDR is what contribute to weaker binding of TFs, I would expect to see that C/EBP α AroLITE group loses weaker binding sites in comparison to WT group, but not at the strongly binding sites. Unexpectedly, the AroLITE NGN2 group almost lost all DNA binding sites (Fig. 4i) – it seems to abolish even the function of HLH DNA-binding domain. Why is that? How does it impact the overall conclusion?
6. The authors claimed that they performed scRNA-seq with C/EBP α variants. It's unclear from which variant Fig. 3f, 3g, and 3l were generated. Why aren't there differential clustering analysis with different variants?
7. Why is AroPERFECT IS10 in Fig. 3A having more aromatic residues but barely any luciferase activity? Likewise, why is AroPERFECT with perfect periodicity completely lost transactivation capacity?

Those observations are contradictory to the overall claim that “IDRs encode periodically arranged aromatic residues that contribute to activity via multivalent, weak interactions with other disordered protein regions.” What’s concerning is that both cases happen to disrupt the minimal activity domain (AD). I would suggest to perform modulation outside of AD regions.

8. All observations taken together appear to fail to coherently support one conclusion. So far there isn’t a consensus pattern and the results seem to differ gene-by-gene.

Minor

9. The title is elusive and ambiguous to follow. Suggest to change to a more informative and clearer title.

10. Fig. 3k. The color scale is problematic, which makes WT look weaker binding than AroPERFECT IS15, although they’re labeled as shared peaks.

11. Some references seem incorrect. For example: “An emerging view in turn suggests that TF IDRs may contribute to transcriptional activity by engaging in multivalent, weak interactions¹⁰”. Reference 10 is this article “Direct neuronal reprogramming: Fast forward from new concepts toward therapeutic approaches”, which is irrelevant to TF IDR.

Reviewer #2:

Remarks to the Author:

Summary of the key results

The authors provide a compelling argument that the patterning of aromatic residues in transcription factors can be altered to encode gain-of-function transcriptional profiles. The authors combine *in vitro* experiments, *in cell* differentiation work, and extensive RNA-seq profiling to provide a compelling narrative that links transcription factor disordered region sequence to molecular function.

Originality and significance:

The work is novel and does a good job of bridging two areas; transcription factor IDRs and low-complexity sequence phase separation. The results are significant and illustrate compelling evidence for how IDR sequence can influence cellular phenotype.

Data & methodology:

The approach is appropriate and the quality of data and figures is excellent as best I can tell

Appropriate use of statistics:

The authors do a good job in explicitly calling out statistical tests and being precise with their null models, all of which help to convey rigor of work.

Conclusions:

The conclusions seem valid and reliable. In general, the authors avoid over-interpretation, and while there are some considerations I think could help further strengthen the manuscript, from my perspective the conclusions drawn are well supported by the data.

Suggested improvements:

I think a major suggestion I would make is to reconsider the title. The challenge is that “prion-like” sequences means different things to different people. I 100% agree with the authors’ use here, and think given their results and message, the title is fine. HOWEVER, from bitter experience, I know that

referring to anything as 'prion-like' raises many (often irrelevant) questions from folks from other fields ('they these cause vCJD? How do these proteins spread? So they are alpha-helical but can convert into beta-sheet, like the prion protein?). My strong suggestion is that these issues can be largely avoided by changing the title to something that avoids the word prion or prion-like. I think the challenge with "suboptimal" as a term in the title is that it requires the reader to know what 'optimal' means, which truthfully is a loaded term (what is the optimal function of a prion-like domain?). With these comments in mind, I might suggest the authors revisit their title to something that clearly reflects the major functional takeaway from the work (that aromatic patterning in transcription factor IDRs can influence transcriptional outputs).

The authors introduce the idea of "Submaximal periodicity", yet this is not actually defined. I think it is important the authors (1) explicitly define what it is and (2) explain why this is an interesting feature to examine.

Fig 1b is beautiful but uninterpretable without the caption. I suggest the authors add labels so we can understand what the circles/bar heights mean intuitively. Also the legend for panel b appears to be in panel c?

I am surprised that removing all the aromatic residues from HOXD4 IDR results in condensates forming at 5 μ M protein. Given the sequence composition, this protein should be highly soluble. At 10% W/V PEG8000 you're at approximately 12.5 mM PEG in solution, and I can't help but wonder if given the 2500x excess of PEG in this system you're actually observing PEG-protein co-condensates? I'm not sure this actually changes the interpretation, but my hunch is that the dominating interaction here is the protein:PEG interaction, as opposed to the protein:protein interaction. This potential interpretation could be completely dismissed if labeled PEG were used and it was clear that no PEG was in the droplets. Absent that data (which I am not suggesting are necessary for publication, but it would be great, of course!), I think it may be prudent to raise this possible explanation, not that it alters the main message but does alter who a reader should think about the molecular driving forces in this experiment.

While not essential, if an ARO-PERFECT style design for NFAT5 or EGR1 showed enhanced transcriptional output this would really help cement this principle of imperfections actually being detrimental to total transcriptional output.

The subcellular localization of the AroPERFECT and AroPLUS HOXD4 variants is striking! Could this be explained by different protein expression levels? I realize the western in 2i is there to make the point that there's autoregulation, but I guess I'm wondering if you cranked up WT expression to get comparable protein levels as is seen for PERFECT and PLUS would analogous transcriptional profiles be obtained? I recognize that at the endogenous loci, we should expect expression to be comparable, so I'm wondering if there's now something altered with protein turnover). Given there is a potential degron between residues 142 and 148 (immediately C-terminal of the IDR) I wonder if there's some funky post-translational regulation going on?

The breadth of systems exemplified in the second half of the paper is really remarkable, and a real strength of this work - regardless of the molecular origin (for which I can imagine there might be some questions/uncertainty) the actual phenotypic effect of changing aromatic periodicity is clear for all to see.

I have a few final suggestions for the discussion. Firstly, the connection between these IDRs and phase separation is clearly made, but in reality, the data are largely agnostic in terms of whether or not phase separation is the actual underlying mechanism here. Perhaps, in reality, phase separation just correlates well another mechanism of interest? I agree that phase separation seems the most parsimonious explanation, but I also think this potential agnosticism is a strength of the work. I would encourage the authors to discuss the fact that while phase separation does enable a mechanistic interpretation, it does not exclude a model where aromatic patterning is tuning the intermolecular interaction of the IDR with TFIID or some other component in the transcriptional machinery (without the need for bona fide phase separation). Secondly, Staller et al. have proposed an 'acidic exposure' model, whereby the balance of hydrophobic/aromatic residues is balanced by charged residues that prevent tight intramolecular interaction, priming aromatic residues to be exposed. In this model, increasing the number of aromatic residues beyond some threshold led to a drop in transcriptional output (see Staller et al. 2022). It would be great if the authors could comment on this interpretation, which I do think is largely in line with the authors' findings here (note that the density of aromatic residues needed to see loss of activity was much higher than the authors test here, even in their AroPERFECT constructs). Perhaps the more evenly spaced aromatic residues further aid in their ability to engage with intermolecular targets, avoiding strong intramolecular interactions observed when aromatic residues are clustered together?

References: appropriate credit to previous work?

The enhanced liquidity upon more evenly-distributed aromatic residues mirrors conclusions from Holehouse et al Biochemistry 2021.

Clarity and context: lucidity of abstract/summary, appropriateness of abstract, introduction and conclusions

The manuscript is extremely clear and well-written.

Reviewer #3:

Remarks to the Author:

Transcription factors (TFs) are the central players of gene regulation and often have an effector domain in addition to a DNA binding domain. Although the DNA binding domains are relatively well characterized, understanding the features of the effector domains, such as transcription activation domains, have been more challenging due to a lack of apparent structure or linear motif logic. The recent advances in this area have described features that correlate well with transactivation domain function, such as long disordered regions with acidic, hydrophobic, and aromatic residues, and built models with predictive power (1,2,3). The manuscript of Naderi & Magelhaes et al further investigates one of these features, the role of periodicity of aromatic residues, towards engineering stronger activation domains. While the manuscript describes a systematic computational approach, the follow up experiments do not provide sufficient support for the conclusions and the general relevance of the observations.

Major points:

- The fact that a part of a protein's function can be improved is not novel as evolution "tinkers" towards needed function in terms of organismal fitness and not towards molecular optimality. The authors interpretation of the function of this lack of molecular optimality is purely speculative and does

not inform about general rules in transcriptional regulation.

- It is not clear if the assays used in cellular differentiation are relevant to judge the in vivo function. In absence of a clearer understanding of molecular mechanisms of the studied domains this seems a rather speculative interpretation that builds on existing (and highly debated) models of phase separation.

- Explaining the submaximal spacing between the aromatic residues with the same rationale used to explain the relevance of weak enhancer binding sites seems to be an over-reaching argument and an over-simplification, particularly in the absence of any molecular mechanism.

- The mutations of the peptide sequences could in principle change the stability of the protein. Whether such difference in total protein amount (Gal4-IDR) underlies the differential signal observed in the luciferase reporter assays needs to be tested. The same problem is indeed observed for the Hoxd4 experiments, where the gene expression analysis should be done on same-level-Hoxd4-expressing system. Similarly, the protein amounts should be controlled for the other lineage specifying TF experiments to test whether the effect observed is simply due to higher protein concentrations.

Further points:

- The formulation of the aim and the conclusion of the study are not clear.

The experiments shown in Ext. Dat. Fig 4 b-c suggest that a simple rule of periodicity in aromatic residues does not exist, let alone explaining the relevance of potential periodicity.

If the aim of the study is to engineer TFs with stronger activation potential (C/EBPa, Ngn2, Myod1) then adding one standard high performing activator domain on them would be logical rather than trying to find highly variant versions of the initial sequence, which not only diverges from functional relevance but even cause loss of function as in Hoxd4.

If the aim of the study is to engineer peptide sequences to create "prion like" features, then testing relevant features such as Q/N repeats would be discussed.

If the aim is to engineer new activator domains, then documented activator domain features such as hydrophobicity and negative charge distribution would be discussed.

- A critical conclusion of the study is the function of TF-aggregates on gene regulation. In order to probe any such link, experimental set-ups as in (4) would be needed.

- The Hoxd4 gene expression experiments (Figure 2g-i) need better controls/explanation/redoing since the mentioned effect is the smaller component (PC2) which does not seem to distinguish the parental cells from the wild-type misexpression. To compare the effect of periodicity of the aromatic residues on gene expression, the use of the same-expression-level lines (as in Ext Dat Fig 6) would be better suited. The similar problems seem to exist for Ngn2 and Myod1 experiments. Protein concentration of the modified C/EBPa, Ngn2 and Myod1 relative to wild type needs to be documented since altered levels of these proteins can in principle result in the same observations made. Similarly, other perturbed functions (which seems to be many, given the PCA analysis and the Volcano plots) need better explanation.

- In Extended Data Figure 6.e, description in the figure legend (granularity vs signal intensity) seems to be swapped "The normalized signal intensity was calculated by dividing standard deviation of

mEGFP signal of each nucleus by the corresponding mean mEGFP signal...Granularity scores of nuclei with corresponding mean nuclear mEGFP intensities."

- In Extended Data Figure 6.e: the granularity test should be done on live samples since fixation can alter the signal (5)

1. Sanborn, Adrian L., et al. "Simple biochemical features underlie transcriptional activation domain diversity and dynamic, fuzzy binding to Mediator." *Elife* 10 (2021)
2. Staller, Max V., et al. "Directed mutational scanning reveals a balance between acidic and hydrophobic residues in strong human activation domains." *Cell systems* 13.4 (2022)
3. Erijman, Ariel, et al. "A high-throughput screen for transcription activation domains reveals their sequence features and permits prediction by deep learning." *Molecular cell* 78.5 (2020)
4. Trojanowski, Jorge, et al. "Transcription activation is enhanced by multivalent interactions independent of phase separation." *Molecular cell* 82.10 (2022)
5. Irgen-Gioro, Shawn, et al. "Fixation can change the appearance of phase separation in living cells." *Elife* 11 (2022)

Methods should be written concisely, but should contain all elements necessary to allow interpretation and replication of the results. As a guideline, Methods sections typically do not exceed 3,000 words. The Methods should be divided into subsections listing reagents and techniques. When citing previous methods, accurate references should be provided and any alterations should be noted. Information must be provided about: antibody dilutions, company names, catalogue numbers and clone numbers for monoclonal antibodies; sequences of RNAi and cDNA probes/primers or company names and catalogue numbers if reagents are commercial; cell line names, sources and information on cell line identity and authentication. Animal studies and experiments involving human subjects must be reported in detail, identifying the committees approving the protocols. For studies involving human subjects/samples, a statement must be included confirming that informed consent was obtained. Statistical analyses and information on the reproducibility of experimental results should be provided

in a section titled "Statistics and Reproducibility".

All Nature Cell Biology manuscripts submitted on or after March 21 2016 must include a Data availability statement at the end of the Methods section. For Springer Nature policies on data availability see <http://www.nature.com/authors/policies/availability.html>; for more information on this particular policy see <http://www.nature.com/authors/policies/data/data-availability-statements-data-citations.pdf>. The Data availability statement should include:

- Accession codes for primary datasets (generated during the study under consideration and designated as "primary accessions") and secondary datasets (published datasets reanalysed during the study under consideration, designated as "referenced accessions"). For primary accessions data should be made public to coincide with publication of the manuscript. A list of data types for which submission to community-endorsed public repositories is mandated (including sequence, structure, microarray, deep sequencing data) can be found here <http://www.nature.com/authors/policies/availability.html#data>.
- Unique identifiers (accession codes, DOIs or other unique persistent identifier) and hyperlinks for datasets deposited in an approved repository, but for which data deposition is not mandated (see here for details <http://www.nature.com/sdata/data-policies/repositories>).
- At a minimum, please include a statement confirming that all relevant data are available from the authors, and/or are included with the manuscript (e.g. as source data or supplementary information), listing which data are included (e.g. by figure panels and data types) and mentioning any restrictions on availability.
- If a dataset has a Digital Object Identifier (DOI) as its unique identifier, we strongly encourage including this in the Reference list and citing the dataset in the Methods.

We recommend that you upload the step-by-step protocols used in this manuscript to the Protocol Exchange. More details can found at www.nature.com/protocolexchange/about.

All imaging data should be accompanied by scale bars, which should be defined in the legend. Cropped images of gels/blots are acceptable, but need to be accompanied by size markers, and to retain visible background signal within the linear range (i.e. should not be saturated). The boundaries of panels with low background have to be demarked with black lines. Splicing of panels should only be considered if unavoidable, and must be clearly marked on the figure, and noted in the legend with a statement on whether the samples were obtained and processed simultaneously. Quantitative comparisons between samples on different gels/blots are discouraged; if this is unavoidable, it should

only be performed for samples derived from the same experiment with gels/blots were processed in parallel, which needs to be stated in the legend.

FIGURE LEGENDS – must not exceed 350 words for each figure to allow fit on a single printed NCB page together with the figure. They must include a brief title for the whole figure, and short

descriptions of each panel with definitions of the symbols used, but without detailing methodology.

The total number of Supplementary Figures (not including the “unprocessed scans” Supplementary Figure) should not exceed the number of main display items (figures and/or tables (see our Guide to Authors and March 2012 editorial <http://www.nature.com/ncb/authors/submit/index.html#supinfo>; <http://www.nature.com/ncb/journal/v14/n3/index.html#ed>). No restrictions apply to Supplementary Tables or Videos, but we advise authors to be selective in including supplemental data.

GUIDELINES FOR EXPERIMENTAL AND STATISTICAL REPORTING

REPORTING REQUIREMENTS – To improve the quality of methods and statistics reporting in our papers we have recently revised the reporting checklist we introduced in 2013. We are now asking all life sciences authors to complete two items: an Editorial Policy Checklist (found here <https://www.nature.com/authors/policies/Policy.pdf>) that verifies compliance with all required editorial policies and a reporting summary (found

here <https://www.nature.com/authors/policies/ReportingSummary.pdf>) that collects information on experimental design and reagents. These documents are available to referees to aid the evaluation of the manuscript. Please note that these forms are dynamic 'smart pdfs' and must therefore be downloaded and completed in Adobe Reader. We will then flatten them for ease of use by the reviewers. If you would like to reference the guidance text as you complete the template, please access these flattened versions at <http://www.nature.com/authors/policies/availability.html>.

Author Rebuttal to Initial comments
--

General response to Reviewers

We thank the reviewers for their valuable comments and suggestions, which guided us to a substantially improved manuscript. The reviewers described the original submission as a study that “offers several novel and potentially important conceptual advances,” the results as “significant” and the conclusions as “valid and reliable”. The reviewers also noted that the manuscript would benefit from more clarity at certain sections, and Reviewer 3 noted a need for more support for the conclusions and the general relevance of the observations. We have revised the manuscript to address these and additional minor concerns. In brief:

1. We provide several new lines of experimental and analytical evidence further supporting the generalizability of the proposed mechanism of aromatic dispersion in IDRs.
 - i) Data on the successful ‘optimization’ on two additional lineage-defining TFs
 - ii) Data on successful ‘de-optimization’ of a periodic TF IDR
 - iii) Analyses of spacer sequences between aromatic residues revealing sequence composition rules of periodic regions
2. We include new data that provide further mechanistic insights into the functions and features of periodic IDRs:
 - i) Cell-based condensate assays that substantiate previous *in vitro* findings
 - ii) Reporter assays that provide strong support for our initial models that periodic TF IDRs are different from known minimal activation domains
3. We include additional data and analyses further supporting our main model that there is a trade-off between the activity and specificity encoded as submaximal aromatic spacing in TF IDRs.
4. Clarified key sections of the text, and revised the title.

The new data are consistent with and go beyond the insights described in the original submission. We show that at least a third of human transcription factors display traces of aromatic periodicity in their IDRs, and the periodic regions are distinct from known minimal activation domains. Increasing periodicity of aromatic residues enhanced liquid-like features of condensates, transcriptional activity, and reprogramming capacity, but compromised binding specificity. The data suggest an important evolutionary trade-off between activity and specificity encoded as submaximal aromatic dispersion in TF IDRs.

For easier inspection of the new text and data, the revised sections are highlighted with blue font in the manuscript.

Point-by-point response to Reviewers' comments:

Reviewer #1:

Remarks to the Author:

Naderi et al. report the finding of periodic blocks of aromatic residuals as a common feature of transcription factors, and that submaximal periodicity is frequently used to engender cell-type specificity of transcription factor function, much like the sub-optimal binding of transcription factors to enhancers. The authors systematically examined the 1500 annotated human transcription factors, and found that nearly 1/3 contain aromatic residue blocks that are periodically distributed. They found that altering the periodicity, through site-directed mutagenesis, can modulate the activation potential of the TFs. Further, they demonstrated that enhancing the periodicity of aromatic residue blocks in several TFs, including HOXD4, C/EBPa in the context of macrophage transdifferentiating, NGN2 in IPS neural differentiation, and MYOD1 in myotube differentiation, led to alteration of transactivation capacity of the domain, phase separation capacity, and cellular phenotype. The authors further demonstrate that periodicity of aromatic residue blocks enhances TF IDR's ability to form in liquid-liquid phase separation, providing a mechanism by which such feature may promote transcriptional activation.

The current study offers several novel and potentially important conceptual advances. First, the periodicity of aromatic residue blocks is related to liquid-liquid phase condensate formation and transcriptional activation is novel, and provides new insights into the grammar of transcription factor IDR. Second, suboptimal arrangement of aromatic residues is a common feature of TF that could encode cell-type specificity of enhancers is interesting and novel. The experimental design and analyses are generally rigorous, thorough, and well done. On the other hands, the manuscript needs further improvement in clarity, and better articulation of the conclusions. More statistics should be performed for certain analyses. See below for details:

We thank the reviewer for the noting the potentially important conceptual advances, and for the constructive comments. We improved the clarity and performed the requested experiments and analyses.

1. The method for calculating the periodicity score states that “The three spacer sub-groups were weighed by 1, 1.1 and 1.2 for the lengths of 4-9, 10-20 and 21-30 residues in a single spacer, respectively.” What is the rationale for the specific weight for each group? Were those number arbitrarily defined or deduced from certain equations? Does the scoring method indicate that the number of periodic blocks weights much more than the spacer distance? Moreover, this algorithm does not seem to account for the uniformity, which later seems to be an important factor, represented by Ω_{Aro} .

The values for the weighing of the spacer lengths were chosen arbitrarily. The rationale for weighing is that the more aromatic-rich a sequence is, the smaller the standard deviation of spacer sequences is expected. In the periodicity score calculations, the number of aromatic residues dominates the effect of weighing. Indeed, the score does not account for uniformity. We

have clarified these points in the methods (at the “Identification of periodic blocks in TF IDRs” section. We stress that we used three independent methods to characterize aromatic periodicity, two of which we developed. Each method has some weaknesses, as noted by the Reviewer, but overall, the outputs of the methods are consistent in that they identify largely the same regions that display traces of periodicity in TF IDRs.

2. Do the authors suggest that out of the 2,022 regions of significant periodicity across human proteins, 63 were located inside TF IDR regions, 333 were located in non-TF IDR regions, and 98% of the rest were overlapping a domain annotated in InterPro? I found this part very confusing for how it demonstrate that periodicity of TF may be submaximal, instead of being just normal.

The Reviewer is correct: 63 periodic regions were located inside TF IDRs, 333 were located in non-TF IDR regions, and 98% of the rest were overlapping a domain annotated in InterPro. TF IDRs were on average less periodic than prion-like domains. It is question what is “normal” periodicity for a protein, so we simply meant to imply that theoretically the periodicity of TF IDRs can be increased (i.e. is submaximal). We clarified the text (first paragraph of the “Submaximal periodicity of aromatic residues in TF IDRs” section in the results).

a) There are several concepts, TF vs. Non-TF, IDR vs. the interPro domain vs. non-IDR. Can the authors delineate those relationships in Venn Diagram or other helpful formats?

We removed the sentence on Interpro domains to avoid confusion.

b) Statistically, TF doesn’t seem to differ from all proteins (Fig. 1g), so it’s unclear to me what is defined as “submaximal” in Fig. 1 and why is TF special.

As noted above, “submaximal” periodicity means that the periodicity of TF IDRs in general can be increased. This notion is supported by other results e.g. that prion-like domains on average are more periodic than TF IDRs. Indeed, one can interpret this as TFs on average are not special, which is a point we were trying to make. We clarified the text accordingly (first paragraph of the “Submaximal periodicity of aromatic residues in TF IDRs” section in the results).

3. It’s not surprising that increasing the number of aromatic residues will maximize the phase separation, shown by AroPLUS. It seems that with the same number of aromatic residues, with increased periodicity alone (or lower Ω_{Aro}) could enhance the transactivation (Fig. 2a) of HOXD4 but not the phase separation *in vitro* (Fig. 2b), which is partially contradictory to the sub-title.

We clarified this in the text. In Figure 2a, we aimed to increase the aromatic periodicity of the HOXD4 IDR using multiple approaches. Increasing periodicity without changing the number of aromatic residues (AroPERFECT mutant) enhanced the liquid-like features of *in vitro* HOXD4 IDR droplets (Fig. 2d). Increasing aromatic dispersion without changing the number of aromatic residues enhanced the liquid-like features of *in vitro* other TF IDR droplets (HOXC4, **Extended**

Data Fig. 4j-k; C/EBP α , Fig. 4c). We clarified the text accordingly. Collectively these results suggest that increasing aromatic dispersion enhances liquid-like features of *in vitro* droplets. We note that this not necessarily predict the effect of the mutations on condensates in cells. Rather, the interactions whose dynamics are important for in vivo function are assayed in the droplet system. We clarified the Abstract, Results and Discussion sections accordingly.

4. In the knock-in assay, there was higher expression of HOXD4 (Fig. 2h and 2i). Was the increased granularity and intense cluster formation reflecting this or other functional change of HOXD4? Can the authors experimentally distinguish the “gain of function” of the protein per se from gain of expression?

We generated cells that express comparable levels of GFP-tagged wild type, AroPERFECT, and AroPLUS HOXD4 proteins using a PiggyBac transposon system. We detected higher granularity in the cells expressing the periodic mutants compared to wild type (**Fig. 3b-c**). The morphology and gene expression phenotypes were also corroborated using these cells (**new Extended Data Fig. 6g-i**). Furthermore, in cell-based condensate tethering system, we found that the condensates formed by the periodic HOXD4 IDR recruit significantly more RNA Polymerase II CTD (**new Fig. 3g-h**). These results collectively suggest that much of the effect of increased periodicity is likely not a result of gain of HOXD4 expression.

5. One missing experiment is the ChIP-seq of AroLITE C/BEPA. If the periodicity of aromatic residuals within IDR is what contribute to weaker binding of TFs, I would expect to see that C/BEPA AroLITE group loses weaker binding sites in comparison to WT group, but not at the strongly binding sites. Unexpectedly, the AroLITE NGN2 group almost lost all DNA binding sites (Fig. 4i) – it seems to abolish even the function of HLH DNA-binding domain. Why is that? How does it impact the overall conclusion?

This is an excellent question, that also revealed surprising new insights. We did not perform ChIP-seq of AroLITE C/BEPA, because this mutant is surprisingly retained in the cytoplasm (**new Extended Data Fig. 9d**). This result suggests a link between phase separation behavior of some IDRs and their nuclear import. We emphasize that the PERFECT IS15 C/EBP α mutant is localized in the nucleus as expected (**new Extended Data Fig. 9d**).

We include new data confirming that the NGN2 AroLITE mutant is nuclear (**new Extended Data Fig. 10e**), and is expressed at comparable levels as the wild type and AroPERFECT NGN2 (**Extended Data Fig 10d**), further substantiating the ChIP-Seq results. We believe that reduced genomic binding by the AroLITE NGN2 supports the idea that IDRs can impact genomic binding (which is important for our trade-off model). The results are consistent with recent IDR-deletion experiment of yeast TFs by the Barkai lab ¹.

6. The authors claimed that they performed scRNA-seq with C/EBPA variants. It's unclear from which variant Fig. 3f, 3g, and 3l were generated. Why aren't there differential clustering analysis with different variants?

We clarified this in the text and in the methods. The scRNA-Seq was performed on cells expressing the wild type, AroPERFECT IS15, and AroPERFECT IS10 variants. The clustering was performed on the merged data. We also performed clustering and analyses of individual samples, which led to consistent results.

7. Why is AroPERFECT IS10 in Fig. 3A having more aromatic residues but barely any luciferase activity? Likewise, why is AroPERFECT with perfect periodicity completely lost transactivation capacity? Those observations are contradictory to the overall claim that “IDRs encode periodically arranged aromatic residues that contribute to activity via multivalent, weak interactions with other disordered protein regions.” What’s concerning is that both cases happen to disrupt the minimal activity domain (AD). I would suggest to perform modulation outside of AD regions.

We thank the reviewer for these insightful suggestions. The AroPERFECT IS10 C/EBP α IDR contains eight more aromatic residues than the wild type, which makes the sequence substantially more hydrophobic. As a result, cells express it at a somewhat lower level in the luciferase experiments (**new Extended Data Fig. 7b**). We controlled for this in the transgenic experiments, where the expression levels are comparable (**Extended Data Fig. 7e**). *In vitro*, the AroPERFECT IS10 C/EBP α IDR droplets have reduced FRAP recovery (**Fig. 4c**), consistent with the lower activity measured in the reprogramming system.

To rule out the contribution of the minimal activation domains (AD) for the elevated activity of the PERFECT mutant, we generated additional PERFECT mutants in which we kept the AD intact. In brief, these results revealed that increasing periodicity of the region outside of the AD enhances the activity of the IDR (**new Fig. 4g**). Moreover, the periodic portion could be complemented by a periodic FUS IDR portion when attached to the AD (**new Fig. 4g**). We included similar experiments with HOXD4 (**new Fig 2f**). Finally, we performed tiling experiments which revealed that the activity of periodic regions cannot be mapped to a short minimal region for additional TF IDRs (OCT4: **new Extended Data Fig. 5d**; EGR1: **new Extended Data Fig 5f**; MYOD1: **new Extended Data Fig. 11b**).

8. All observations taken together appear to fail to coherently support one conclusion. So far there isn’t a consensus pattern and the results seem to differ gene-by-gene.

We propose a model that there is a trade-off between the activity and specificity of human transcription factors. We found that this trade-off is encoded as submaximal aromatic dispersion in the IDRs of numerous TFs, and provide several lines of experimental and analytical evidence in support of this model for multiple TFs.

The Reviewer is correct that there appear to be gene-specific effects. We believe that this simply reflects the fact that aromatic dispersion is not the *only* important feature of TFs IDRs, and aromatic dispersion evolves in the context of other sequence features. We provide substantial new data that explain specific nuances. For example:

i) The nature of spacer sequences creates constraints for the effect of aromatic dispersion. This is supported by several lines of analytical (**new Extended Data Fig. 1d-f, 4e**) and experimental approaches (**Fig. 2g**).

ii) The mode of activity of periodic IDRs appears distinct from and complementary to the activity conferred by minimal activation domains, which also create some constraints. In support of this view, we experimentally separated the two modes, and show that the activity of IDRs can be enhanced in portions that do not contain minimal activation domains (**new Fig. 2f, 4g, Extended Data Fig. 5d-f, 11b**).

Minor

9. The title is elusive and ambiguous to follow. Suggest to change to a more informative and clearer title.

We thank the Reviewer for this comment, which was also raised by Reviewer 2. We changed the title to “Suboptimization of human transcription factors”. We believe that this title generally captures the key message, is consistent with previous literature (Farley et al., Science 2015), and avoids the issues mentioned by both Reviewers.

10. Fig. 3k. The color scale is problematic, which makes WT looks weaker binding than AroPERFECT IS15, although they’re labeled as shared peaks.

There is on average weaker binding of WT at the shared peaks (also seen in the metaplots above the heatmaps, which are agnostic to color scale).

11. Some references seem incorrect. For example: “An emerging view in turn suggests that TF IDRs may contribute to transcriptional activity by engaging in multivalent, weak interactions¹⁰” . Reference 10 is this article “Direct neuronal reprogramming: Fast forward from new concepts toward therapeutic approaches”, which is irrelevant to TF IDR.

Thank you! We checked and fixed referencing.

Reviewer #2:

Remarks to the Author:

Summary of the key results

The authors provide a compelling argument that the patterning of aromatic residues in transcription factors can be altered to encode gain-of-function transcriptional profiles. The authors combine in vitro experiments, in cell differentiation work, and extensive RNA-seq profiling to provide a compelling narrative that links transcription factor disordered region sequence to molecular function.

Originality and significance:

The work is novel and does a good job of bridging two areas; transcription factor IDRs and low-complexity sequence phase separation. The results are significant and illustrate compelling evidence for how IDR sequence can influence cellular phenotype.

Data & methodology:

The approach is appropriate and the quality of data and figures is excellent as best I can tell

Appropriate use of statistics:

The authors do a good job in explicitly calling out statistical tests and being precise with their null models, all of which help to convey rigor of work.

Conclusions:

The conclusions seem valid and reliable. In general, the authors avoid over-interpretation, and while there are some considerations, I think could help further strengthen the manuscript, from my perspective the conclusions drawn are well supported by the data.

We thank the reviewer for the very positive comments!

Suggested improvements:

I think a major suggestion I would make is to reconsider the title. The challenge is that “prion-like” sequences means different things to different people. I 100% agree with the authors’ use here, and think given their results and message, the title is fine. HOWEVER, from bitter experience, I know that referring to anything as ‘prion-like’ raises many (often irrelevant) questions from folks from other fields (‘they these cause vCJD? How do these proteins spread? So they are alpha-helical but can convert into beta-sheet, like the prion protein?’). My strong suggestion is that these issues can be largely avoided by changing the title to something that avoids the word prion or prion-like.

We thank the Reviewer for the useful insights and suggestion. We changed the title to “Suboptimization of human transcription factors”. We believe that this title generally captures the key message, is consistent with previous literature (Farley et al., Science 2015)², and avoids the issues mentioned.

I think the challenge with “suboptimal” as a term in the title is that it requires the reader to know what ‘optimal’ means, which truthfully is a loaded term (what is the optimal function of a prion-like domain?). With these comments in mind, I might suggest the authors revisit their title to something that clearly reflects the major functional takeaway from the work (that aromatic patterning in transcription factor IDRs can influence transcriptional outputs).

We changed the title to “Suboptimization of human transcription factors”. We believe that this title generally captures the key message that there is a trade-off between activity and specificity encoded as aromatic dispersion in many TF IDRs.

The authors introduce the idea of “Submaximal periodicity”, yet this is not actually defined. I think it is important the authors (1) explicitly define what it is and (2) explain why this is an interesting feature to examine.

As noted above, “submaximal” periodicity means that the periodicity of TF IDRs in general can be increased. This notion is supported by other results e.g., that prion-like domains on average are more periodic than TF IDRs. We clarified the text (first paragraph of the “Submaximal periodicity of aromatic residues in TF IDRs” section in the results). Furthermore, we explain the reasons why this is an interesting feature to examine (third paragraph in the revised Introduction).

Fig 1b is beautiful but uninterpretable without the caption. I suggest the authors add labels so we can understand what the circles/bar heights mean intuitively. Also the legend for panel b appears to be in panel c?

The labels for Figure 1b are in the figure, but were placed too close to panel c to make it visually intuitive. We updated the figure to make it clear.

I am surprised that removing all the aromatic residues from HOXD4 IDR results in condensates forming at 5 μ M protein. Given the sequence composition, this protein should be highly soluble. At 10% W/V PEG8000 you’re at approximately 12.5 mM PEG in solution, and I can’t help but wonder if given the 2500x excess of PEG in this system you’re actually observing PEG-protein co-condensates? I’m not sure this actually changes the interpretation, but my hunch is that the dominating interaction here is the protein:PEG interaction, as opposed to the protein:protein interaction. This potential interpretation could be completely dismissed if labeled PEG were used and it was clear that no PEG was in the droplets. Absent that data (which I am not suggesting are necessary for publication, but it would be great, of course!), I think it may be prudent to raise this possible explanation, not that it alters the main message but does alter who a reader should think about the molecular driving forces in this experiment.

We thank the reviewer for these suggestions. We ran the HOXD4 IDR droplet assays with fluorescently labeled PEG. In summary, the PEG does not seem to contribute to droplets. The data are summarized below.

We ran droplet assays with wild type and AroLITE HOXD4 IDR at various concentrations in the presence of 10% PEG-rhodamine (i.e. Rhodamine-conjugated PEG 8000). We observed slight, but detectable enrichment of PEG-rhodamine in the droplets (**Reviewer Figure 1A** below). Subsequent experiments revealed that the enrichment is likely explained by rhodamine partitioning into the IDR droplets. First, we confirmed that the PEG-rhodamine alone did not form droplets at 10% concentration in the absence of protein (Panel B). Moreover, we ran droplet assays in which HOXD4 IDR proteins were incubated either with 10% PEG-rhodamine, or 10% unconjugated PEG and equimolar unconjugated rhodamine. We observed that the partitioning ratio of rhodamine was similar under both conditions, when tested for both wild type and AroLITE HOXD4 IDR (Panel C-D). As a further control, the partitioning ratio of the proteins into droplets were similar in the presence of conjugated or unconjugated PEG (Panel C-D). These results suggest that rhodamine itself has a propensity to partition into HOXD4 IDR droplets, but the PEG does not contribute to the formation of IDR droplets in the assays.

Reviewer Figure 1

Given the initial opinion of the reviewer that these data are not essential, we would prefer to keep them as part of the response letter. We have opted in “transparent peer review” and thus hope that the response letter with the data will be released as online supporting material with the manuscript.

While not essential, if an ARO-PERFECT style design for NFAT5 or EGR1 showed enhanced transcriptional output this would really help cement this principle of imperfections actually being detrimental to total transcriptional output.

This is a very interesting suggestion. NFAT5 and EGR1 have unusual, highly periodic IDRs already. Therefore, we tested the Reviewer's idea with a corollary experiment, by *reducing* the periodicity of the EGR1 IDR. We created two mutants. In the first one the aromatic residues are arranged in a single continuous patch, and in the second the aromatic residues are arranged in three patches. Both mutants had significantly lower activity than the wild type, highly periodic IDR in the luciferase reporter system. These data are included in **new Extended Data Fig. 5e-f**, and further strengthen the link between aromatic dispersion and transcriptional activity.

The subcellular localization of the AroPERFECT and AroPLUS HOXD4 variants is striking! Could this be explained by different protein expression levels? I realize the western in 2i is there to make the point that there's autoregulation, but I guess I'm wondering if you cranked up WT expression to get comparable protein levels as is seen for PERFECT and PLUS would analogous transcriptional profiles be obtained? I recognize that at the endogenous loci, we should expect expression to be comparable, so I'm wondering if there's now something altered with protein turnover). Given there is a potential degron between residues 142 and 148 (immediately C-terminal of the IDR) I wonder if there's some funky post-translational regulation going on?

Indeed, in the knock-in cells where the HOXD4 variants are knocked-in into the endogenous locus, the subcellular localization of the AroPERFECT and AroPLUS mutants could be due to differences in the protein levels. To test subcellular localization of the HOXD4 variants, we also generated knock-in cells using a PiggyBac transposon system. The PiggyBac transposon contained a Dox-inducible promoter (and not the endogenous HOXD4 promoter). We found that the nuclear granularity was significantly higher in the cells expressing the AroPERFECT and AroPLUS mutants compared to cells that expressing wild type HOXD4 at similar levels (**Fig. 3b-c**).

The breadth of systems exemplified in the second half of the paper is really remarkable, and a real strength of this work - regardless of the molecular origin (for which I can imagine there might be some questions/uncertainty) the actual phenotypic effect of changing aromatic periodicity is clear for all to see.

We thank the reviewer for positive comments!

I have a few final suggestions for the discussion. Firstly, the connection between these IDRs and phase separation is clearly made, but in reality, the data are largely agnostic in terms of whether or not phase separation is the actual underlying mechanism here. Perhaps, in reality, phase separation just correlates well another mechanism of interest? I agree that phase separation

seems the most parsimonious explanation, but I also think this potential agnosticism is a strength of the work. I would encourage the authors to discuss the fact that while phase separation does enable a mechanistic interpretation, it does not exclude a model where aromatic patterning is tuning the intermolecular interaction of the IDR with TFIID or some other component in the transcriptional machinery (without the need for bona fide phase separation). Secondly, Staller et al. have proposed an ‘acidic exposure’ model, whereby the balance of hydrophobic/aromatic residues is balanced by charged residues that prevent tight intramolecular interaction, priming aromatic residues to be exposed. In this model, increasing the number of aromatic residues beyond some threshold led to a drop in transcriptional output (see Staller et al. 2022). It would be great if the authors could comment on this interpretation, which I do think is largely in line with the authors’ findings here (note that the density of aromatic residues needed to see loss of activity was much higher than the authors test here, even in their AroPERFECT constructs). Perhaps the more evenly spaced aromatic residues further aid in their ability to engage with intermolecular targets, avoiding strong intramolecular interactions observed when aromatic residues are clustered together?

We thank the reviewer for these insightful suggestions. We generally agree with the reviewer on all accounts, and revised the discussion accordingly.

References: appropriate credit to previous work?

The enhanced liquidity upon more evenly-distributed aromatic residues mirrors conclusions from Holehouse et al Biochemistry 2021.

We now include reference to the cited paper in the revised Discussion.

Clarity and context: lucidity of abstract/summary, appropriateness of abstract, introduction and conclusions

The manuscript is extremely clear and well-written.

We thank the reviewer for the positive comments and useful comments and suggestions!

Reviewer #3:

Remarks to the Author:

Transcription factors (TFs) are the central players of gene regulation and often have an effector domain in addition to a DNA binding domain. Although the DNA binding domains are relatively well characterized, understanding the features of the effector domains, such as transcription activation domains, have been more challenging due to a lack of apparent structure or linear motif logic. The recent advances in this area have described features that correlate well with transactivation domain function, such as long disordered regions with acidic, hydrophobic, and aromatic residues, and built models with predictive power (1,2,3). The manuscript of Naderi & Magelhaes et al further investigates one of these features, the role of periodicity of aromatic residues, towards engineering stronger activation domains. While the manuscript describes a systematic computational approach, the follow up experiments do not provide sufficient support for the conclusions and the general relevance of the observations.

We thank the reviewer for comments, and apologize for an apparent lack of clarity of what the main model and key insights of the paper are.

We propose a model that there is a trade-off between the activity and specificity of human transcription factors. We found that this trade-off is encoded as submaximal aromatic dispersion in the IDRs of numerous TFs. We provide several lines of experimental and analytical evidence in support of this model.

Periodic IDRs appear different than the minimal activation domains characterized in the literature, including the papers cited by the Reviewer. The Sanborn et al., Staller et al., Erijman et al., studies identified short sequences that comprise ‘minimal’ activation domains, with an average size of ~40 amino acids (roughly the size used in the tiling experiments in those papers, see e.g. Figure 1E in Sanborn et al.). What all these papers reveal is that the short minimal activation domains are almost invariably embedded in long IDR sequences, and the IDR portions outside the minimal activation domains have no activity. On the other hand, one of the key insights in our study is that long IDR sequences can have activity that is distinct from that of minimal activation domains. For example, we show with tiling experiments of multiple TF IDRs, that the activity of periodic IDRs (100-200 amino acids) is NOT encoded in short sequences, as is the activity of the minimal activation domains. We therefore strongly disagree with the opinion that our study “further investigates one of these features”. Rather, the data we present are consistent with the view that long IDRs have a mode of activity different than that of well-studied minimal activation domains. In other words, the IDR sequences have important functional contributions encoded in the portions that are *not* minimal activation domains. One compelling illustration of this notion is that 94% (500/531) of periodic regions we identify do not overlap an annotated minimal activation domain, i.e., we identified traces of aromatic periodicity in portions of the IDRs where the minimal activation domain predicting algorithms *do not predict anything* (see e.g., **Extended Data Fig. 1c**).

In the revised manuscript we provide substantial new data that support our conclusions and the general relevance of our model. The new data are described in detail at the responses to the specific comments below.

Major points:

- The fact that a part of a protein's function can be improved is not novel as evolution "tinkers" towards needed function in terms of organismal fitness and not towards molecular optimality. The authors interpretation of the function of this lack of molecular optimality is purely speculative and does not inform about general rules in transcriptional regulation.

We agree with the Reviewer that is a well-known idea that evolution "tinkers". Using the Reviewer's analogy, we describe in this paper a mechanism *how* evolution tinkers. We also agree with the Reviewer that the model that multiple features of the same molecular entity evolve in a trade-off is well known. Indeed, the specificity-activity trade-off has been described for antibodies ³, for enzymes including the Cas9-nucleases ⁴, and for enhancers ^{2,5}. It has not however, to our knowledge, been proposed for transcription factors or their IDRs. We clarify this in the text (revised Introduction and Discussion sections).

As for general rules of transcriptional regulation, our data are consistent with the view that periodic IDRs appear to have a different mode of action than minimal activation domains extensively studied in the literature ⁶⁻⁹. We consider this an important new insight.

1. We show with tiling experiments that the activity of multiple periodic IDRs does not map to a short sequence portion (HOXD4: **Fig. 2e**; C/EBP α : **Fig 4f**; OCT4: **new Extended Data Fig. 5d**; EGR1: **new Extended Data Fig. 5f**; MYOD1: **new Extended Data Fig. 11b**). For C/EBP α , we include new data that optimizing aromatic dispersion of the region *outside* of the minimal activation domain (AD) enhances the activity of the IDR (**new Fig. 4g**). Moreover, the portion outside the AD could be complemented by a periodic FUS IDR portion when attached to the AD, both in the case of HOXD4 and C/EBP α (**new Fig. 2f, 4g**).
2. We provide new data using a cell-based condensate assay system that HOXD4 and C/EBP α periodic IDR condensates are more enriched in RNAPII CTD than the corresponding wild type IDR condensates (**new Fig. 3g-h, 4d-e**). This appears different than the Mediator recruitment described for the minimal activation domains in the papers that the Reviewer cite (see e.g., Sanborn et al)⁶. The notion of direct RNAPII recruitment is consistent with previous work showing that hydrogels formed by the highly periodic FUS IDR bind RNAPII CTD ¹⁰.

- It is not clear if the assays used in cellular differentiation are relevant to judge the in vivo function. In absence of a clearer understanding of molecular mechanisms of the studied domains this seems a rather speculative interpretation that builds on existing (and highly debated) models of phase separation.

We use the cellular reprogramming assays as these assays are one the most stringent functional tests for transcription factor function. We also note that we did knock-in wild type and periodic HOXD4 mutants into the endogenous HOXD4 locus. We found that the periodic HOXD4

proteins drove significantly higher production of their own RNA template, suggesting enhanced *in vivo* activity (**Fig. 3f**). Additional data on the clarification of the molecular mechanisms are described in the response to the comment above.

We stress for the Reviewer that our underlying hypothesis is that weak, multivalent *interactions* TF IDRs engage in are important for TF function. The *in vitro* droplet assays provide a system to measure such *interactions*, because such interactions can drive phase separation of the isolated proteins. In the manuscript, we establish a compelling relationship between the dynamics of such interactions (read out as enhanced liquid-like features of *in vitro* droplets using FRAP) and transcriptional activity for at least four TFs. We note that this not necessarily predict the effect of the mutations on condensates in cells. As such, any “debate” whether TF condensates are phase-separated or not *in vivo* has no relevance to the model investigated in this study.

We clarified the Abstract, Results and Discussion sections accordingly.

- Explaining the submaximal spacing between the aromatic residues with the same rationale used to explain the relevance of weak enhancer binding sites seems to be an over-reaching argument and an over-simplification, particularly in the absence of any molecular mechanism.

We propose a model that there is a trade-off between the activity and specificity of human transcription factors. Trade-offs between activity and specificity have been described for antibodies ³, for enzymes including the Cas9-nucleases ⁴, and for enhancers ^{2,5}, but not for TFs. We found that in TF IDRs, this trade-off is encoded as submaximal aromatic dispersion, and provide several lines of experimental and analytical evidence in support of this model. For example, increasing aromatic dispersion of the C/EBP α IDR enhanced transcriptional activity (**Fig. 4a, 4g**) and macrophage reprogramming (**Fig. 5c-e**), but led to generally stronger and more promiscuous genomic binding (**Fig. 5f-k**).

On the level of molecular mechanism, we described in detail above substantial new data further suggesting that periodic IDRs have a mode of action different than extensively studied minimal activation domains (see the response to the Reviewer’s summary, and major point #1). We already included data suggesting that the increased activity of periodic C/EBP α (and other TF) IDR is associated with the dynamics of self-association. In the revised manuscript we include new data that increased aromatic dispersion of the HOXD4 and C/EBP α IDRs results in enhanced RNAPII CTD recruitment in a cell-based condensate model system (**new Fig. 3g-h, 4d-e**). These experiments provide additional insights into the underlying mechanism.

We clarified these points in the text.

- The mutations of the peptide sequences could in principle change the stability of the protein. Whether such difference in total protein amount (Gal4-IDR) underlies the differential signal observed in the luciferase reporter assays needs to be tested. The same problem is indeed observed for the Hoxd4 experiments, where the gene expression analysis should be done on same-level-Hoxd4-expressing system. Similarly, the protein amounts should be controlled for the

other lineage specifying TF experiments to test whether the effect observed is simply due to higher protein concentrations.

We thank the Reviewer for the useful suggestion. We now include additional controls of protein levels. The data collectively suggest that the functional differences in the transactivation and reprogramming assays are not caused by differences in expression levels of the proteins.

We ran Western blots on cell lysates containing the GAL4 DBD-IDR proteins for multiple IDRs. We show that for six out of seven IDRs (HOXD4, C/EBP α , OCT4, PDX1, FOXA3, MYOD1), the periodic IDR construct is expressed at similar levels as the corresponding wild type IDR construct (**new Extended Data Fig. 4b, 4g, 5c, 7b, 11a**). Therefore, the increased activity of the periodic IDR mutants cannot be explained by higher protein levels in the reporter system. Interestingly, the transcriptionally inert LITE mutants in general tend to be expressed at higher levels than the wild type IDR constructs (**new Extended Data Fig. 4b, 4g, 5c, 7b, 11a**). Since the AroLITE mutants have generally no if any activity in the GAL4 DBD system, we can rule out that the lack of activity is caused by reduced expression level compared to corresponding wild type sequences.

We performed morphology and qRT-PCR analyses on PiggyBac cell lines that express wild type and periodic HOXD4 proteins in comparable levels. The results are consistent with the results presented before (**new Extended Data Fig. 6g-i**). The quantification of HOXD4 nuclear granularity was also performed on cells that express the proteins in similar levels (**Fig. 3b-c**).

For the reprogramming systems, we already included FACS data showing that the wild type and mutant C/EBP α proteins are expressed at comparable levels in the cell population (**Extended Data Fig. 7e**). In the revised manuscript, we include additional FACS data showing that the wild type and AroPERFECT IS15 C/EBP α proteins are expressed at similar levels in the cell lines that were used for the ChIP-Seq experiments (**new Extended Data Fig. 9e**). For NGN2, the Western blots are included in **Extended Data Fig. 10b**.

The data collectively suggest that the functional differences in the transactivation and reprogramming assays are not caused by differences in expression levels of the proteins.

Further points:

- The formulation of the aim and the conclusion of the study are not clear. The experiments shown in Ext. Dat. Fig 4 b-c suggest that a simple rule of periodicity in aromatic residues does not exist, let alone explaining the relevance of potential periodicity.

We clarified the aim and conclusions in the text (in the Introduction and Discussion sections), and included substantial new data that further clarify the sequence rules. We propose a model that there is a trade-off between the activity and specificity of human transcription factors. We found that this trade-off is encoded as submaximal aromatic dispersion in the IDRs of numerous TFs, and provide several lines of experimental and analytical evidence in support of this model.

We also agree with the Reviewer that aromatic dispersion is not the *only* important feature of TFs IDRs, and aromatic dispersion evolves in the context of other sequence features. We provide substantial new data supporting this model:

i) The nature of spacer sequences creates constraints for the effect of aromatic dispersion. This is supported by several lines of analytical (**new Extended Data Fig. 1d-f, 4e**) and experimental approaches (**Fig. 2g**).

ii) The mode of activity of periodic IDRs appears distinct from and complementary to the activity conferred by minimal activation domains, which also create some constraints. In support of this view, we experimentally separated the two modes, and show that the activity of IDRs can be enhanced in portions that do not contain minimal activation domains (**new Fig. 2f, 4g, Extended Data Fig. 5d-f, 11b**).

If the aim of the study is to engineer TFs with stronger activation potential (C/EBP α , Ngn2, Myod1) then adding one standard high performing activator domain on them would be logical rather than trying to find highly variant versions of the initial sequence, which not only diverges from functional relevance but even cause loss of function as in Hoxd4. If the aim of the study is to engineer peptide sequences to create “prion like” features, then testing relevant features such as Q/N repeats would be discussed. If the aim is to engineer new activator domains, then documented activator domain features such as hydrophobicity and negative charge distribution would be discussed.

The aim of the study is to test whether there is a trade-off between activity and specificity encoded in TF IDRs. We described the changes, clarifications and new data included in the revised manuscript above (e.g., at “Further points” #1, and in the response to the Reviewer’s summary, among others).

- A critical conclusion of the study is the function of TF-aggregates on gene regulation. In order to probe any such link, experimental set-ups as in (4) would be needed.

This is not a conclusion of the study. The main response and changes we made with regards to clarifying the aims and conclusions of the study are described in detail above.

We have performed experiments using the cell-based condensate system cited by the Reviewer, and include the data in the revised manuscript. We found that increased aromatic dispersion of the HOXD4 and C/EBP α IDR results in enhanced RNAPII CTD recruitment to IDR condensates in the cell-based condensate model system (**new Fig. 3g-h, 4d-e**). The results provide further insights into the mechanism of periodic aromatic regions.

We stress for the Reviewer that our underlying hypothesis is that weak, multivalent *interactions* TF IDRs engage in are important for TF function. The *in vitro* droplet assays provide a system to measure such *interactions*, because such interactions can drive phase separation of the isolated proteins. In the manuscript, we establish a compelling relationship between the dynamics of such

interactions (read out as enhanced liquid-like features of *in vitro* droplets using FRAP) and transcriptional activity for at least four TFs. We note that this not necessarily predict the effect of the mutations on condensates in cells. As such, any “debate” whether TF condensates are phase-separated or not *in vivo* has no relevance to the model investigated in this study.

We clarified the Abstract, Results and Discussion sections accordingly.

- The *Hoxd4* gene expression experiments (Figure 2g-i) need better controls/explanation/redone since the mentioned effect is the smaller component (PC2) which does not seem to distinguish the parental cells from the wild-type misexpression. To compare the effect of periodicity of the aromatic residues on gene expression, the use of the same-expression-level lines (as in Ext Dat Fig 6) would be better suited. The similar problems seem to exist for *Ngn2* and *Myod1* experiments. Protein concentration of the modified C/EBP α , *Ngn2* and *Myod1* relative to wild type needs to be documented since altered levels of these proteins can in principle result in the same observations made. Similarly, other perturbed functions (which seems to be many, given the PCA analysis and the Volcano plots) need better explanation.

We provide additional controls and clarifications in the revised manuscript.

First, neither PC1 nor PC2 really distinguishes between the expression profiles of the parental (HAP1) cell and the wild-type HOXD4-GFP knock-in cells, suggesting that the expression profiles are virtually identical. We interpret this as the knock-in procedure did not cause artificial changes in the function of HOXD4 locus, in other words, there is no “wild-type misexpression”.

As suggested by the Reviewer, we performed additional experiments using the PiggyBac cells which express wild type and periodic HOXD4 transgenes at similar levels. We observed morphology differences, and gene expression changes, consistent with gain-of-function effects of the periodic mutant (**new Extended Data Fig. 6g-i**). Furthermore, in the cell-based condensate tethering system requested by the Reviewer, we found that the condensates formed by the periodic HOXD4 IDR recruit significantly more RNA Polymerase II CTD (**new Fig. 3g-h**). These results collectively suggest that much of the effect of increased periodicity is likely not a result of gain of HOXD4 expression.

The Reviewer is indeed correct that there are similarities between the expression profiles of the endogenous AroPERFECT and AroPLUS HOXD4 knock-in cell lines and HOXD4 knock-out cells. These genes comprise a cluster that are controlled by PBX4/HOX heterodimers. In the knock-in cells therefore, there appears to be either insufficient levels of PBX4 to heterodimerize with the overproduced periodic HOXD4 mutants, or the mutations inhibit heterodimerization in cells (**Extended Data Fig. 6e**).

For the reprogramming systems, we already included FACS data showing that the wild type and mutant C/EBP α proteins are expressed at comparable levels in the cell population (**Extended Data Fig. 7e**). In the revised manuscript, we include additional FACS data showing that the wild type and AroPERFECT IS15 C/EBP α proteins are expressed at similar levels in the cell lines

that were used for the ChIP-Seq experiments (**new Extended Data Fig. 9e**). For NGN2, the Western blots are included in **Extended Data Fig. 10b**.

We explain all these points better in the Results and Methods sections of the revised manuscript.

- In Extended Data Figure 6.e, description in the figure legend (granularity vs signal intensity) seems to be swapped “The normalized signal intensity was calculated by dividing standard deviation of mEGFP signal of each nucleus by the corresponding mean mEGFP signal...Granularity scores of nuclei with corresponding mean nuclear mEGFP intensities.”

Thank you! We fixed the figure legends.

- In Extended Data Figure 6.e: the granularity test should be done on live samples since fixation can alter the signal (5)

We attempted the imaging experiments with live cells. The level of HOXD4 expression is too low to use the GFP fluorescence in the STED setup. Therefore, we needed to use anti-GFP immunofluorescence. We note that the cell-based condensate tethering experiments requested by the Reviewer, were performed using live cells (**new Fig. 3g-h**).

1. Sanborn, Adrian L., et al. "Simple biochemical features underlie transcriptional activation domain diversity and dynamic, fuzzy binding to Mediator." *Elife* 10 (2021)
2. Staller, Max V., et al. "Directed mutational scanning reveals a balance between acidic and hydrophobic residues in strong human activation domains." *Cell systems* 13.4 (2022)
3. Erijman, Ariel, et al. "A high-throughput screen for transcription activation domains reveals their sequence features and permits prediction by deep learning." *Molecular cell* 78.5 (2020)
4. Trojanowski, Jorge, et al. "Transcription activation is enhanced by multivalent interactions independent of phase separation." *Molecular cell* 82.10 (2022)
5. Irgen-Girol, Shawn, et al. "Fixation can change the appearance of phase separation in living cells." *Elife* 11 (2022)

References

- 1 Brodsky, S. *et al.* Intrinsically Disordered Regions Direct Transcription Factor In Vivo Binding Specificity. *Molecular cell* **79**, 459-471 e454, doi:10.1016/j.molcel.2020.05.032 (2020).
- 2 Farley, E. K. *et al.* Suboptimization of developmental enhancers. *Science* **350**, 325-328, doi:10.1126/science.aac6948 (2015).
- 3 Rabia, L. A., Desai, A. A., Jhajj, H. S. & Tessier, P. M. Understanding and overcoming trade-offs between antibody affinity, specificity, stability and solubility. *Biochem Eng J* **137**, 365-374, doi:10.1016/j.bej.2018.06.003 (2018).
- 4 Kim, Y. H. *et al.* Sniper2L is a high-fidelity Cas9 variant with high activity. *Nat Chem Biol* **19**, 972-980, doi:10.1038/s41589-023-01279-5 (2023).
- 5 Crocker, J. *et al.* Low affinity binding site clusters confer hox specificity and regulatory robustness. *Cell* **160**, 191-203, doi:10.1016/j.cell.2014.11.041 (2015).
- 6 Sanborn, A. L. *et al.* Simple biochemical features underlie transcriptional activation domain diversity and dynamic, fuzzy binding to Mediator. *eLife* **10**, doi:10.7554/eLife.68068 (2021).
- 7 Erijman, A. *et al.* A High-Throughput Screen for Transcription Activation Domains Reveals Their Sequence Features and Permits Prediction by Deep Learning. *Molecular cell* **78**, 890-902 e896, doi:10.1016/j.molcel.2020.04.020 (2020).
- 8 Staller, M. V. *et al.* A High-Throughput Mutational Scan of an Intrinsically Disordered Acidic Transcriptional Activation Domain. *Cell Syst* **6**, 444-455 e446, doi:10.1016/j.cels.2018.01.015 (2018).
- 9 Staller, M. V. *et al.* Directed mutational scanning reveals a balance between acidic and hydrophobic residues in strong human activation domains. *Cell Syst*, doi:10.1016/j.cels.2022.01.002 (2022).
- 10 Kwon, I. *et al.* Phosphorylation-regulated binding of RNA polymerase II to fibrous polymers of low-complexity domains. *Cell* **155**, 1049-1060, doi:10.1016/j.cell.2013.10.033 (2013).

Decision Letter, first revision:

*Please delete the link to your author homepage if you wish to forward this email to co-authors.

Dear Dr Hnisz,

Your manuscript, "Suboptimization of human transcription factors", has now been seen by three of our original referees, who are experts in transcriptional regulation and epigenetics (referee 1); biomolecular condensation (referee 2); and chromatin remodelling and sequencing (referee 3). As you will see from their comments (attached below) they find this work of interest, but have raised some important points. Apologies for the delay. We went to approach Reviewer #1 for their comments on Reviewer #3's current concerns, and have appended this further commenting into Reviewer #1's report accordingly. Thank you for your patience while we discussed this with the Reviewers and within the editorial team. Although we are also very interested in this study, we believe that their concerns should be addressed before we can consider publication in Nature Cell Biology.

Nature Cell Biology editors discuss the referee reports in detail within the editorial team, including the chief editor, to identify key referee points that should be addressed with priority, and requests that are overruled as being beyond the scope of the current study. To guide the scope of the revisions, I have listed these points below. We are committed to providing a fair and constructive peer-review process, so please feel free to contact me if you would like to discuss any of the referee comments further.

In particular, it would be essential to:

A) Appropriately tone down claims of potentially generalizable principles of "any biomolecule", and instead describe this in terms of transcription factors (Reviewers #3 and #1),

B) Appropriately describe the terms "trade off" and optimization, and discuss the potential role of TF IDRs in enhancing activation activity in the text, as per Reviewers #2 and #3.

B) Control for protein expression levels of your constructs with further data (Reviewers #1 and #3) as well as clarify gene expression levels and analysis (Reviewers #1 and #3).

D) Assess whether differentiation effects are seen in multiple clones (Reviewers #1 and #3).

E) All other referee concerns pertaining to strengthening existing data, providing controls, methodological details, clarifications and textual changes, should also be addressed.

F) Finally please pay close attention to our guidelines on statistical and methodological reporting (listed below) as failure to do so may delay the reconsideration of the revised manuscript. In particular please provide:

- a Supplementary Figure including unprocessed images of all gels/blots in the form of a multi-page pdf file. Please ensure that blots/gels are labeled and the sections presented in the figures are clearly

indicated.

We therefore invite you to take these points into account when revising the manuscript. In addition, when preparing the revision please:

- ensure that it conforms to our format instructions and publication policies (see below and <https://www.nature.com/nature/for-authors>).
- provide a point-by-point rebuttal to the full referee reports verbatim, as provided at the end of this letter.
- provide the completed Reporting Summary (found here <https://www.nature.com/documents/nr-reporting-summary.pdf>). This is essential for reconsideration of the manuscript and will be available to editors and referees in the event of peer review. For more information see <http://www.nature.com/authors/policies/availability.html> or contact me.

When submitting the revised version of your manuscript, please pay close attention to our [href="https://www.nature.com/nature-portfolio/editorial-policies/image-integrity">Digital Image Integrity Guidelines](https://www.nature.com/nature-portfolio/editorial-policies/image-integrity). and to the following points below:

Nature Cell Biology is committed to improving transparency in authorship. As part of our efforts in this direction, we are now requesting that all authors identified as 'corresponding author' on published papers create and link their Open Researcher and Contributor Identifier (ORCID) with their account on the Manuscript Tracking System (MTS), prior to acceptance. ORCID helps the scientific community achieve unambiguous attribution of all scholarly contributions. You can create and link your ORCID from the home page of the MTS by clicking on 'Modify my Springer Nature account'. For more information please visit www.springernature.com/orcid.

This journal strongly supports public availability of data. Please place the data used in your paper into a public data repository, or alternatively, present the data as Supplementary Information. If data can

only be shared on request, please explain why in your Data Availability Statement, and also in the correspondence with your editor. Please note that for some data types, deposition in a public repository is mandatory - more information on our data deposition policies and available repositories appears below.

[Redacted]

We would like to receive the revision within four weeks. If submitted within this time period, reconsideration of the revised manuscript will not be affected by related studies published elsewhere, or accepted for publication in Nature Cell Biology in the meantime. We would be happy to consider a revision even after this timeframe, but in that case we will consider the published literature at the time of resubmission when assessing the file.

We hope that you will find our referees' comments, and editorial guidance helpful. Please do not hesitate to contact me if there is anything you would like to discuss.

Best wishes,

Daryl

Daryl Jason Verzosa David, PhD

Senior Editor, Nature Cell Biology
Nature Portfolio

Heidelberger Platz 3, 14197 Berlin, Germany
Email: daryl.david@nature.com
ORCID: <https://orcid.org/0000-0002-9253-4805>

Reviewers' Comments:

Reviewer #1:

Remarks to the Author:

The authors have fully and adequately addressed my comments on the first submission. The revised manuscript is much clearer in the messaging, and the conceptual advances, in particular the concept

of sub-optimal TF amino acid sequences, are now clearly presented. The experimental and analytical evidence has improved. I now recommend acceptance.

Reviewer #1's additional comments on Reviewer #3's current concerns:

I have now have a chance to consider reviewer #3's arguments against the publication of the manuscript. While I feel that the points raised by reviewer #3 are in general well intended and aimed to further enhance the rigor of the manuscript, I do believe that the authors conclusion is well supported by evidence provided, and the statements in the current manuscript are generally justified (with perhaps one exception pointed out by reviewer #3). Please see my detailed assessment below:

Reviewer #3's comments:

While the revised the manuscript claims to address some of the concerns of the reviewers, the main conclusions that would justify highly visible publication remain speculative, although stated very strongly and in very general terms, and in view of this reviewer are insufficiently supported.

Reviewer #1's further comments: My opinion is the opposite. I believe that the authors made a strong case for the suboptimization of TFs and regulation of their activities genome-wide.

Reviewer #3's comments:

The manuscript at its current state somewhat constructs relevance around the TF specificity argument, stating that this resembles weak binding sites in enhancer elements. Changes in DNA sequence can indeed change the specificity for TF binding. For example, if a genomic region is manipulated to have a higher affinity for a certain TF, now the region starts to be bound by this factor even at lower concentration leading to activation in other cell types.

The manuscript shows interesting examples of how distribution of prolines may potentially affect the transcription activation potential of transcription factor IDRs. However, there is still insufficient evidence that specificity is affected by the occasional increase of transcription activation potential by modulating proline distributions in these TFs. Therefore, with its current overinterpretation, I do not consider the manuscript suitable for publication.

Reviewer #1's further comments: I believe that the authors provided very compelling evidence arguing for the role of periodicity and distribution of aromatic residues in IDR formation and TF activation domain functions. The authors assessed this using HOXD4, OCT4, PDX1, FOXA3, EGR1, C/EBPa, NGN2, MYOD1, in multiple experimental regimes, and some in in vivo settings. The breadth of the experimental assays and the assays used to support the central theme of the study certainly meets, and well exceeds, similar articles in top tier journals.

Reviewer #3's comments:

The specific concerns are listed below:

- Since the manuscript is originally focusing on the manipulation of the transcription output potential of the TFs, it would be more adequate if the title suggests something about changing or optimizing the

transcription output potential of human transcription factors' rather than 'suboptimization of human transcription factors'. The phrasing of the title simply refer to another paper discussing that the enhancers have mostly weak affinity for TF binding in relation to establishing binding specificity. The manuscript here does not originally tackle the specificity question and has not enough breadth to justify that as a main point/conclusion.

Reviewer #1's further comments: I think that the authors refocuses appropriately on an exciting new concept, the suboptimization of TF, in the revision, after considering criticisms and suggestions from reviewers comments on the first submission. I am of the view that the authors' experimental and analyses do support the conclusions.

Reviewer #3's comments:

- in the abstract, the use of the term "any biomolecule" is not justified.

Reviewer #1's further comments: I agree that the term "any biomolecule" is an overstatement and should be removed. Later the authors also used the term "other biopolymers", which has the same effect. I suggest that the authors tone down on these statements, and perhaps focus on TF proteins.

Reviewer #3's comments:

- the use of the term 'trade-off' argues that it is disadvantageous for some functions which is not justified in the context of TF's function: the ability to regulate gene expression in correct time, space and dose.

Reviewer #1's further comments: I am not concerned about this.

Reviewer #3's comments:

- The referral to the statement 'suggesting that TF IDRs may contribute to transcriptional activity and also to binding specificity in vivo' to support the interpretations of the data seems speculative. With the same logic one would say changing protein concentration changes binding specificity, simply due to the observation that you would get lower signal and dynamic range in a chip-seq experiment with lower protein concentration. Indeed weak enhancer motifs have been suggested to exist in order to respond to a TF upon a certain concentration (e.g. <https://www.biorxiv.org/content/10.1101/2022.05.27.493636v1>, <https://pubmed.ncbi.nlm.nih.gov/27155014/>).

Reviewer #1's further comments: The statement referred to here is appropriate in my view.

Reviewer #3's comments:

-The section 'increasing aromatic dispersion in TF IDRs enhances transactivation' nicely shows that the effects seen depend on other features such as the spacer sequences and the charge of the backbone, which seems to have higher effect than the prolines. It seems that there would be more potent ways to increase TF activity, modulate transcription potential and allow rational design of a very active TF if ever needed.

Reviewer #1's further comments: I do not think that this point is very relevant to the manuscript.

Reviewer #3's comments:

- The RNA-seq experiment done in the section "Evidence for gain-of-function of periodic HOXD4 mutants in vivo" is based on cell-lines that have different levels of hoxd4 protein. Therefore, the changes cannot be attributed to proline periodicity, this needs to be explicitly stated in the section. While the efforts on the extended data figure 6 in principle aims to tackle part of this problem, the presented data comes across as having discrepancies. It seems that aroplus has different levels of hox-gfp than the wildtype (panel f) which is opposite in FACS analysis (panel h). The FACS also does not seem to reflect the representative images in panel g which has for example GFP negative cells in Aroplus that are not observed in FACS profile. In addition, the gene expression changes in panel i are not shown for aroperfect but only for aroplus. Also, Extended Data Figure 6g, needs same LUT and visualization parameters on the representative images.

Reviewer #1's further comments: I agree that it is important to check the protein levels in these experiments to make sure that they are comparable. The authors should address this concern.

Reviewer #3's comments:

- The hoxd4 rna-seq section concludes both loss of function and gain of function by solely looking at the up/down regulated genes in these cell-lines, which the authors claim are the direct hox targets. The gene expression changes are interpreted as decreased-expression being about disruption of interactions with pbx and increased-expression being about increased transactivation function of the engineered proteins. None of these are either shown or tested, and comes across speculative. The variations in the experiment rather show that by changing protein sequence other features, such as protein interactions and protein stability, are changed. There could even be more changes such as posttranslational modifications, folding etc, making the comparisons uninformative.

Reviewer #1's further comments: This is a minor point that the authors ought to be able to address by rephrasing the statements.

Reviewer #3's comments:

- Extended Data Figure 6.e: 'Cluster 1 and 2 represent the HOXD4 knockout specific genes': these 2 clusters rather seem to be shared with knockout, aroplus and aroperfect? Cluster 5 does not seem to be shared between knockout, aroperfect and aroplus. It is more like opposite response in wt vs aroplus and aroperfect. Also the statement: 'Cluster 4 represents the genes with differential expression in AroPLUS.' seems not true, they are shared in knockout/aroplus and aroperfect. Could it be that there is a wrong labeling/naming of the clusters?

Reviewer #1's further comments: This is a minor point that the authors ought to be able to address with further clarification.

Reviewer #3's comments:

- Extended Data Figure 6f vs 6i: Is IFI16 gene same or increased in aroplus?

Reviewer #1's further comments: This is a minor point that the authors ought to be able to address with further clarification.

Reviewer #3's comments:

- Figure 3h (and same experiment type done in other sections): for this type of non-normal distribution, it is appropriate to use median, rather than mean. Also, a non-parametric test, e.g. Wilcoxon suits better. The imaging and quantification of this section in methods needs explanation (microscopy: widefield/confocal, image processing: center of the signal on two channels/ projection type/background subtraction, bleed-through correction from cfp to yfp etc.) The signal needs to be calculated also taking CFP-lacI intensity into account, which seems to be different for example in Figure 4d.

Reviewer #1's further comments: This is a minor point that the authors ought to be able to address with additional analysis to improve rigor.

Reviewer #3's comments:

- Figure 5f shows that the regions labeled as Aroperfect15-specific have also binding in wildtype. They may simply be under the threshold of peak calling they used but the presence of the signal is clear. This is also true for other ChIPs shown in the manuscript. This is a strong indication that the 'specificity' is not the main phenomenon here.

Reviewer #1's further comments: This is a minor point that the authors ought to be able to address with further analysis that improves rigor.

Reviewer #3's comments:

- Does AroLITE C/EBPa not localize to nucleoplasm (Ext Fig 9d)? Does Extended Data Figure 9d suggest that the levels of aroperfect are higher, or the representative images do not reflect the quantification? It seems that the population with no gfp is absent in aroperfect in panel e.

- Extended figure 10: arolite seems much higher than wt in panel d, which is opposite in panel e.

Reviewer #1's further comments: This is a minor point that the authors ought to be able to address with further clarification.

Reviewer #3's comments:

- In the quantification of differentiation for ngn2 experiments, aroperfect is claimed to be better at 'differentiating' as opposed to chip-seq pca at Extended figure 10 h. The effect is also seen in Extended figure 10 k(wild type more differential than parental as compared to aroperfect, cluster 5&6 shows more similarity between arolite and aroperfect). There are clear differences in heterogeneity as seen by non-differentiating cell populations in the experiments(figure6d). These all could be clonal effects. Thus, the phenotype conclusions in these comparisons seem to be inconclusive and uninformative.

Reviewer #1's further comments: It is not fare to dismiss the observations as clonal effects. If the authors have reproducible observations from multiple clones, that would help address the concerns.

Reviewer #3's comments:

- The concerns about the protein levels need better addressing and quantification across experiments. For example, with the tools the authors present, it is easy to perform FACS or image analysis of flag

staining on Myod induction lines. Many things can lead to loss of function, thus maybe less interesting but increased yet specific potency is often more relevant thus needs for a proof for controlling other variables such as same or lower protein levels for aroperfect/plus.

Reviewer #1's further comments: This is a good point and additional QC data would be very helpful.

Reviewer #2:

Remarks to the Author:

The authors have done an excellent job addressing my queries as well as those of the the other reviewers.

My final question pertains to the revised title: "Suboptimization of human transcription factors", which I worry may be so vague people won't understand WHAT is suboptimized; perhaps

"Suboptimization of disordered regions in human transcription factors" would strike a balance between being punchy but making clear what the core of the paper is about? This is just a suggestion, though!

Reviewer #3:

Remarks to the Author:

While the revised the manuscript claims to address some of the concerns of the reviewers, the main conclusions that would justify highly visible publication remain speculative, although stated very strongly and in very general terms, and in view of this reviewer are insufficiently supported.

The manuscript at its current state somewhat constructs relevance around the TF specificity argument, stating that this resembles weak binding sites in enhancer elements. Changes in DNA sequence can indeed change the specificity for TF binding. For example, if a genomic region is manipulated to have a higher affinity for a certain TF, now the region starts to be bound by this factor even at lower concentration leading to activation in other cell types.

The manuscript shows interesting examples of how distribution of prolines may potentially affect the transcription activation potential of transcription factor IDRs. However, there is still insufficient evidence that specificity is affected by the occasional increase of transcription activation potential by modulating proline distributions in these TFs. Therefore, with its current overinterpretation, I do not consider the manuscript suitable for publication.

The specific concerns are listed below:

- Since the manuscript is originally focusing on the manipulation of the transcription output potential of the TFs, it would be more adequate if the title suggests something about changing or optimizing the transcription output potential of human transcription factors' rather than 'suboptimization of human transcription factors'. The phrasing of the title simply refer to another paper discussing that the enhancers have mostly weak affinity for TF binding in relation to establishing binding specificity. The manuscript here does not originally tackle the specificity question and has not enough breadth to justify that as a main point/conclusion.

- in the abstract, the use of the term "any biomolecule" is not justified.
- the use of the term 'trade-off' argues that it is disadvantageous for some functions which is not justified in the context of TF's function: the ability to regulate gene expression in correct time, space and dose.
- The referral to the statement 'suggesting that TF IDRs may contribute to transcriptional activity and also to binding specificity in vivo' to support the interpretations of the data seems speculative. With the same logic one would say changing protein concentration changes binding specificity, simply due to the observation that you would get lower signal and dynamic range in a chip-seq experiment with lower protein concentration. Indeed weak enhancer motifs have been suggested to exist in order to respond to a TF upon a certain concentration (e.g. <https://www.biorxiv.org/content/10.1101/2022.05.27.493636v1>, <https://pubmed.ncbi.nlm.nih.gov/27155014/>).
- The section 'increasing aromatic dispersion in TF IDRs enhances transactivation' nicely shows that the effects seen depend on other features such as the spacer sequences and the charge of the backbone, which seems to have higher effect than the prolines. It seems that there would be more potent ways to increase TF activity, modulate transcription potential and allow rational design of a very active TF if ever needed.
- The RNA-seq experiment done in the section "Evidence for gain-of-function of periodic HOXD4 mutants in vivo" is based on cell-lines that have different levels of hoxd4 protein. Therefore, the changes cannot be attributed to proline periodicity, this needs to be explicitly stated in the section. While the efforts on the extended data figure 6 in principle aims to tackle part of this problem, the presented data comes across as having discrepancies. It seems that aroplus has different levels of hox-gfp than the wildtype (panel f) which is opposite in FACS analysis (panel h). The FACS also does not seem to reflect the representative images in panel g which has for example GFP negative cells in Aroplus that are not observed in FACS profile. In addition, the gene expression changes in panel i are not shown for aroperfect but only for aroplus. Also, Extended Data Figure 6g, needs same LUT and visualization parameters on the representative images.
- The hoxd4 rna-seq section concludes both loss of function and gain of function by solely looking at the up/down regulated genes in these cell-lines, which the authors claim are the direct hox targets. The gene expression changes are interpreted as decreased-expression being about disruption of interactions with pbx and increased-expression being about increased transactivation function of the engineered proteins. None of these are either shown or tested, and comes across speculative. The variations in the experiment rather show that by changing protein sequence other features, such as protein interactions and protein stability, are changed. There could even be more changes such as posttranslational modifications, folding etc, making the comparisons uninformative.
- Extended Data Figure 6.e: 'Cluster 1 and 2 represent the HOXD4 knockout specific genes': these 2 clusters rather seem to be shared with knockout, aroplus and aroperfect? Cluster 5 does not seem to be shared between knockout, aroperfect and aroplus. It is more like opposite response in wt vs aroplus and aroperfect. Also the statement: 'Cluster 4 represents the genes with differential expression in AroPLUS.' seems not true, they are shared in knockout/aroplus and aroperfect. Could it be that there is a wrong labeling/naming of the clusters?

- Extended Data Figure 6f vs 6i: Is IFI16 gene same or increased in aroplus?
- Figure 3h (and same experiment type done in other sections): for this type of non-normal distribution, it is appropriate to use median, rather than mean. Also, a non-parametric test, e.g. Wilcoxon suits better. The imaging and quantification of this section in methods needs explanation (microscopy: widefield/confocal, image processing: center of the signal on two channels/ projection type/background subtraction, bleed-through correction from cfp to yfp etc.) The signal needs to be calculated also taking CFP-lacI intensity into account, which seems to be different for example in Figure 4d.
- Figure 5f shows that the regions labeled as Aroperfect15-specific have also binding in wildtype. They may simply be under the threshold of peak calling they used but the presence of the signal is clear. This is also true for other ChIPs shown in the manuscript. This is a strong indication that the 'specificity' is not the main phenomenon here.
- Does AroLITE C/EBPa not localize to nucleoplasm (Ext Fig 9d)? Does Extended Data Figure 9d suggest that the levels of aroperfect are higher, or the representative images do not reflect the quantification? It seems that the population with no gfp is absent in aroperfect in panel e.
- Extended figure 10: arolite seems much higher than wt in panel d, which is opposite in panel e.
- In the quantification of differentiation for ngn2 experiments, aroperfect is claimed to be better at 'differentiating' as opposed to chip-seq pca at Extended figure 10 h. The effect is also seen in Extended figure 10 k(wild type more differential than parental as compared to aroperfect, cluster 5&6 shows more similarity between arolite and aroperfect). There are clear differences in heterogeneity as seen by non-differentiating cell populations in the experiments(figure6d). These all could be clonal effects. Thus, the phenotype conclusions in these comparisons seem to be inconclusive and uninformative.
- The concerns about the protein levels need better addressing and quantification across experiments. For example, with the tools the authors present, it is easy to perform FACS or image analysis of flag staining on Myod induction lines. Many things can lead to loss of function, thus maybe less interesting but increased yet specific potency is often more relevant thus needs for a proof for controlling other variables such as same or lower protein levels for aroperfect/plus.

GUIDELINES FOR SUBMISSION OF NATURE CELL BIOLOGY ARTICLES

READABILITY OF MANUSCRIPTS – Nature Cell Biology is read by cell biologists from diverse backgrounds, many of whom are not native English speakers. Authors should aim to communicate

their findings clearly, explaining technical jargon that might be unfamiliar to non-specialists, and avoiding non-standard abbreviations. Titles and abstracts should concisely communicate the main findings of the study, and the background, rationale, results and conclusions should be clearly explained in the manuscript in a manner accessible to a broad cell biology audience. Nature Cell Biology uses British spelling.

ARTICLE FORMAT

ABSTRACT – should not exceed 150 words and should be unreferenced. This paragraph is the most visible part of the paper and should briefly outline the background and rationale for the work, and accurately summarize the main results and conclusions. Key genes, proteins and organisms should be specified to ensure discoverability of the paper in online searches.

TEXT – the main text consists of the Introduction, Results, and Discussion sections and must not exceed 3500 words including the abstract. The Introduction should expand on the background relating to the work. The Results should be divided in subsections with subheadings, and should provide a concise and accurate description of the experimental findings. The Discussion should expand on the findings and their implications. All relevant primary literature should be cited, in particular when discussing the background and specific findings.

REFERENCES – are limited to a total of 70 in the main text and Methods combined,. They must be numbered sequentially as they appear in the main text, tables and figure legends and Methods and must follow the precise style of Nature Cell Biology references. References only cited in the Methods should be numbered consecutively following the last reference cited in the main text. References only

associated with Supplementary Information (e.g. in supplementary legends) do not count toward the total reference limit and do not need to be cited in numerical continuity with references in the main text. Only published papers can be cited, and each publication cited should be included in the numbered reference list, which should include the manuscript titles. Footnotes are not permitted.

Methods should be written concisely, but should contain all elements necessary to allow interpretation and replication of the results. As a guideline, Methods sections typically do not exceed 3,000 words. The Methods should be divided into subsections listing reagents and techniques. When citing previous methods, accurate references should be provided and any alterations should be noted. Information must be provided about: antibody dilutions, company names, catalogue numbers and clone numbers for monoclonal antibodies; sequences of RNAi and cDNA probes/primers or company names and catalogue numbers if reagents are commercial; cell line names, sources and information on cell line identity and authentication. Animal studies and experiments involving human subjects must be reported in detail, identifying the committees approving the protocols. For studies involving human subjects/samples, a statement must be included confirming that informed consent was obtained. Statistical analyses and information on the reproducibility of experimental results should be provided in a section titled "Statistics and Reproducibility".

All Nature Cell Biology manuscripts submitted on or after March 21 2016, must include a Data availability statement as a separate section after Methods but before references, under the heading "Data Availability". For Springer Nature policies on data availability see <http://www.nature.com/authors/policies/availability.html>; for more information on this particular policy see <http://www.nature.com/authors/policies/data/data-availability-statements-data-citations.pdf>. The Data availability statement should include:

- Accession codes for primary datasets (generated during the study under consideration and designated as "primary accessions") and secondary datasets (published datasets reanalysed during the study under consideration, designated as "referenced accessions"). For primary accessions data should be made public to coincide with publication of the manuscript. A list of data types for which submission to community-endorsed public repositories is mandated (including sequence, structure, microarray, deep sequencing data) can be found here <http://www.nature.com/authors/policies/availability.html#data>.
- Unique identifiers (accession codes, DOIs or other unique persistent identifier) and hyperlinks for datasets deposited in an approved repository, but for which data deposition is not mandated (see here for details <http://www.nature.com/sdata/data-policies/repositories>).
- At a minimum, please include a statement confirming that all relevant data are available from the authors, and/or are included with the manuscript (e.g. as source data or supplementary information), listing which data are included (e.g. by figure panels and data types) and mentioning any restrictions on availability.
- If a dataset has a Digital Object Identifier (DOI) as its unique identifier, we strongly encourage including this in the Reference list and citing the dataset in the Methods.

We recommend that you upload the step-by-step protocols used in this manuscript to the Protocol Exchange. More details can be found at www.nature.com/protocolexchange/about.

DISPLAY ITEMS – main display items are limited to 6-8 main figures and/or main tables. For Supplementary Information see below.

FIGURES – Colour figure publication costs \$395 per colour figure. All panels of a multi-panel figure must be logically connected and arranged as they would appear in the final version. Unnecessary figures and figure panels should be avoided (e.g. data presented in small tables could be stated briefly in the text instead).

All imaging data should be accompanied by scale bars, which should be defined in the legend. Cropped images of gels/blots are acceptable, but need to be accompanied by size markers, and to retain visible background signal within the linear range (i.e. should not be saturated). The boundaries of panels with low background have to be demarked with black lines. Splicing of panels should only be considered if unavoidable, and must be clearly marked on the figure, and noted in the legend with a statement on whether the samples were obtained and processed simultaneously. Quantitative comparisons between samples on different gels/blots are discouraged; if this is unavoidable, it has to be performed for samples derived from the same experiment with gels/blots were processed in parallel, which needs to be stated in the legend.

- We do not recommend using Adobe Photoshop for designing figures, but we can accept Photoshop generated (.PSD or .TIFF) files only if each element included in the figure (text, labels, pictures, graphs, arrows and scale bars) are on separate layers. All text should be editable in 'type layers' and line-art such as graphs and other simple schematics should be preserved and embedded within 'vector

smart objects' - not flattened raster/bitmap graphics.

Regardless of format, all figures must be vector graphic compatible files, not supplied in a flattened raster/bitmap graphics format, but should be fully editable, allowing us to highlight/copy/paste all text and move individual parts of the figures (i.e. arrows, lines, x and y axes, graphs, tick marks, scale bars etc). The only parts of the figure that should be in pixel raster/bitmap format are photographic images or 3D rendered graphics/complex technical illustrations.

Unprocessed scans of all key data generated through electrophoretic separation techniques need to be presented in a supplementary figure that should be labeled and numbered as the final supplementary figure, and should be mentioned in every relevant figure legend. This figure does not count towards the total number of figures and is the only figure that can be displayed over multiple pages, but should be provided as a single file, in PDF or TIFF format. Data in this figure can be displayed in a relatively informal style, but size markers and the figures panels corresponding to the presented data must be indicated.

The total number of Supplementary Figures (not including the “unprocessed scans” Supplementary Figure) should not exceed the number of main display items (figures and/or tables (see our Guide to Authors and March 2012 editorial <http://www.nature.com/ncb/authors/submit/index.html#suppinfo>; <http://www.nature.com/ncb/journal/v14/n3/index.html#ed>). No restrictions apply to Supplementary Tables or Videos, but we advise authors to be selective in including supplemental data.

GUIDELINES FOR EXPERIMENTAL AND STATISTICAL REPORTING

REPORTING REQUIREMENTS – We ask authors to complete a Reporting Summary that collects information on experimental design and reagents. We hope this will aid in your evaluation of the paper. The Reporting Summary can be found here (<https://www.nature.com/documents/nr-reporting-summary.pdf>) Please note that these forms are dynamic ‘smart pdfs’ and must therefore be downloaded and completed in Adobe Reader. We will then flatten them for ease of use. If you would like to reference the guidance text as you complete the template, please access these flattened versions at <http://www.nature.com/authors/policies/availability.html>.

We strongly recommend the presentation of source data for graphical and statistical analyses as a separate Supplementary Table, and request that source data for all independent repeats are provided when representative experiments of multiple independent repeats, or averages of two independent experiments are presented. This supplementary table should be in Excel format, with data for different figures provided as different sheets within a single Excel file. It should be labelled and numbered as one of the supplementary tables, titled “Statistics Source Data”, and mentioned in all relevant figure legends.

Author Rebuttal, first revision:

Reviewers' Comments:

Author responses are in blue font

Reviewer #1:

Remarks to the Author:

The authors have fully and adequately addressed my comments on the first submission. The revised manuscript is much clearer in the messaging, and the conceptual advances, in particular the concept of sub-optimal TF amino acid sequences, are now clearly presented. The experimental and analytical evidence has improved. I now recommend acceptance.

Reviewer #1's additional comments on Reviewer #3's current concerns:

I have now had a chance to consider reviewer #3's arguments against the publication of the manuscript. While I feel that the points raised by reviewer #3 are in general well intended and aimed to further enhance the rigor of the manuscript, I do believe that the authors conclusion is well supported by evidence provided, and the statements in the current manuscript are generally justified (with perhaps one exception pointed out by reviewer #3). Please see my detailed assessment below:

We thank the Reviewer for the significant time and effort with helping us improve the manuscript. The typical review process puts significant burden on reviewers having to evaluate a large amount of data, the appropriate literature context and multiple divergent opinions. We are grateful for the Reviewer for going the extra mile helping us and the editorial team navigate the review process, and most importantly, helping us publish a better version of the science for the benefit of the readers.

Reviewer #3's comments:

While the revised the manuscript claims to address some of the concerns of the reviewers, the main conclusions that would justify highly visible publication remain speculative, although stated very strongly and in very general terms, and in view of this reviewer are insufficiently supported.

Reviewer #1's further comments: My opinion is the opposite. I believe that the authors made a strong case for the suboptimization of TFs and regulation of their activities genome-wide.

We agree with Reviewer 1. As detailed below at several comments, we present substantially more data than published papers that make similar claims as we do. Even so, we removed the most general claim that appear to have been the most contentious.

Reviewer #3's comments:

The manuscript at its current state somewhat constructs relevance around the TF specificity argument, stating that this resembles weak binding sites in enhancer elements. Changes in DNA sequence can indeed change the specificity for TF binding. For example, if a genomic region is manipulated to have a higher affinity for a certain TF, now the region starts to be bound by this factor even at lower concentration leading to activation in other cell types. The manuscript shows interesting examples of how distribution of prolines may potentially affect the transcription

activation potential of transcription factor IDRs. However, there is still insufficient evidence that specificity is affected by the occasional increase of transcription activation potential by modulating proline distributions in these TFs. Therefore, with its current overinterpretation, I do not consider the manuscript suitable for publication.

Reviewer #1's further comments: I believe that the authors provided very compelling evidence arguing for the role of periodicity and distribution of aromatic residues in IDR formation and TF activation domain functions. The authors assessed this using HOXD4, OCT4, PDX1, FOXA3, EGR1, C/EBP α , NGN2, MYOD1, in multiple experimental regimes, and some in in vivo settings. The breadth of the experimental assays and the assays used to support the central theme of the study certainly meets, and well exceeds, similar articles in top tier journals.

We agree with Reviewer 1. Making the case for reduced specificity of optimized TFs and suggesting a trade-off between activity and specificity (which appears to be the main concern for Reviewer 3), our manuscript includes substantially more data than papers in the published literature that proposed similar ideas. The original reference that introduced the suboptimization concept for enhancer elements included dissection of *one* enhancer element, and demonstration of loss of tissue specificity for *one* enhancer variant using *one* type of assay (Farley et al., Science 2015). Since then, the Farley lab published *one* more enhancer example experimentally dissected with *one* type of assay (Jindal et al., 2023). The other foundational reference for suboptimization included experimental dissection of *one* enhancer element in Drosophila embryos with *one* type of assay (Crocker et al., 2015). We experimentally scrutinized over half a dozen transcription factors. Data consistent with altered gene specificity include i) qPCR data for optimizing HOXD4, ii) RNA-Seq, scRNA-Seq, CHIP-Seq and reporter data for optimizing C/EBP α , iii) RNA-Seq and CHIP-Seq data for optimizing NGN2, and iv) RNA-Seq data for optimizing MYOD1. For C/EBP α we show data that the optimized protein more strongly binds low-affinity sites than the wild type protein when they are expressed at similar levels (Figure 5f-k)! Of course, the CHIP-Seq data provide more direct mechanistic explanation for the change in specificity than RNA-readouts. In summary, the level of evidence we provide is far beyond the most important literature references.

Reviewer #3's comments:

The specific concerns are listed below:

- Since the manuscript is originally focusing on the manipulation of the transcription output potential of the TFs, it would be more adequate if the title suggests something about changing or optimizing the transcription output potential of human transcription factors' rather than 'suboptimization of human transcription factors'. The phrasing of the title simply refer to another paper discussing that the enhancers have mostly weak affinity for TF binding in relation to establishing binding specificity. The manuscript here does not originally tackle the specificity question and has not enough breadth to justify that as a main point/conclusion.

Reviewer #1's further comments: I think that the authors refocuses appropriately on an exciting new concept, the suboptimization of TF, in the revision, after considering criticisms and

suggestions from reviewers comments on the first submission. I am of the view that the authors' experimental and analyses do support the conclusions.

We agree with Reviewer 1. As detailed above, we present substantially more data than published papers that make similar claims as we do. Also, as explained below at the response to Reviewer 2, we changed the title to be more specific.

Reviewer #3's comments:

- in the abstract, the use of the term “any biomolecule” is not justified.

Reviewer #1's further comments: I agree that the term “any biomolecule” is an overstatement and should be removed. Later the authors also used the term “other biopolymers”, which has the same effect. I suggest that the authors tone down on these statements, and perhaps focus on TF proteins.

We removed the statements on “any biomolecules/biopolymers” from the abstract, results and discussion sections, and instead focus on transcription factors.

Reviewer #3's comments:

- the use of the term 'trade-off' argues that it is disadvantageous for some functions which is not justified in the context of TF's function: the ability to regulate gene expression in correct time, space and dose.

Reviewer #1's further comments: I am not concerned about this.

The term trade-off refers to the idea that multiple features of TFs are inherently linked, and as a consequence, changing one leads to changes in the other. The term suboptimization means that as the consequence of the trade-off, neither of two features are optimal in a TF, whose *overall* phenotypic effect is optimized by evolution, see e.g: (Shoval et al., 2012). We explained suboptimization and trade-off in the text (last paragraph of the introduction).

Reviewer #3's comments:

- The referral to the statement 'suggesting that TF IDRs may contribute to transcriptional activity and also to binding specificity in vivo' to support the interpretations of the data seems speculative. With the same logic one would say changing protein concentration changes binding specificity, simply due to the observation that you would get lower signal and dynamic range in a chip-seq experiment with lower protein concentration. Indeed weak enhancer motifs have been suggested to exist in order to respond to a TF upon a certain concentration (e.g. <https://www.biorxiv.org/content/10.1101/2022.05.27.493636v1>, <https://pubmed.ncbi.nlm.nih.gov/27155014/>).

Reviewer #1's further comments: The statement referred to here is appropriate in my view.

The cited statement is in the introduction and refers to the published papers that the sentence cites. The sentence does not refer to interpretation of our data, but of the published literature. We note again, that in our data, for C/EBP α for example, we show that the optimized TF does bind more strongly to low-affinity sites than the wild type protein when they are expressed at similar levels (Figure 5f-k)! Reviewer 3 appears to set this as a standard for discussing “binding specificity” and our data clearly meet it.

Reviewer #3's comments:

-The section 'increasing aromatic dispersion in TF IDRs enhances transactivation' nicely shows that the effects seen depend on other features such as the spacer sequences and the charge of the backbone, which seems to have higher effect than the prolines. It seems that there would be more potent ways to increase TF activity, modulate transcription potential and allow rational design of a very active TF if ever needed.

Reviewer #1's further comments: I do not think that this point is very relevant to the manuscript.

We agree with Reviewer 1. For Reviewer 3 we note that the aim of the manuscript was to test the activity-specificity trade-off model for transcription factors. In our view, it is an important evolutionary principle that applies beyond transcription factors. As far as implications for designing TF variants, we simply note that immunogenicity in *in vivo* reprogramming experiments is a major hurdle. Therefore, enhancing TF function via slight changes in aromatic dispersion may be more useful in an *in vivo* approach as opposed to e.g., adding *viral* activation domains as Reviewer 3 suggested before.

For sake of clarity, the manuscript is not about modulating proline distribution, but the dispersion of aromatic residues (tyrosine, phenylalanine, tryptophan).

Reviewer #3's comments:

- The RNA-seq experiment done in the section "Evidence for gain-of-function of periodic HOXD4 mutants in vivo" is based on cell-lines that have different levels of hoxd4 protein. Therefore, the changes cannot be attributed to proline periodicity, this needs to be explicitly stated in the section. While the efforts on the extended data figure 6 in principle aims to tackle part of this problem, the presented data comes across as having discrepancies. It seems that aroplus has different levels of hox-gfp than the wildtype (panel f) which is opposite in FACS analysis (panel h). The FACS also does not seem to reflect the representative images in panel g which has for example GFP negative cells in Aroplus that are not observed in FACS profile. In addition, the gene expression changes in panel i are not shown for aroperfect but only for aroplus. Also, Extended Data Figure 6g, needs same LUT and visualization parameters on the representative images.

Reviewer #1's further comments: I agree that it is important to check the protein levels in these experiments to make sure that they are comparable. The authors should address this concern.

In summary, there are no discrepancies in the cited data. We further clarify the experiments, and added additional controls for protein levels.

For sake of clarity, the manuscript is not about modulating ‘proline periodicity’, but the dispersion of aromatic residues (tyrosine, phenylalanine, tryptophan).

We state multiple times in the cited section explicitly that the AroPLUS and AroPERFECT HOXD4 proteins are expressed at higher levels than the wild type in the endogenous knock-in cell lines. For example: “The wild type HOXD4-mEGFP protein was modestly enriched in the nucleus, while AroPERFECT and AroPLUS HOXD4 were expressed at higher levels and formed intense nuclear clusters (Fig. 3a).”

The result that the AroPLUS and AroPERFECT HOXD4 proteins are expressed at higher levels than the wild type in the endogenous knock-in cell lines is an insight, and not a ‘problem’. Since HOXD4 is known to autoregulate its own gene, the higher expression level of the AroPERFECT and AroPLUS HOXD4 suggests that these mutants drive higher expression levels of their own genes, consistent with higher transcriptional activity of the mutants.

There are no ‘discrepancies’ in Extended Data Figure 6. There are multiple experiments performed on different starting material. We made clarifications in the methods/figures to better explain these data.

i) The average expression level of AroPLUS HOXD4-GFP is very similar to the average expression level of the wild type HOXD4-GFP in the cell population. First, in Ext. Data Fig 6f and 6g, *bulk cell population* selected for the marker on the PiggyBac transposon. The cell populations are not clonal. In the cultures, there is a minority of cells in the AroPLUS HOXD4-GFP-expression population that have unusually high GFP fluorescence (this is clear in the images in panel 6g). This phenomenon explains, why on the Western blot there may be a slightly higher level of AroPLUS HOXD4-GFP in the population of cells (panel 6f). For clarity, this is the reason why high-level expressors are filtered out from the granularity analysis shown in Figure 3c. **We added the additional information on the cells to Extended Data Fig. 6f-h).**

ii) The FACS analysis on Extended Data Figure 6h is performed *on clonal cell lines* that were isolated from the PiggyBac population. The lines were isolated to make sure that for the expression analysis shown in panel i only cells that have comparable protein levels are used. We note that we included a “PERFECT” sample on the FACS by mistake which could have contributed to the confusion. **We added these information in the Figure panels (f,g,h), and removed the “PERFECT” part of the plot.**

iii) To make this clear again: the FACS (panel h) was done after isolating clones, on the clones. The imaging (panel g) was done on a bulk population of cells before isolating clones. Therefore, the images in panel g are not representative images of cells that underwent clonal isolation, but representative images of the bulk cell population. **We added clarification in the Figure panels.**

iv) There are no GFP negative cells in the AroPLUS HOXD4-GFP images (panel g). The fluorescence imaging is done on a confocal microscope. The cells clump into dome-shaped colonies, and as a result, not every cell is in focus. When one zooms in, it is clear that every cell contains GFP signal.

v) “In addition, the gene expression changes in panel i are not shown for aroperfect but only for aroplus.” Yes, this is correct. We had technical difficulties with isolating AroPERFECT clonal lines. This is a minor point without any relevance to the overall conclusions.

vi) “Also, Extended Data Figure 6g, needs same LUT and visualization parameters on the representative images.” The fluorescence images that show the knock in cells are all processed the exact same way, have the same LUT, except the parental cells. This is customary to show that the lack of signal in the negative control is not due to thresholding of the signal.

We added additional controls on protein levels, and reproduction of the key experimental data, including protein level controls as Supplementary Figures

We reproduced the key data with independently generated cell populations, and include them as Supplementary Figure 1b-c.

Reviewer #3's comments:

- The *hoxd4* rna-seq section concludes both loss of function and gain of function by solely looking at the up/down regulated genes in these cell-lines, which the authors claim are the direct *hox* targets. The gene expression changes are interpreted as decreased-expression being about disruption of interactions with *pbx* and increased-expression being about increased transactivation function of the engineered proteins. None of these are either shown or tested, and comes across speculative. The variations in the experiment rather show that by changing protein sequence other features, such as protein interactions and protein stability, are changed. There could even be more changes such as posttranslational modifications, folding etc, making the comparisons uninformative.

Reviewer #1's further comments: This is a minor point that the authors ought to be able to address by rephrasing the statements.

The speculation why downregulated genes are observed was added for the specific request of the Reviewer. It is not important. What is important is that we observe 396 upregulated genes (that are downregulated in the knockout), and that the upregulated genes are consistent with the nuclear and cellular phenotypes. In other words, if the Reviewer's idea is correct that changing aromatic dispersion disrupts every conceivable feature of the protein, why and how would that lead to *upregulation* of genes, let alone the autoregulated transgene itself? These genes are downregulated in the knockout.

We clarified the sentence summarizing these data: “These results indicate that increased aromatic dispersion in the HOXD4 IDR is associated with enhanced activity and altered gene specificity which in part appears to be gain-of-function.”

Reviewer #3's comments:

- Extended Data Figure 6.e: 'Cluster 1 and 2 represent the HOXD4 knockout specific genes': these 2 clusters rather seem to be shared with knockout, aroplus and aroperfect? Cluster 5 does not seem to be shared between knockout, aroperfect and aroplus. It is more like opposite

response in wt vs aroplus and aroperfect. Also the statement: 'Cluster 4 represents the genes with differential expression in AroPLUS.' seems not true, they are shared in knockout/aroplus and aroperfect. Could it be that there is a wrong labeling/naming of the clusters?

Reviewer #1's further comments: This is a minor point that the authors ought to be able to address with further clarification.

We clarified the descriptions of the gene clusters in the legends for Extended Data Fig. 6e.

Reviewer #3's comments:

- Extended Data Figure 6f vs 6i: Is IFI16 gene same or increased in aroplus?

Reviewer #1's further comments: This is a minor point that the authors ought to be able to address with further clarification.

Slightly higher. The difference is small, and the experiment is done on the bulk PiggyBac cell population, so we prefer to not make claims on this.

Reviewer #3's comments:

- Figure 3h (and same experiment type done in other sections): for this type of non-normal distribution, it is appropriate to use median, rather than mean. Also, a non-parametric test, e.g. Wilcoxon suits better. The imaging and quantification of this section in methods needs explanation (microscopy: widefield/confocal, image processing: center of the signal on two channels/ projection type/background subtraction, bleed-through correction from cfp to yfp etc.) The signal needs to be calculated also taking CFP-lacI intensity into account, which seems to be different for example in Figure 4d.

Reviewer #1's further comments: This is a minor point that the authors ought to be able to address with additional analysis to improve rigor.

We expanded the description of the methods as requested, and now show median and use Wilcoxon test in Figure 3h and 4d. The differences are clear, and statistically significant regardless of which test is used. As additional controls, quantifications of CFP enrichment in the tethered foci were added as Extended Data Fig. 6j.

Reviewer #3's comments:

- Figure 5f shows that the regions labeled as Aroperfect15-specific have also binding in wildtype. They may simply be under the threshold of peak calling they used but the presence of the signal is clear. This is also true for other ChIPs shown in the manuscript. This is a strong indication that the 'specificity' is not the main phenomenon here.

Reviewer #1's further comments: This is a minor point that the authors ought to be able to address with further analysis that improves rigor.

It is the feature of any genomic data that signal is distributed on a continuum, and peak-calling imposes an arbitrary threshold on that continuum above which a peak is called (using various statistical tests). The dissociation/association constants of biochemical interactions can span a range of values (typically between low nM and low mM in cells), and it is an arbitrary decision what is the magnitude of difference between two constants above which one calls an interaction “specific”. It is the consequence of the underlying biology that read densities in peak regions are not binary, consistent with the nature of the cellular biochemistry that is probed with the experiment. Reviewer 3’s argument implies that there is no such thing as “specificity”.

Our data shows that i) differences in the read densities at the majority of binding sites can be detected with standard, stringent statistical cutoffs, ii) that most sites that show the biggest difference in binding have higher read densities in the AroPERFECT samples, that iii) the sites with the biggest differences in read densities are enriched in canonical motifs of bZIP factors other than C/EBP α , and iv) the differential binding is associated with differences of the expression levels of genes near the differentially bound sites (Figure 5f-k).

Optimizing C/EBP α clearly leads to more promiscuous genomic binding, which is exactly how we phrased the conclusion. We leave it up to the readers to decide whether the data justify using the term “specificity”.

Reviewer #3’s comments:

- Does AroLITE C/EBP α not localize to nucleoplasm (Ext Fig 9d)? Does Extended Data Figure 9d suggest that the levels of aroperfect are higher, or the representative images do not reflect the quantification? It seems that the population with no gfp is absent in aroperfect in panel e.

- Extended figure 10: arolite seems much higher than wt in panel d, which is opposite in panel e.

Reviewer #1’s further comments: This is a minor point that the authors ought to be able to address with further clarification.

Extended Data Figure 9d and 9e show results of two different experiments with two different sets of samples. Extended Data Figure 9d shows images of the bulk transduced cell populations. The GFP levels in the bulk transduced cell populations were quantified with FACS analysis displayed in Extended Data Figure 7e, which shows that there is some heterogeneity in the population and the mean levels are very similar for the bulk cell population. Extended Data Figure 9e shows the FACS analysis of the individual clones that were isolated for the ChIP-Seq analysis. **To prevent confusion, we moved the images of the bulk cell population next to the FACS data on the bulk cell population in Extended Data Fig. 7e.** Indeed, AroLITE C/EBP α does not localize to the nucleoplasm, which the reason those cells were not included in the ChIP-Seq experiments (this was a previous reviewer question).

In Extended Data Figure 10, once again the confusion seems to be caused by the fact that the Western blot in panel d was performed on bulk unsorted cell populations, and the images in panel e are of isolated clonal cell lines. Since we only used clonal cell lines for the

differentiation, RNA-Seq and ChIP assays, **we removed the Western blot, added new images in panel d, and added quantification of the FLAG-NGN2 fluorescence intensity as new panel e.**

Reviewer #3's comments:

- In the quantification of differentiation for ngn2 experiments, aroperfect is claimed to be better at 'differentiating' as opposed to chip-seq pca at Extended figure 10 h. The effect is also seen in Extended figure 10k (wild type more differential than parental as compared to aroperfect, cluster 5&6 shows more similarity between arolite and aroperfect). There are clear differences in heterogeneity as seen by non-differentiating cell populations in the experiments (figure6d). These all could be clonal effects. Thus, the phenotype conclusions in these comparisons seem to be inconclusive and uninformative.

Reviewer #1's further comments: It is not fare to dismiss the observations as clonal effects. If the authors have reproducible observations from multiple clones, that would help address the concerns.

There are significantly more cells that survive the selection in the neural differentiation medium (Fig. 6d-e), and the average density of neurite projections is an average higher in wells that contain the cells expressing the AroPERFECT NGN2 transgene compared to the cells that express the wild type NGN2 transgene (Fig 6d, 6f). The ChIP-Seq PCA is not inconsistent with these findings as the reviewer suggests. The PCA analysis shows that the expression profiles of AroLITE and WT-expressing cells are not entirely the same, and that there are small differences. The differences in the expression profile does not mean that the AroPERFECT was worse at 'differentiating'.

The differentiation assays, morphology analysis was performed on **three independent clones of each transgenic genotype**. We added the results of two clones, in addition to the ones used in Figure 6, and the control quantifications as **new Supplementary Figure 2a-c**.

Reviewer #3's comments:

- The concerns about the protein levels need better addressing and quantification across experiments. For example, with the tools the authors present, it is easy to perform FACS or image analysis of flag staining on Myod induction lines. Many things can lead to loss of function, thus maybe less interesting but increased yet specific potency is often more relevant thus needs for a proof for controlling other variables such as same or lower protein levels for aroperfect/plus.

Reviewer #1's further comments: This is a good point and additional QC data would be very helpful.

We added FLAG Western blots for the MYOD1-induction cells in Extended Data Figure 11a. The average expression level in the PERFECT-expressing cells is similar to the average

level in WT-expressing cells (which is the key comparison). As with other systems, the AroLITE protein is expressed at higher levels than WT.

Reviewer #2:

Remarks to the Author:

The authors have done an excellent job addressing my queries as well as those of the other reviewers.

My final question pertains to the revised title: "Suboptimization of human transcription factors", which I worry may be so vague people won't understand WHAT is suboptimized; perhaps

"Suboptimization of disordered regions in human transcription factors" would strike a balance between being punchy but making clear what the core of the paper is about? This is just a suggestion, though!

We thank the Reviewer for the suggestion. **We revised the first two sentences in the abstract to make this clearer.** We prefer to keep the short title, for the reason that it is the *transcription factor* that is suboptimized and not the IDR! Therefore, the shorter version appears more accurate to us.

We thank the Reviewer for the useful and insightful suggestions throughout the entire revision process, which undoubtedly made the manuscript better.

Reviewer #3:

Remarks to the Author:

While the revised the manuscript claims to address some of the concerns of the reviewers, the main conclusions that would justify highly visible publication remain speculative, although stated very strongly and in very general terms, and in view of this reviewer are insufficiently supported.

The manuscript at its current state somewhat constructs relevance around the TF specificity argument, stating that this resembles weak binding sites in enhancer elements. Changes in DNA sequence can indeed change the specificity for TF binding. For example, if a genomic region is manipulated to have a higher affinity for a certain TF, now the region starts to be bound by this factor even at lower concentration leading to activation in other cell types.

The manuscript shows interesting examples of how distribution of prolines may potentially affect the transcription activation potential of transcription factor IDRs. However, there is still insufficient evidence that specificity is affected by the occasional increase of transcription activation potential by modulating proline distributions in these TFs. Therefore, with its current overinterpretation, I do not consider the manuscript suitable for publication.

It appears that the key concern of Reviewer 3 is that the “main conclusions” are too speculative, regarding “TF specificity”. Below we highlight simple facts that in our view decisively refute Reviewer 3’s argument.

In making the case for reduced specificity of optimized TFs and suggesting a trade-off between activity and specificity (which appears to be the main concern for Reviewer 3), our manuscript includes substantially more data than papers in the published literature that proposed similar ideas. The original reference that introduced the suboptimization concept for enhancer elements included dissection of *one* enhancer element, and demonstration of loss of tissue specificity for *one* enhancer variant using *one* type of assay (Farley et al., Science 2015). Since then, the Farley lab published *one* more enhancer example experimentally dissected with *one* type of assay (Jindal et al., 2023). The other foundational reference for suboptimization included experimental dissection of *one* enhancer element in *Drosophila* embryos with *one* type of assay (Crocker et al., 2015). We experimentally scrutinized over half a dozen transcription factors. Data consistent with altered gene specificity include i) qPCR data for optimizing HOXD4, ii) RNA-Seq, scRNA-Seq, ChIP-Seq and reporter data for optimizing C/EBP α , iii) RNA-Seq and ChIP-Seq data for optimizing NGN2, and iv) RNA-Seq data for optimizing MYOD1. For C/EBP α we show data that the optimized protein more strongly binds low-affinity sites than the wild type protein when they are expressed at similar levels! Of course, the ChIP-Seq data provide more direct mechanistic explanation for the change in specificity than RNA-readouts. Regardless, the level of evidence we provide is far beyond the most important literature references.

We do heed the criticism and concern of Reviewer 3, and have modified the most contentious statements.

For sake of clarity, the manuscript is not about modulating proline distribution, but the dispersion of aromatic residues (tyrosine, phenylalanine, tryptophan).

Finally, we thank Reviewer 3 for the numerous specific comments that helped us further strengthen the data, and clarify the experiments for the readers.

The specific concerns are listed below:

We address each specific comment above, at the Responses to Reviewer 1, because Reviewer 1 provided important context and guidance on the comments by Reviewer 3.

- Since the manuscript is originally focusing on the manipulation of the transcription output potential of the TFs, it would be more adequate if the title suggests something about changing or optimizing the transcription output potential of human transcription factors' rather than 'suboptimization of human transcription factors'. The phrasing of the title simply refer to another paper discussing that the enhancers have mostly weak affinity for TF binding in relation to establishing binding specificity. The manuscript here does not originally tackle the specificity question and has not enough breadth to justify that as a main point/conclusion.

- in the abstract, the use of the term "any biomolecule" is not justified.

- the use of the term 'trade-off' argues that it is disadvantageous for some functions which is not justified in the context of TF's function: the ability to regulate gene expression in correct time, space and dose.

- The referral to the statement 'suggesting that TF IDRs may contribute to transcriptional activity and also to binding specificity in vivo' to support the interpretations of the data seems speculative. With the same logic one would say changing protein concentration changes binding specificity, simply due to the observation that you would get lower signal and dynamic range in a chip-seq experiment with lower protein concentration. Indeed weak enhancer motifs have been suggested to exist in order to respond to a TF upon a certain concentration (e.g. <https://www.biorxiv.org/content/10.1101/2022.05.27.493636v1>, <https://pubmed.ncbi.nlm.nih.gov/27155014/>).

-The section 'increasing aromatic dispersion in TF IDRs enhances transactivation' nicely shows that the effects seen depend on other features such as the spacer sequences and the charge of the backbone, which seems to have higher effect than the prolines. It seems that there would be more potent ways to increase TF activity, modulate transcription potential and allow rational design of a very active TF if ever needed.

- The RNA-seq experiment done in the section "Evidence for gain-of-function of periodic HOXD4 mutants in vivo" is based on cell-lines that have different levels of hoxd4 protein. Therefore, the changes cannot be attributed to proline periodicity, this needs to be explicitly stated in the section. While the efforts on the extended data figure 6 in principle aims to tackle part of this problem, the presented data comes across as having discrepancies. It seems that aroplus has different levels of hox-gfp than the wildtype (panel f) which is opposite in FACS analysis (panel h). The FACS also does not seem to reflect the representative images in panel g which has for example GFP negative cells in Aroplus that are not observed in FACS profile. In addition, the gene expression changes in panel i are not shown for aroperfect but only for aroplus. Also, Extended Data Figure 6g, needs same LUT and visualization parameters on the representative images.

- The hoxd4 rna-seq section concludes both loss of function and gain of function by solely looking at the up/down regulated genes in these cell-lines, which the authors claim are the direct hox targets. The gene expression changes are interpreted as decreased-expression being about disruption of interactions with pbx and increased-expression being about increased transactivation function of the engineered proteins. None of these are either shown or tested, and comes across speculative. The variations in the experiment rather show that by changing protein sequence other features, such as protein interactions and protein stability, are changed. There could even be more changes such as posttranslational modifications, folding etc, making the comparisons uninformative.

- Extended Data Figure 6.e: 'Cluster 1 and 2 represent the HOXD4 knockout specific genes': these 2 clusters rather seem to be shared with knockout, aroplus and aroperfect? Cluster 5 does not seem to be shared between knockout, aroperfect and aroplus. It is more like opposite response in wt vs aroplus and aroperfect. Also the statement: 'Cluster 4 represents the genes with differential expression in AroPLUS.' seems not true, they are shared in knockout/aroplus and aroperfect. Could it be that there is a wrong labeling/naming of the clusters?

- Extended Data Figure 6f vs 6i: Is IFI16 gene same or increased in aroplus?

- Figure 3h (and same experiment type done in other sections): for this type of non-normal distribution, it is appropriate to use median, rather than mean. Also, a non-parametric test, e.g. Wilcoxon suits better. The imaging and quantification of this section in methods needs explanation (microscopy: widefield/confocal, image processing: center of the signal on two channels/ projection type/background subtraction, bleed-through correction from cfp to yfp etc.) The signal needs to be calculated also taking CFP-lacI intensity into account, which seems to be different for example in Figure 4d.

- Figure 5f shows that the regions labeled as Aroperfect15-specific have also binding in wildtype. They may simply be under the threshold of peak calling they used but the presence of the signal is clear. This is also true for other ChIPs shown in the manuscript. This is a strong indication that the 'specificity' is not the main phenomenon here.

- Does AroLITE C/EBPa not localize to nucleoplasm (Ext Fig 9d)? Does Extended Data Figure 9d suggest that the levels of aroperfect are higher, or the representative images do not reflect the quantification? It seems that the population with no gfp is absent in aroperfect in panel e.

- Extended figure 10: arolite seems much higher than wt in panel d, which is opposite in panel e.

- In the quantification of differentiation for *ngn2* experiments, aroperfect is claimed to be better at 'differentiating' as opposed to chip-seq pca at Extended figure 10 h. The effect is also seen in Extended figure 10 k(wild type more differential than parental as compared to aroperfect, cluster 5&6 shows more similarity between arolite and aroperfect). There are clear differences in heterogeneity as seen by non-differentiating cell populations in the experiments(figure6d). These all could be clonal effects. Thus, the phenotype conclusions in these comparisons seem to be inconclusive and uninformative.

- The concerns about the protein levels need better addressing and quantification across experiments. For example, with the tools the authors present, it is easy to perform FACS or image analysis of flag staining on Myod induction lines. Many things can lead to loss of function, thus maybe less interesting but increased yet specific potency is often more relevant thus needs for a proof for controlling other variables such as same or lower protein levels for aroperfect/plus.

References

- Crocker, J., Abe, N., Rinaldi, L., McGregor, A.P., Frankel, N., Wang, S., Alsawadi, A., Valenti, P., Plaza, S., Payre, F., *et al.* (2015). Low affinity binding site clusters confer hox specificity and regulatory robustness. *Cell* *160*, 191-203.
- Jindal, G.A., Bantle, A.T., Solvason, J.J., Grudzien, J.L., D'Antonio-Chronowska, A., Lim, F., Le, S.H., Song, B.P., Ragsac, M.F., Klie, A., *et al.* (2023). Single-nucleotide variants within heart enhancers increase binding affinity and disrupt heart development. *Developmental cell* *58*, 2206-2216 e2205.
- Shoval, O., Sheftel, H., Shinar, G., Hart, Y., Ramote, O., Mayo, A., Dekel, E., Kavanagh, K., and Alon, U. (2012). Evolutionary trade-offs, Pareto optimality, and the geometry of phenotype space. *Science* *336*, 1157-1160.

Decision Letter, second revision:

Our ref: NCB-A50491B

10th January 2024

Dear Dr. Hnisz,

Thank you for submitting your revised manuscript "Suboptimization of human transcription factors" (NCB-A50491B). It has now been seen by the original referees and their comments are below. The reviewers find that the paper has improved in revision, and therefore we'll be happy in principle to publish it in Nature Cell Biology, pending minor revisions to satisfy the referees' final requests and to comply with our editorial and formatting guidelines.

Thank you again for your interest in Nature Cell Biology Please do not hesitate to contact me if you have any questions.

Sincerely,
Daryl

Daryl Jason Verzosa David, PhD

Senior Editor, Nature Cell Biology
Nature Portfolio
Advisory Editor, npj Biological Physics and Mechanics

Heidelberger Platz 3, 14197 Berlin, Germany
Email: daryl.david@nature.com
ORCID: <https://orcid.org/0000-0002-9253-4805>

Reviewer #1 (Remarks to the Author):

The authors have further revised the manuscript and responded to the remaining concerns from the reviewers. In my view the revised manuscript is very strong and the message very timely. I recommend its publication without delay.

Decision Letter, final checks:

Our ref: NCB-A50491B

24th January 2024

Dear Dr. Hnisz,

Thank you for your patience as we've prepared the guidelines for final submission of your Nature Cell Biology manuscript, "Suboptimization of human transcription factors" (NCB-A50491B). Please carefully follow the step-by-step instructions provided in the attached file, and add a response in each row of the table to indicate the changes that you have made. Please also check and comment on any additional marked-up edits we have proposed within the text. Ensuring that each point is addressed will help to ensure that your revised manuscript can be swiftly handed over to our production team.

In recognition of the time and expertise our reviewers provide to Nature Cell Biology's editorial process, we would like to formally acknowledge their contribution to the external peer review of your manuscript entitled "Suboptimization of human transcription factors". For those reviewers who give their assent, we will be publishing their names alongside the published article.

Nature Cell Biology offers a Transparent Peer Review option for new original research manuscripts submitted after December 1st, 2019. As part of this initiative, we encourage our authors to support increased transparency into the peer review process by agreeing to have the reviewer comments, author rebuttal letters, and editorial decision letters published as a Supplementary item. When you submit your final files please clearly state in your cover letter whether or not you would like to participate in this initiative. Please note that failure to state your preference will result in delays in accepting your manuscript for publication.

Cover suggestions

COVER ARTWORK: We welcome submissions of artwork for consideration for our cover. For more information, please see our guide for cover artwork.

Nature Cell Biology has now transitioned to a unified Rights Collection system which will allow our Author Services team to quickly and easily collect the rights and permissions required to publish your work. Approximately 10 days after your paper is formally accepted, you will receive an email in providing you with a link to complete the grant of rights. If your paper is eligible for Open Access, our Author Services team will also be in touch regarding any additional information that may be required

to arrange payment for your article.

Please note that *Nature Cell Biology* is a Transformative Journal (TJ). Authors may publish their research with us through the traditional subscription access route or make their paper immediately open access through payment of an article-processing charge (APC). Authors will not be required to make a final decision about access to their article until it has been accepted. Find out more about Transformative Journals

Please use the following link for uploading these materials:
[Redacted]

Best regards,

Kendra Donahue
Staff
Nature Cell Biology

On behalf of

Daryl Jason Verzosa David, PhD

Senior Editor, Nature Cell Biology
Nature Portfolio
Advisory Editor, npj Biological Physics and Mechanics

Heidelberger Platz 3, 14197 Berlin, Germany

Email: daryl.david@nature.com
ORCID: <https://orcid.org/0000-0002-9253-4805>

Reviewer #1:

Remarks to the Author:

The authors have further revised the manuscript and responded to the remaining concerns from the reviewers. In my view the revised manuscript is very strong and the message very timely. I recommend its publication without delay.

Author Rebuttal, Second Revision:

Response to Reviewer comments

(Author response in blue)

Reviewer #1 (Remarks to the Author):

The authors have further revised the manuscript and responded to the remaining concerns from the reviewers. In my view the revised manuscript is very strong and the message very timely. I recommend its publication without delay.

Thank you for the useful comments and guidance throughout the whole review process!

Final Decision Letter:

Dear Dr Hnisz,

I am pleased to inform you that your manuscript, "An activity-specificity trade-off encoded in human transcription factors", has now been accepted for publication in Nature Cell Biology.

Thank you for sending us the final manuscript files to be processed for print and online production,

and for returning the manuscript checklists and other forms. Your manuscript will now be passed to our production team who will be in contact with you if there are any questions with the production quality of supplied figures and text.

Please note that *Nature Cell Biology* is a Transformative Journal (TJ). Authors may publish their research with us through the traditional subscription access route or make their paper immediately open access through payment of an article-processing charge (APC). Authors will not be required to make a final decision about access to their article until it has been accepted. Find out more about Transformative Journals

Authors may need to take specific actions to achieve compliance with funder and institutional open access mandates. If your research is supported by a funder that requires immediate open access (e.g. according to Plan S principles) then you should select the gold OA route,

and we will direct you to the compliant route where possible. For authors selecting the subscription publication route, the journal's standard licensing terms will need to be accepted, including self-archiving policies. Those licensing terms will supersede any other terms that the author or any third party may assert apply to any version of the manuscript.

If you have not already done so, we strongly recommend that you upload the step-by-step protocols used in this manuscript to the Protocol Exchange (www.nature.com/protocolexchange), an open online resource established by Nature Protocols that allows researchers to share their detailed experimental know-how. All uploaded protocols are made freely available, assigned DOIs for ease of citation and are fully searchable through nature.com. Protocols and Nature Portfolio journal papers in which they are used can be linked to one another, and this link is clearly and prominently visible in the online versions of both papers. Authors who performed the specific experiments can act as primary authors for the Protocol as they will be best placed to share the methodology details, but the Corresponding Author of the present research paper should be included as one of the authors. By uploading your Protocols to Protocol Exchange, you are enabling researchers to more readily reproduce or adapt the methodology you use, as well as increasing the visibility of your protocols and papers. You can also establish a dedicated page to collect your lab Protocols. Further information can be found at www.nature.com/protocolexchange/about

With kind regards,
Daryl

Daryl Jason Verzosa David, PhD

Senior Editor, Nature Cell Biology
Nature Portfolio
Advisory Editor, npj Biological Physics and Mechanics

Heidelberger Platz 3, 14197 Berlin, Germany

Email: daryl.david@nature.com
ORCID: <https://orcid.org/0000-0002-9253-4805>

** Visit the Springer Nature Editorial and Publishing website at www.springernature.com/editorial-and-publishing-jobs for more information about our career opportunities. If you have any questions please click here.**